# Learning to Shape Rewards using a Game of Two Partners

## Abstract

Reward shaping (RS) is a powerful method in reinforcement learning (RL) for overcoming the problem of sparse or uninformative rewards. However, RS typically relies on manually engineered shaping-reward functions whose construction is time-consuming and error-prone. It also requires domain knowledge which runs contrary to the goal of autonomous learning. We introduce Reinforcement Learning Optimising Shaping Algorithm (ROSA), an automated RS framework in which the shaping-reward function is constructed in a novel Markov game between two agents. A reward-shaping agent (Shaper) uses *switching controls* to determine which states to add shaping rewards and their optimal values while the other agent (Controller) learns the optimal policy for the task using these shaped rewards. We prove that ROSA, which easily adopts existing RL algorithms, learns to construct a shaping-reward function that is tailored to the task thus ensuring efficient convergence to high performance policies. We demonstrate ROSA's congenial properties in three carefully designed experiments and show its superior performance against state-of-the-art RS algorithms in challenging sparse reward environments.

## 1 Introduction

Reinforcement learning (RL) offers the potential for autonomous agents to learn complex behaviours without the need for human intervention [39]. Despite the notable success of RL in a variety domains [8, 31, 35, 24], enabling RL algorithms to learn successfully in numerous real-world tasks remains a challenge [41]. A key obstacle to the success of RL algorithms is the requirement of a rich reward signal that can guide the agent towards an optimal policy [7]. In many settings of interest such as physical tasks and video games, rich informative signals of the agent's performance are not readily available [14]. For example, in the video game Super Mario [33], the agent must perform sequences of hundreds of actions while receiving no rewards for it to successfully complete its task. In this setting, the sparse reward provides infrequent feedback of the agent's performance. This leads to RL algorithms requiring large numbers of samples (and high expense) for solving problems [14]. Consequently, there is great need for RL techniques that solve these problems efficiently.

Various biological systems have mechanisms that produce rewards for activities that present little or no *direct* biological utility. Such mechanisms, which have been fashioned over millions of years are designed to provide *intrinsic rewards* to promote behaviours that improve chances of outcomes with biological utility [1]. In RL, reward shaping (RS) is a tool to introduce intrinsic rewards known as *shaping rewards* that supplement environment reward signals. These rewards can encourage exploration and insert structural knowledge in the absence of informative environment rewards which can improve learning outcomes [10]. In general however, RS algorithms assume hand-crafted and domain-specific shaping functions whose construction is typically highly labour intensive. This runs contrary to the aim of autonomous learning. Moreover, poor choices of shaping rewards can *worsen* the agent's performance [9]. To resolve these issues, a useful shaping reward must be obtained autonomously.

Inspired by naturally occurring systems, we tackle the problem of sparse and uninformative rewards by developing a framework that autonomously constructs shaping rewards during learning. Our framework, ROSA, works by introducing an additional RL agent, Shaper, that *adaptively* learns to construct shaping rewards by observing Controller (the agent whose goal is to solve the environment task) while Controller learns to solve its task. This generates tailored shaping rewards without the

need for domain knowledge or manual engineering. The shaping rewards supplement the environment reward and promote effective learning, our framework therefore addresses the key challenges in RS.

The resulting framework is a two-player nonzero-sum Markov game (MG) [34] – an extension of Markov decision process (MDP) that involves *two* independent learners with distinct objectives. In our framework, the two agents that have distinct learning agendas, *cooperate* to achieve Controller's objective. This MG formulation confers various advantages:

**1)** The shaping-reward function is constructed fully autonomously. The game also ensures the shaping reward improves Controller's performance unlike RS methods that can lower performance.

**2)** By learning the shaping-reward function *while* Controller learns its optimal policy, Shaper learns to *adaptively* facilitate Controller's learning and improve Controller's performance.

**3)** Both learning processes converge so Controller learns the optimal value function for its task.

**4)** ROSA learns *subgoals* [28] to decompose complex tasks and promote complex exploration patterns.

An integral component of ROSA is a novel combination of RL and *switching controls* [2, 22]. This enables Controller to quickly determine useful states to learn to add and calibrate shaping rewards (i.e. the states in which adding shaping rewards improve Controller's performance) and disregard others. This is in contrast with an RL controller (i.e. Controller) which must learn its best actions at every state. This leads to Shaper quickly finding shaping rewards that guide Controller's learning process toward optimal trajectories (and away from suboptimal trajectories, c.f. Experiment 1).

For our two-player framework to succeed we have to overcome several obstacles. Solving MGs involves finding a stable point in which each player responds optimally to the actions of the other. In our MG, this stable point describes a pair of policies for which Shaper introduces a shaping reward that improves performance and, with that, Controller executes an optimal policy for the task. Tractable methods for solving MGs are rare with convergence of MG methods being seldom guaranteed except in a few special cases [42]. Nevertheless, using special features in the design of our game, we prove the existence of a stable point solution of our MG. We then prove the convergence of our learning method to the solution of the game and show that the solution coincides with the solution of the Controller's problem. This ensures Shaper learns a shaping-reward function that improves Controller's performance and that Controller learns the optimal value function for the task.

## 2 RELATED WORK

**Reward Shaping (RS)** adds a *shaping function* $F$ to supplement the agent's reward to boost learning. RS however has some critical limitations. First, RS does not offer a means of finding $F$. Second, poor choices of $F$ can *worsen* the agent's performance [9]. Last, adding shaping rewards can change the underlying problem therefore generating policies that are completely irrelevant to the task [20]. In [27] it was established that potential-based reward shaping (PBRS) which adds a shaping function of the form $F(s_{t+1}, s_t) = \gamma\phi(s_{t+1}) - \phi(s_t)$ preserves the optimal policy of the problem. Recent variants of PBRS include potential-based advice which defines $F$ over the state-action space [13] and approaches that include time-varying shaping functions [11]. Although the last issue can be addressed using potential-based reward shaping (PBRS) [27], the first two issues remain. To avoid manual engineering of $F$, useful shaping rewards must be obtained autonomously. Towards this [45] introduce an RS method that adds a shaping-reward function prior which fits a distribution from data obtained over many tasks. Recently, [16] use a bilevel technique to learn a scalar coefficient for an already-given shaping-reward function. Nevertheless, constructing $F$ *while training* can produce convergence issues since the reward function now changes during training. [17]. Moreover, while $F$ is being learned the reward can be corrupted by inappropriate signals that hinder learning.

**Curiosity based reward shaping** aims to encourage the agent to explore states by rewarding the agent for novel state visitations using exploration heuristics. One approach is to use state visitation counts [29]. More elaborate approaches such as [6] introduce a measure of *state novelty* using the prediction error of features of the visited states from a random network. [30] use the prediction error of the next state from a learned dynamics model and [15] maximise the information gain about the agent's belief of the system dynamics. In general, these methods provide no performance guarantees nor do they ensure the optimal policy (of the underlying MDP) is preserved. Moreover, they naively reward exploration to unvisited states without consideration of the environment reward. This can lead to spurious objectives being maximised (see Experiment 3 in §6).

Within these two categories, closest to our work are bilevel approaches for learning the shaping function [16, 36]. Unlike [16] which requires a useful shaping reward to begin with, ROSA constructs

a shaping reward function from scratch leading to a fully autonomous method. Moreover, in [16, 36], the agent's policy and shaping rewards are learned with *consecutive* updates. In contrast, ROSA performs these operations *concurrently* leading to a faster, more efficient procedure. Also in contrast to [16, 36], ROSA learns shaping rewards only at relevant states, this confers high computational efficiency (see Experiment 2, §6)). As we describe, ROSA, which successfully *learns the shaping-reward function* $F$, uses a similar form as PBRS. However in ROSA, $F$ is augmented to include the actions of another RL agent to learn the shaping rewards online. Lastly, unlike curiosity-based methods e.g., [6, 30], our method preserves the agent's optimal policy for the task (see Experiment 3, §6) and introduces intrinsic rewards that promote complex learning behaviour (see Experiment 1, §6) .

## 3    PRELIMINARIES & NOTATIONS

In RL, an agent sequentially selects actions to maximise its expected returns. The underlying problem is typically formalised as a MDP $\langle \mathcal{S}, \mathcal{A}, P, R, \gamma \rangle$ where $\mathcal{S}$ is the set of states, $\mathcal{A}$ is the discrete set of actions, $P : \mathcal{S} \times \mathcal{A} \times \mathcal{S} \to [0, 1]$ is a transition probability function describing the system's dynamics, $R : \mathcal{S} \times \mathcal{A} \to \mathbb{R}$ is the reward function measuring the agent's performance, and the factor $\gamma \in [0, 1)$ specifies the degree to which the agent's rewards are discounted over time [39]. At time $t$ the system is in state $s_t \in \mathcal{S}$ and the agent must choose an action $a_t \in \mathcal{A}$ which transitions the system to a new state $s_{t+1} \sim P(\cdot|s_t, a_t)$ and produces a reward $R(s_t, a_t)$. A policy $\pi : \mathcal{S} \times \mathcal{A} \to [0, 1]$ is a probability distribution over state-action pairs where $\pi(a|s)$ represents the probability of selecting action $a \in \mathcal{A}$ in state $s \in \mathcal{S}$. The goal of an RL agent is to find a policy $\pi^\star \in \Pi$ that maximises its expected returns given by the value function: $v^\pi(s) = \mathbb{E}[\sum_{t=0}^\infty \gamma^t R(s_t, a_t)|a_t \sim \pi(\cdot|s_t)]$ where $\Pi$ is the agent's policy set. We denote this MDP by $\mathfrak{M}$.

**A two-player Markov game (MG)** is an augmented MDP involving two agent that simultaneously take actions over many rounds [34]. In the classical MG framework, each agent's rewards and the system dynamics are now influenced by the actions of *both* agents. Therefore, each agent $i \in \{1, 2\}$ has its reward function $R_i : \mathcal{S} \times \mathcal{A}_1 \times \mathcal{A}_2 \to \mathbb{R}$ and action set $\mathcal{A}_i$ and its goal is to maximise its *own* expected returns. The system dynamics, now influenced by both agents, are described by a transition probability $P : \mathcal{S} \times \mathcal{A}_1 \times \mathcal{A}_2 \times \mathcal{S} \to [0, 1]$. As we discuss in the next section, ROSA induces a specific MG in which the dynamics are influenced by *only* Controller.

In RS the question of which $\phi$ to insert has not been addressed. Moreover it has been shown that poor choices of $\phi$ *hinder* learning [9]. Consequently, in general RS methods rely on hand-crafted shaping-reward functions that are constructed using domain knowledge (whenever available). In the absence of a useful shaping-reward function $F$, the challenge is to *learn* a shaping-reward function that leads to more efficient learning while preserving the optimal policy. Naturally, we can formalise the problem of learning such an $F$ by constructing $F$ as a parametric function of $\boldsymbol{\theta} \in \mathbb{R}^m$: $\hat{F}(s_{t+1}, s_t; \boldsymbol{\theta}) \coloneqq \gamma \hat{\phi}(s_{t+1}, \boldsymbol{\theta}) - \hat{\phi}(s_t, \boldsymbol{\theta})$. Now the problem is to find $\boldsymbol{\theta}^\star \in \mathbb{R}^m$ for $\phi(s) = \hat{\phi}(s, \boldsymbol{\theta}^\star)$ such that $F(s_{t+1}, s_t) = \hat{F}(s_{t+1}, s_t; \boldsymbol{\theta}^\star)$, i.e., we aim to find $\boldsymbol{\theta}^\star$ that yields a useful shaping-reward function. Determining this function is a significant challenge; poor choices can hinder the learning process, moreover attempting to learn the shaping-function while learning the RL agent's policy presents convergence issues given the two concurrent learning processes [44]. Another issue is that using an optimisation procedure to find $\boldsymbol{\theta}^\star$ directly does not make use of information generated by intermediate state-action-reward tuples of the RL problem which can help to guide the optimisation.

## 4    OUR FRAMEWORK

We now describe the problem setting, details of our framework, and how it learns the shaping-reward function. We then describe Controller's and Shaper's objectives. We also describe the switching control mechanism used by Shaper and the learning process for both agents.

To tackle the challenges described above, we introduce Shaper an *adaptive* agent with its own objective that determines the best shaping rewards to give to Controller. Using observations of the actions taken Controller, Shaper's goal is to construct shaping rewards (which the RL learner Controller cannot generate itself) to guide the Controller towards quickly learning its optimal policy. To do this, Shaper learns how to choose the values of an shaping-reward at each state. Simultaneously, Controller performs actions to maximise its rewards using its own policy. Crucially, the two agents tackle distinct but complementary set problems. The problem for Controller is to learn to solve the

task by finding its optimal policy, the problem for Shaper is to learn how to add shaping rewards to aid Controller. The objective for Controller is given by:

$$v_1^{\pi,\pi^2}(s) = \mathbb{E}\left[\sum_{t=0}^\infty \gamma^t \left(R(s_t, a_t) + \hat{F}(s_t, a_t^2, s_{t-1}, a_{t-1}^2)\right) \Big| s = s_0\right],$$

where $a \sim \pi$ is Controller's action, $\hat{F}$ is the shaping-reward function which is given by $\hat{F}(\cdot) \equiv \phi(s_t, a_t^2) - \gamma^{-1}\phi(s_{t-1}, a_{t-1}^2)$ for any $s_t, s_{t-1} \in \mathcal{S}$ and $a_t^2 \sim \pi^2$ is chosen by Shaper (and $a_t^2 \equiv 0, \forall t < 0$) using the policy $\pi^2 : \mathcal{S} \times \mathcal{A}_2 \to [0, 1]$ where $\mathcal{A}_2 \subset \mathbb{R}^p$ is the action set for Shaper. The function $\phi : \mathcal{S} \times \mathcal{A}_2 \to \mathbb{R}$ is a continuous map that satisfies the condition $\phi(s, 0) \equiv 0$ for any $s \in \mathcal{S}$ ($\phi$ can be, for example, a neural network with fixed weights with input $(s, a^2)$, $\mathcal{A}_2$ can be for example a set of integers $\{1, \ldots, K\}$). Therefore, Shaper determines the output of $\hat{F}$ (which it does through its choice of $a_t^2$). With this, Shaper constructs a shaping-reward function which is tailored for the specific setting. The transition probability $P : \mathcal{S} \times \mathcal{A} \times \mathcal{S} \to [0, 1]$ takes the state and *only* Controller's actions as inputs. Formally, the MG is defined by a tuple $\mathcal{G} = \langle \mathcal{N}, \mathcal{S}, \mathcal{A}, \mathcal{A}_2, P, \hat{R}_1, \hat{R}_2, \gamma \rangle$ where the new elements are $\mathcal{N} = \{1, 2\}$ which is the set of agents, $\hat{R}_1 := R + \hat{F}$ is the new Controller reward function which now contains a shaping reward $\hat{F}$, the function $\hat{R}_2 : \mathcal{S} \times \mathcal{A} \times \mathcal{A}_2 \to \mathbb{R}$ is the one-step reward for Shaper (we give the details of this function later) and lastly the transition probability $P : \mathcal{S} \times \mathcal{A} \times \mathcal{S} \to [0, 1]$ takes the state and *only* Controller action as inputs.

As Controller's policy can be learned using any RL method, ROSA easily adopts any existing RL algorithm for Controller. Note that unlike reward-shaping methods e.g. [27], the function $\phi$ now contains an action term $a^2$ which is chosen by Shaper which enables the best shaping-reward function to be learned online. The presence of an action term may spoil the policy invariance result in [27]. We however prove an policy invariance result (Prop. 1) analogous to that in [27] and show RIGA preserves the optimal policy for $\mathfrak{M}$.

## 4.1 Switching Controls

So far Shaper's problem involves learning to construct shaping rewards at *every* state including those that are irrelevant for guiding Controller. In order to increase the (computational) efficiency of Shaper's learning process, we now replace the policy space for Shaper with a form of policies known as *switching controls*. This enables Shaper to decide at which states to learn the value of shaping rewards it would like to add. Therefore, now Shaper is tasked with learning how to shape Controller's rewards *only* at states that are important for guiding Controller to its optimal policy. This enables Shaper to quickly determine its policy $\pi^2$ and how to choose the values of $F$ unlike Controller whose policy must learned for all states.

Now at each state Shaper first makes a *binary decision* to decide to *switch on* its shaping reward $F$ for Controller affecting a switch $I_t$ which takes values in $\{0, 1\}$. This leads to an MG in which, unlike classical MGs, Shaper now uses *switching controls* to perform its actions.

The new Controller objective is: $v_1^{\pi,\pi^2}(s_0, I_0) = \mathbb{E}\left[\sum_{t=0}^\infty \gamma^t \left\{R(s_t, a_t) + \hat{F}(s_t, a_t^2; s_{t-1}, a_{t-1}^2)I_t\right\}\right]$, where $I_{\tau_{k+1}} = 1 - I_{\tau_k}$, which is the switch for the shaping rewards which is 0 or 1 (and $I_t \equiv 0, \forall t \leq 0$) and $\{\tau_k\}$ are times that a switch takes place for example if the switch is first turned on at state $s_5$ then turned off at $s_7$, then $\tau_1 = 5$ and $\tau_2 = 7$ (we will shortly describe these in detail). The switch $I_t$ is managed by Shaper, therefore by switching $I_t$ between 0 or 1, Shaper activates or deactivates the shaping reward.

We now describe how at each state both the decision to activate a shaping reward and their magnitudes are determined. Recall that $a_t^2 \sim \pi^2$ determines the shaping reward through $F$. At any $s_t$, the decision to turn on $I_t$ and shape rewards is decided by a (categorical) policy $\mathfrak{g}_2 : \mathcal{S} \to \{0, 1\}$. Therefore, $\mathfrak{g}_2$ determines whether a (or no) Shaper policy $\pi^2$ should be used to execute an action $a_t^2 \sim \pi^2$. It can now be seen the sequence of times $\tau_k = \inf\{t > \tau_{k-1} | s_t \in \mathcal{S}, \mathfrak{g}_2(s_t) = 1\}$ are **rules** *that depend on the state*. Hence, by learning an optimal $\mathfrak{g}_2$, Shaper learns the best states to activate $F$.

**Summary of events:**

At a time $t \in 0, 1 \ldots$

- | Both players make an observation of the state $s_t \in \mathcal{S}$.
- | Controller takes an action $a_t$ sampled from its policy $\pi$.
- | Shaper decides whether or not to activate the shaping reward using $\mathfrak{g}_2 : \mathcal{S} \to \{0, 1\}$
- | If $\mathfrak{g}_2(s_t) = 0$:
     - The switch is not activated ($I_t = 0$). Controller receives a reward $r \sim R(s_t, a_t)$ and the system transitions to the next state $s_{t+1}$.
- | If $\mathfrak{g}_2(s_t) = 1$:
     - Shaper takes an action $a_t^2$ sampled from its policy $\pi^2$.
     - The switch is activated ($I_t = 1$), Controller receives a reward $R(s_t, a_t) + \hat{F}(s_t, a_t^2; s_{t-1}, a_{t-1}^2) \times 1$ and the system transitions to the next state $s_{t+1}$.

We set $\tau_k \equiv 0 \forall k \leq 0$ and $a_{\tau_k}^2 \equiv 0, \forall k \in \mathbb{N}$ ($a_{\tau_k+1}^2, \ldots, a_{\tau_{k+1}-1}^2$ remain non-zero) and $a_k^2 \equiv 0 \ \forall k \leq 0$ and use the shorthand $I(t) \equiv I_t$ .

## 4.2 THE SHAPER'S OBJECTIVE

The goal of Shaper is to guide Controller to efficiently learn to maximise its own objective (given in $\mathfrak{M}$). The shaping reward $F$ is activated by switches controlled by Shaper. As we later describe, the termination times occur according to some external (probabilistic) rule. To induce Shaper to selectively choose when to switch on the shaping reward, each switch activation incurs a fixed cost for Shaper. The cost has two effects: first it reduces the complexity of Shaper problem since its decision space is to determine which *subregions* of $\mathcal{S}$ it should activate the shaping rewards (and their magnitudes). Second, it ensures that the *information-gain* from Shaper encouraging Controller to explore a given set of states is sufficiently high to merit activating the stream of rewards. Given these remarks the objective for Shaper is given by

$$v_2^{\pi,\pi^2}(s_0, I_0) = \mathbb{E}_{\pi,\pi^2} \left[ \sum_{t=0}^{\infty} \gamma^t \left( \hat{R}_1(s_t, I_t, a_t, a_t^2, a_{t-1}^2) + \sum_{k \geq 1}^{\infty} c(I_t, I_{t-1}) \delta_{\tau_{2k-1}}^t + L(s_t) \right) \right]. \quad (1)$$

The objective encodes Shaper agenda, namely to maximise the expected return.[1] Therefore, using its shaping rewards, Shaper seeks to guide Controller towards optimal trajectories (potentially away from suboptimal trajectories, c.f. Experiment 1) and enable Controller to learn faster (c.f. Cartpole experiment in Sec. 6). The function $c : \{0, 1\}^2 \to \mathbb{R}_{<0}$ is a strictly negative cost function which imposes a cost for each switch and is modulated by the Kronecker-delta function $\delta_{\tau_{2k-1}}^t$ which is 1 whenever $t = \tau_{2k-1}$ and 0 otherwise (this restricts the costs to only the points at which the shaping reward is activated). Lastly, the term $L : \mathcal{S} \to \mathbb{R}$ is a Shaper *bonus reward* for when Controller visits infrequently visited states and tends to 0 as the states are revisited. For this there are various possibilities e.g. model prediction error [37], count-based exploration bonus [38].

With this, Shaper constructs a shaping-reward function that supports Controller's learning which is tailored for the specific setting. This avoids inserting hand-designed exploration heuristics into Controller's objective as in curiosity-based methods [6, 30] and classical reward shaping [27]. We later prove that with this objective, Shaper's optimal policy maximises Controller's (extrinsic) return (Prop. 1). Additionally, we show that the framework preserves the optimal policy of $\mathfrak{M}$.

There are various possibilities for the *termination* times $\{\tau_{2k}\}$ (recall that $\{\tau_{2k+1}\}$ are the times which the shaping reward $F$ is *switched on* using $\mathfrak{g}_2$). One is for Shaper to determine the sequence. Another is to build a construction of $\{\tau_{2k}\}$ that directly incorporates the information gain that a state visit provides — we defer the details of this arrangement to Sec. 10 of the Appendix.

## 4.3 THE OVERALL LEARNING PROCEDURE

The game $\mathcal{G}$ is solved using our multi-agent RL algorithm (ROSA). In the next section, we show the convergence properties of ROSA. Here, we first give a description of ROSA (the full code is in Sec. 8 of the Appendix). The ROSA algorithm consists of two independent procedures: Controller learns its own policy while Shaper learns which states to perform a switch and the shaping reward magnitudes. In our implementation, we used proximal policy optimization (PPO) [32] as the learning algorithm

---

[1]Note that we can now see that $\hat{R}_2 \equiv R(s_t, a_t) + \hat{F}(s_t, a_t^2; s_{t-1}, a_{t-1}^2)I_t + \sum_{k \geq 1}^{\infty} c(I_t, I_{t-1})\delta_{\tau_{2k-1}}^t$.

for all policies: Controller's policy, switching control policy, and the reward magnitude policy. For Shaper $L$ term we used $L(s_t) := \|\hat{h}(s_t) - h(s_t)\|_2^2$ as in RND [6] where $h$ is a random initialised, fixed target network while $\hat{h}$ is the predictor network that seeks to approximate the target network. We constructed $\hat{F}$ using a fixed neural network $f : \mathbb{R}^d \mapsto \mathbb{R}^m$ and a one-hot encoding of the action of Shaper. Specifically, $\hat{\phi}(s_t, a_t^2) := f(s_t) \cdot i(a_t^2)$ where $i(a_t^2)$ is a one-hot encoding of the action $a_t^2$ picked by Shaper. Thus, $\hat{F}(s_t, a_t^2; s_{t-1}, a_{t-1}^2) = f(s_t) \cdot i(a_t^2) - \gamma^{-1} f(s_{t-1}) \cdot i(a_{t-1}^2)$. The action set of Shaper is thus $\mathcal{A}_2 := \{0, 1, ..., m\}$ where each element is an element of $\mathbb{N}$, and $\pi_2$ is a MLP $\pi_2 : \mathbb{R}^d \mapsto \mathbb{R}^m$. Precise details are in the Supplementary Materials Section 8.

---

**Algorithm 1: R**einforcement Learning **O**ptimising **S**haping **A**lgorithm (ROSA)

---

**Input:** Initial Controller policy $\pi_0$, Shaper policies $\mathfrak{g}_{2_0}, \pi_0^2$, RL learning algorithm $\Delta$
**Output:** Optimised Controller policy $\pi^*$

1  **for** $t = 1, T$ **do**
2      Given environment state $s_t$, sample $a_t$ from $\pi(s_t)$ and obtain $s_{t+1}, r_{t+1}$ by applying $a_t$ to environment
3      Evaluate $\mathfrak{g}_2(s_t)$ according to Prop. 2
4      **if** $\mathfrak{g}_2(s_t) = 1$ **then**
5          Shaper samples an action $a_{t+1}^2 \sim \pi^2(\cdot|s_{t+1})$
6          Shaper computes $r_{t+1}^i = \hat{F}(s_t, a_t^2, s_{t+1}, a_{t+1}^2)$,
7          Set shaped reward $r = r_{t+1} + r_{t+1}^i$
8      **else**
9          Set $r = r_{t+1}$
10      Update $\pi, \mathfrak{g}_2, \pi^2$ using $(s_t, a_t, r, s_{t+1})$ and $\Delta$ **// Learn the individual policies**

---

## 5 CONVERGENCE AND OPTIMALITY OF OUR METHOD

The ROSA framework enables Shaper to learn a shaping-reward function with which Controller can learn the optimal policy for the task. The interaction between the two RL agents induces two concurrent learning processes which can occasion convergence issues [44]. We now show that our method converges and the solution ensures higher performing Controller policy than would be achieved by solving $\mathfrak{M}$ directly. To do this, we first study the stable point solutions of $\mathcal{G}$.

Unlike MDPs, the existence of a stable point solution in Markov policies is not guaranteed for MGs [5] and is rarely computable.[2] MGs also often have multiple stable points that can be inefficient [25]; in $\mathcal{G}$ the outcome of such stable point profiles would be a poor performing Controller policy. To ensure the framework is useful, we must verify that the solution of $\mathcal{G}$ corresponds to $\mathfrak{M}$. We solve these challenges with the following scheme: **[A]** The method preserves the optimal policy of $\mathfrak{M}$. **[B]** A stable point of the game $\mathcal{G}$ in Markov policies exists and is the convergence point of ROSA. **[C]** The convergence point of ROSA yields a payoff that is (weakly) greater than that from solving $\mathfrak{M}$ directly. **[D]** ROSA converges to the stable point solution of $\mathcal{G}$.

We now prove [A] which shows the solution to $\mathfrak{M}$ is preserved under the influence of Shaper:

**Proposition 1** *The following statements hold:*

    *i)* $\displaystyle\max_{\pi \in \Pi} v_1^{\pi, \pi^2}(s) = \max_{\pi \in \Pi} v_1^\pi(s), \ \ \forall s \in \mathcal{S}, \forall \pi^2 \in \Pi^2$, *where* $v_1^\pi(s) = \mathbb{E}\left[\sum_{t=0}^\infty \gamma^t R(s_t, a_t)\right]$.

    *ii)* *The Shaper's optimal policy maximises* $v_1^\pi(s)$ *for any* $s \in \mathcal{S}$.

Result (i) says that the Controller's problem is preserved under the influence of the Shaper. Moreover the (expected) total return received by the agents is that from the environment (extrinsic rewards). Result (ii) establishes that Shaper's optimal policy induces Shaper to maximise its (Controller's) extrinsic total return. The result is established by a careful adaptation of the policy invariance result in [27] to our multi-agent switching control framework in which the shaping reward is no longer added at all states. Building on Prop. 1, we deduce the following result:

---

[2] Special exceptions are *team* MGs where agents share an objective and *zero-sum* MGs [34].

**Corollary 1** *ROSA preserves the MDP $\mathfrak{M}$. In particular, let $(\hat{\pi}^1, \hat{\pi}^2)$ be a stable point policy profile[3] of the MG induced by ROSA $\mathcal{G}$ then $\hat{\pi}^1$ is a solution to the MDP, $\mathfrak{M}$.*

Therefore, the introduction of Shaper does not alter the fundamentals of the problem.

We now show that $\mathcal{G}$ belongs to a special class of MGs which possess a stable point Markov policies which can be computed as a limit point of a sequence of Bellman operations. We later exploit this result to prove the convergence of ROSA.

We begin by defining some objects which are central to the analysis. For any $\pi \in \Pi$ and $\pi^2 \in \Pi^2$, given a function $V^{\pi, \pi^2} : \mathcal{S} \times \mathbb{N} \to \mathbb{R}$, we define the *intervention operator* $\mathcal{M}^{\pi, \pi^2}$ by $\mathcal{M}^{\pi, \pi^2} V^{\pi, \pi^2}(s_{\tau_k}, I(\tau_k)) := \hat{R}_1(s_{\tau_k}, I(\tau_k), a_{\tau_k}, a^2_{\tau_k}, \cdot) + c(I_k, I_{k-1}) + \gamma \sum_{s' \in \mathcal{S}} P(s'; a_{\tau_k}, s) V^{\pi, \pi^2}(s', I(\tau_{k+1}))$ for any $s_{\tau_k} \in \mathcal{S}$ and $\forall \tau_k$ where $a_{\tau_k} \sim \pi(\cdot | s_{\tau_k})$ and where $a^2_{\tau_k} \sim \pi^2(\cdot | s_{\tau_k})$. We define the Bellman operator $T$ of the game $\mathcal{G}$ by $TV^{\pi, \pi^2}(s_{\tau_k}, I(\tau_k)) := \max \left\{ \mathcal{M}^{\pi, \pi^2} \Lambda(s_{\tau_k}, I(\tau_k)), R(s_{\tau_k}, a) + \gamma \max_{a \in \mathcal{A}} \sum_{s' \in \mathcal{S}} P(s'; a, s_{\tau_k}) \Lambda(s', I(\tau_k)) \right\}$. Given a value function $\{V_i\}_{i \in \{1, 2\}}$, the quantity $\mathcal{M}V_i$ measures the expected future stream of rewards for agent $i$ after an immediate switch minus the cost of switching.

We now show that a stable point of $\mathcal{G}$ can be computed using dynamic programming:

**Theorem 1** *Let $V : \mathcal{S} \times \mathbb{N} \to \mathbb{R}$ then the game $\mathcal{G}$ has a stable point which is a given by $\lim_{k \to \infty} T^k V^{\boldsymbol{\pi}} = \sup_{\hat{\boldsymbol{\pi}} \in \Pi} V^{\hat{\boldsymbol{\pi}}} = V^{\boldsymbol{\pi}^\star}$, where $\hat{\boldsymbol{\pi}}$ is a stable policy profile for the MG, $\mathcal{G}$.*

Theorem 1 proves that the MG $\mathcal{G}$ (which is the game that is induced when Shaper influences Controller) has a stable point which is the limit of a dynamic programming method. In particular, it proves the that the stable point of $\mathcal{G}$ is the limit point of the sequence $T^1 V, T^2 V, \ldots,$. Crucially, (by Corollary 1) the limit point corresponds to the solution of the MDP $\mathcal{M}$. Theorem 1 is proven by firstly proving that $\mathcal{G}$ has a dual representation as an MDP whose solution corresponds to the stable point of the MG. Theorem 1 enables a distributed Q-learning method [4] to tractably solve the MG.

Having constructed a procedure to find the optimal Controller policy, our next result characterises Shaper policy $\mathfrak{g}_2$ and the optimal times to activate $F$.

**Proposition 2** *The policy $\mathfrak{g}_2$ is given by the following expression: $\mathfrak{g}_2(s_t) = H(\mathcal{M}^{\pi, \pi^2} V^{\pi, \pi^2} - V^{\pi, \pi^2})(s_t, I_t), \quad \forall (s_t, I_t) \in \mathcal{S} \times \{0, 1\}$, where $V$ is the solution in Theorem 1 and $H$ is the Heaviside function, moreover Shaper's switching times are $\tau_k = \inf\{\tau > \tau_{k-1} | \mathcal{M}^{\pi, \pi^2} V^{\pi, \pi^2} = V^{\pi, \pi^2}\}$.*

Hence, Prop. 2 also characterises the (categorical) distribution $\mathfrak{g}_2$. Moreover, given the function $V$, the times $\{\tau_k\}$ can be determined by evaluating if $\mathcal{M}V = V$ holds. In general, introducing shaping rewards may undermine learning and worsen overall performance. We now prove that the ROSA framework introduces an shaping rewards that yield higher total (environment) returns for Controller as compared to solving $\mathfrak{M}$ directly ([C]).

**Proposition 3** *Controller's (extrinsic) expected return $v_1^{\pi, \pi^2}$ whilst playing $\mathcal{G}$ is (weakly) higher than $v_1^{\pi}$, the (extrinsic) expected return for $\mathfrak{M}$ i.e. $v_1^{\pi, \pi^2}(s, \cdot) \geq v_1^{\pi}(s), \ \forall s \in \mathcal{S}$.*

Prop. 3 shows that the stable point of $\mathcal{G}$ improves outcomes for Controller. Unlike reward shaping methods in general, the stable points generated *never* lead to a reduction to the total (environment) return for Controller as compared to its total return without $F$. Note that by Prop. 1, Theorem 3 compares the environment (extrinsic) rewards accrued by the agents so that the presence of Shaper increases the total expected environment rewards.

We now prove the convergence to the solution with (linear) function approximators. In what follows, we define a *projection* $\Pi$ on a function $\Lambda$ by: $\Pi\Lambda := \arg\min_{\bar{\Lambda} \in \{\Psi r | r \in \mathbb{R}^p\}} \|\bar{\Lambda} - \Lambda\|$.

**Theorem 2** *ROSA converges to the stable point of $\mathcal{G}$, moreover, given a set of linearly independent basis functions $\Psi = \{\psi_1, \ldots, \psi_p\}$ with $\psi_k \in L_2, \forall k$, ROSA converges to a limit point $r^\star \in \mathbb{R}^p$ which*

---

[3]By stable point profile we mean a configuration in which no agent can increase their expected return by deviating unilaterally from their policy given the other agents' policies, i.e. a Markov perfect equilibrium [12].

*is the unique solution to* $\Pi\mathfrak{F}(\Psi r^\star) = \Psi r^\star$ *where* $\mathfrak{F}$ *is defined by:* $\mathfrak{F}\Lambda := \hat{R}_1 + \gamma P \max\{\mathcal{M}\Lambda, \Lambda\}$ *where* $r^\star$ *satisfies:* $\|\Psi r^\star - Q^\star\| \leq (1-\gamma^2)^{-1/2} \|\Pi Q^\star - Q^\star\|$.

Theorem 2 establishes the solution to $\mathcal{G}$ can be computed using ROSA. This means that Shaper converges to a shaping-reward function that necessarily improves Controller's performance and Controller learns the optimal value function for the task. Secondly, the theorem establishes the convergence of ROSA to the solution using (linear) function approximators. Lastly, the approximation error is bounded by the smallest error that can be achieved given the basis functions.

## 6 EXPERIMENTS

We performed a series of experiments to test if ROSA **(1)** learns a beneficial shaping-reward function **(2)** decomposes complex tasks into sub-goals, and **(3)** tailors shaping rewards to encourage Controller to capture environment rewards (as opposed to merely pursuing novelty). In these tasks, we compared the performance of our method to random network distillation (RND) [6], intrinsic curiosity module (ICM) [30], learning intrinsic reward policy gradient (LIRPG) [43], bi-level optimization of parameterized reward shaping (BiPaRS-IMGL) [16][4] and vanilla PPO [32]. We then compared our method against these baselines on performance benchmarks including Sparse Cartpole, Gravitar, Solaris, and Super Mario. Lastly, we ran a detailed suite of ablation studies (supplementary material).

**1. Beneficial shaping reward.** Our method is able to learn a shaping-reward function that leads to improved Controller performance. In particular, it is able to learn to shape rewards that encourage the RL agent to avoid suboptimal – but easy to learn – policies in favour of policies that attain the maximal return. To demonstrate this, we designed a Maze environment with two terminal states: a suboptimal goal state that yields a reward of $0.5$ and an optimal goal state which yields a reward of $1$. In this maze design, the sub-optimal goal is more easily reached.

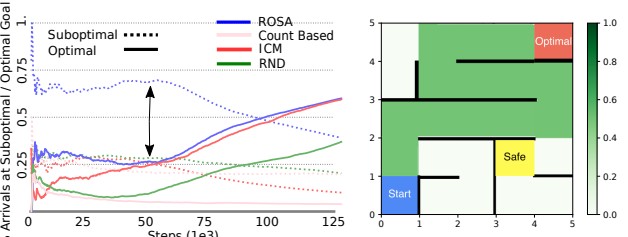

Figure 1: *Left.* Proportion of optimal and suboptimal goal arrivals. Our method has a marked inflection (arrow) where arrivals at the sub-optimal goal decrease and arrivals at the optimal goal increase. Shaper has learned to guide Controller to forgo the suboptimal goal in favour of the optimal one. *Right.* Heatmap showing where our method adds rewards.

A good shaping-reward function discourages the agent from visiting the sub-optimal goal. As shown in Fig. 1[5] our method achieves this by learning to place high shaping rewards (dark green) on the path that leads to the optimal goal.

**2. Subgoal discovery.** We used the Subgoal Maze introduced in [21] to test if ROSA can discover sub-goals. The environment has two rooms separated by a gateway. To solve this, the agent has to discover the subgoal of reaching the gateway before it can reach the goal. Rewards are $-0.01$ everywhere except at the goal state where the reward is $1$. As shown in Fig. 2, our method successfully solves this environment whereas other methods fail. Our method assigns importance to reaching the gateway, depicted by the heatmap of added shaped rewards.

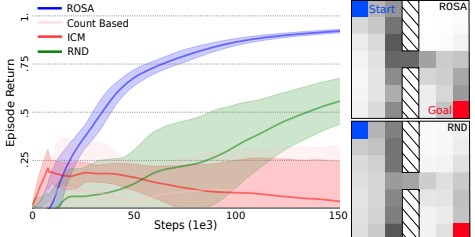

Figure 2: Discovering subgoals on Subgoal Maze. *Left.* Learning curves. *Right.* Heatmap of shaping rewards guiding Controller to gate.

**3. Ignoring non-beneficial shaping reward.** Switching control gives our method the power to learn when to attend to shaping rewards and when to ignore them. This allows us to learn to ignore "red-herrings", i.e., unexplored parts of the state space where there is no real environment reward, but where surprise or novelty metrics would place high shaping reward. To verify this claim, we use a modified Maze environment called Red-Herring Maze which features a large part of the state space that has no environment reward, but with the goal (and accompanying real reward) in a different part

---

[4]BiPaRS-IMGL requires a manually crafted shaping-reward (only available in Cartpole).

[5]The sum of curves for each method may be less that 1 if the agent fails to arrive at either goal.

of the state space. Ideally, we expect that the reward shaping method can learn to quickly ignore the large part of the state space. Fig. 3 shows that our method outperforms all other baselines. Moreover, the heatmap shows that while RND is easily dragged to reward exploring novel but non rewarding states our method learns to ignore them.

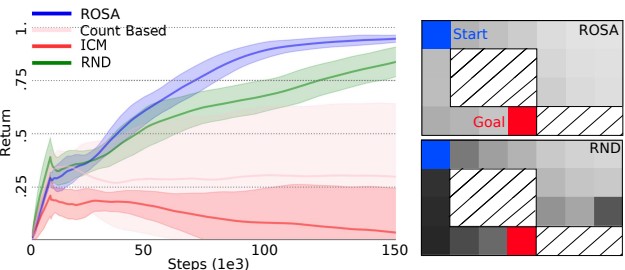

Figure 3: Red-Herring Maze. Ignoring non-beneficial shaping reward. *Left.* Learning curves. *Right.* Heatmap of added shaping rewards. ROSA ignores the RHS of the maze, while RND incorrectly adds unuseful shaping rewards there.

**Learning Performance.** We compared our method with the baselines in four challenging sparse rewards environments: Cartpole, Gravitar, Solaris, and Super Mario. These environments vary in state representation, transition dynamics and reward sparsity. In Cartpole, a penalty of $-1$ is received only when the pole collapses; in Super Mario Brothers the agent can go for 100s of steps without encountering a reward. Fig. 4 shows learning curves. In terms of performance, ROSA either markedly outperforms the best competing baseline (Cartpole, Gravitar) or is on par with them (Solaris, Super Mario) showing that it is robust to the nature of the environment and underlying sparse reward. Moreover, ROSA does not exhibit the failure modes where after good initial performance it deteriorates. E.g., in Solaris both ICM and RND have good initial performance but deteriorate sharply while ROSA's performance remains satisfactory.

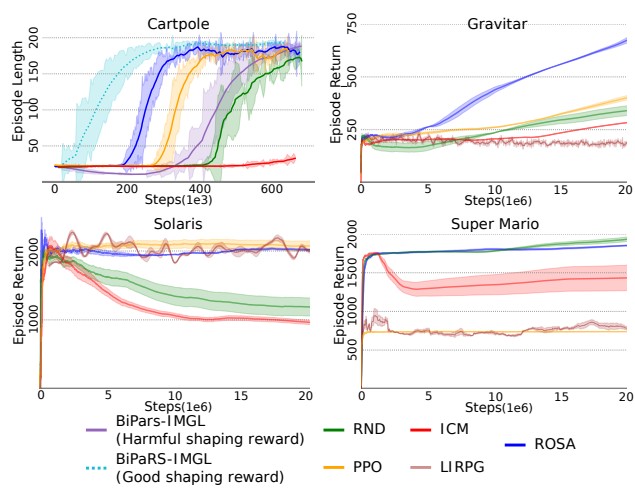

Figure 4: Benchmark performance.

## 7 CONCLUSION

In this paper, we presented a novel solution method to solve the problem of reward shaping. Our Markov game framework of a primary Controller and a secondary reward shaping agent is guaranteed to preserve the underlying learning task for Controller whilst guiding Controller to higher performance policies. Moreover, our method is able to decompose complex learning tasks into subgoals and to adaptively guide Controller by selectively choosing the states to add shaping rewards. By presenting a theoretically sound and empirically robust approach to solving the reward shaping problem, our method opens up the applicability of RL to a range of real-world control problems. The most significant contribution of this paper, however, is the novel construction that marries RL, multi-agent RL and game theory which leads to new solution method in RL. We believe this powerful approach can be adopted to solve other open challenges in RL.

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

# Appendix

# Table of Contents

## 8 ALGORITHM

---

**Algorithm 2: R**einforcement Learning **O**ptimising **S**haping **A**lgorithm ROSA

---

**Input:** Environment $E$

       Initial Controller policy $\pi_0$ with parameters $\theta_{\pi_0}$

       Initial Shaper switch policy $\mathfrak{g}_{2_0}$ with parameters $\theta_{\mathfrak{g}_{2_0}}$

       Initial Shaper action policy $\pi_0^2$ with parameters $\theta_{\pi_0^2}$

       Randomly initialised fixed neural network $\phi(\cdot, \cdot)$

       Neural networks $h$ (fixed) and $\hat{h}$ for RND with parameter $\theta_{\hat{h}}$

       Buffer $B$

       Number of rollouts $N_r$, rollout length $T$

       Number of mini-batch updates $N_u$

       Switch cost $c(\cdot)$, Discount factor $\gamma$, learning rate $\alpha$

**Output:** Optimised Controller policy $\pi^*$

1   $\pi, \pi^2, \mathfrak{g}_2 \leftarrow \pi_0, \pi_0^2, \mathfrak{g}_{2_0}$

2   **for** $n = 1, N_r$ **do**

3      **// Collect rollouts**

4      **for** $t = 1, T$ **do**

5          Get environment states $s_t$ from $E$

6          Sample $a_t$ from $\pi(s_t)$

7          Apply action $a_t$ to environment $E$, and get reward $r_t$ and next state $s_{t+1}$

8          Sample $g_t$ from $\mathfrak{g}_2(s_t)$ **// Switching control**

9          **if** $g_t = 1$ **then**

10             Sample $a_t^2$ from $\pi^2(s_t)$

11             Sample $a_{t+1}^2$ from $\pi^2(s_{t+1})$

12             $r_t^i = \gamma \phi(s_{t+1}, a_{t+1}^2) - \phi(s_t, a_t^2)$ **// Calculate** $F(s_t, a_t, s_{t+1}, a_{t+1})$

13          **else**

14             $a_t^2, r_t^i = 0, 0$ **// Dummy values**

15          Append $(s_t, a_t, g_t, a_t^2, r_t, r_t^i, s_{t+1})$ to $B$

16      **for** $u = 1, N_u$ **do**

17          Sample data $(s_t, a_t, g_t, a_t^2, r_t, r_t^i, s_{t+1})$ from $B$

18          **if** $g_t = 1$ **then**

19             Set shaped reward to $r_t^s = r_t + r_t^i$

20          **else**

21             Set shaped reward to $r_t^s = r_t$

22          **// Update RND**

23          $\text{Loss}_{\text{RND}} = ||h(s_t) - \hat{h}(s_t)||^2$

24          $\theta_{\hat{h}} \leftarrow \theta_{\hat{h}} - \alpha \nabla \text{Loss}_{\text{RND}}$

25          **// Update Shaper**

26          $l_t = ||h(s_t) - \hat{h}(s_t)||^2$ **// Compute** $L(s_t)$

27          $c_t = c(\cdot) g_t$

28          Compute $\text{Loss}_{\pi^2}$ using $(s_t, a_t, g_t, c_t, r_t, r_t^i, l_t, s_{t+1})$ using PPO loss **// Section 4.2**

29          Compute $\text{Loss}_{\mathfrak{g}_2}$ using $(s_t, a_t, g_t, c_t, r_t, r_t^i, l_t, s_{t+1})$ using PPO loss **// Section 4.2**

30          $\theta_{\pi^2} \leftarrow \theta_{\pi^2} - \alpha \nabla \text{Loss}_{\pi^2}$

31          $\theta_{\mathfrak{g}_2} \leftarrow \theta_{\mathfrak{g}_2} - \alpha \nabla \text{Loss}_{\mathfrak{g}_2}$

32          **// Update Controller**

33          Compute $\text{Loss}_\pi$ using $(s_t, a_t, r_t^s, s_{t+1})$ using PPO loss **// Section 4**

34          $\theta_\pi \leftarrow \theta_\pi - \alpha \nabla \text{Loss}_\pi$

---

## 9 FURTHER IMPLEMENTATION DETAILS

Details of Shaper and $F$ (shaping reward)

| Object | Description |
|---|---|
| $f$ | Fixed feed forward NN that maps $\mathbb{R}^d \mapsto \mathbb{R}^m$ [512, `ReLU`, 512, `ReLU`, 512, $m$] |
| $\mathcal{A}_2$ | Discrete integer action set which is size of output of $f$, i.e., $\mathcal{A}_2$ is set of integers $\{1, ..., m\}$ |
| $\pi_2$ | Fixed feed forward NN that maps $\mathbb{R}^d \mapsto \mathbb{R}^m$ [512, `ReLU`, 512, `ReLU`, 512, $m$] |
| Potential function $\phi$ | $\phi(s, a^2) = f(s) \cdot a^2$ |
| $F$ | $\gamma\phi(s_{t+1}, a^2_{t+1})$ - $\phi(s_t, a^2_t)$, $\gamma = 0.95$ |

$d$=Dimensionality of states; $m \in \mathbb{N}$ - tunable free parameter.

In all experiments we used the above form of $F$ as follows: a state $s_t$ is input to the $\pi_2$ network and the network outputs logits $p_t$. We softmax and sample from $p_t$ to obtain the action $a^2_t$. This action is one-hot encoded. Then, the action $a^2_t$ is multiplied with $f(s_t)$ to compute the second term of $F$. A similar process is used to compute the first term. In this way the policy of Shaper chooses the shaping reward.

## 10 SHAPER TERMINATION TIMES

There are various possibilities for the *termination* times $\{\tau_{2k}\}$ (recall that $\{\tau_{2k+1}\}$ are the times which the shaping reward $F$ is *switched on* using $\mathfrak{g}_2$). One is for Shaper to determine the sequence. Another is to build a construction of $\{\tau_{2k}\}$ that directly incorporates the information gain that a state visit provides: let $w : \Omega \to \{0, 1\}$ be a random variable with $\Pr(w = 1) = p$ and $\Pr(w = 0) = 1-p$ where $p \in ]0, 1]$. Then for any $k = 1, 2, \ldots$, and denote by $\Delta L(s_{\tau_k}) := L(s_{\tau_k}) - L(s_{\tau_k - 1})$, then we can set:

$$I(s_{\tau_{2k+1}+j}) = \begin{cases} I(s_{\tau_{2k+1}}), & \text{if } w\Delta L(s_{\tau_k+j}) > 0, \\ I(s_{\tau_{2k+2}}), & w\Delta L(s_{\tau_k+j}) \leq 0. \end{cases} \tag{2}$$

To explain, since $\{\tau_{2k}\}_{k \geq 0}$ are the times at which $F$ is switched off then if $F$ is deactivated at exactly after $j$ time steps then $I(s_{\tau_{2k+1}+l}) = I(s_{\tau_{2k+1}})$ for any $0 \leq l < j$ and $I(s_{\tau_{2k+1}+j}) = I(s_{\tau_{2k+2}})$. We now see that (2) terminates $F$ when either the random variable $w$ attains a 0 or when $\Delta L(s_{\tau_k+j}) \leq 0$ which occurs when the exploration bonus in the current state is lower than that of the previous state.

## 11 EXPERIMENTAL DETAILS

### 11.1 ENVIRONMENTS & PREPROCESSING DETAILS

The table below shows the provenance of environments used in our experiments.

| | |
|---|---|
| Atari & Cartpole | https://github.com/openai/gym |
| Maze | https://github.com/MattChanTK/gym-maze |
| Super Mario Brothers | https://github.com/Kautenja/gym-super-mario-bros |

Furthermore, we used preprocessing settings as indicated in the following table.

| Setting | Value |
|---|---|
| Max frames per episode | Atari & Mario $\rightarrow$ 18000 / Maze & Cartpole $\rightarrow$ 200 |
| Observation concatenation | Preceding 4 observations |
| Observation preprocessing | Standardization followed by clipping to [-5, 5] |
| Observation scaling | Atari & Mario $\rightarrow$ (84, 84, 1) / Maze & Cartpole $\rightarrow$ None |
| Reward (extrinsic and intrinsic) preprocessing | Standardization followed by clipping to [-1, 1] |

### 11.2 HYPERPARAMETER SETTINGS

In the table below we report all hyperparameters used in our experiments. Hyperparameter values in square brackets indicate ranges of values that were used for performance tuning.

| | |
|---|---|
| Clip Gradient Norm | 1 |
| $\gamma_E$ | 0.99 |
| $\lambda$ | 0.95 |
| Learning rate | $1\times10^{-4}$ |
| Number of minibatches | 4 |
| Number of optimization epochs | 4 |
| Policy architecture | CNN (Mario/Atari) or MLP (Cartpole/Maze) |
| Number of parallel actors | 2 (Cartpole/Maze) or 20 (Mario/Atari) |
| Optimization algorithm | Adam |
| Rollout length | 128 |
| Sticky action probability | 0.25 |
| Use Generalized Advantage Estimation | True |
| Coefficient of extrinsic reward | [1, 5] |
| Coefficient of intrinsic reward | [1, 2, 5, 10, 20, 50] |
| $\gamma_I$ | 0.99 |
| Probability of terminating option | [0.5, 0.75, 0.8, 0.9, 0.95] |
| RND output size | [2, 4, 8, 16, 32, 64, 128, 256] |

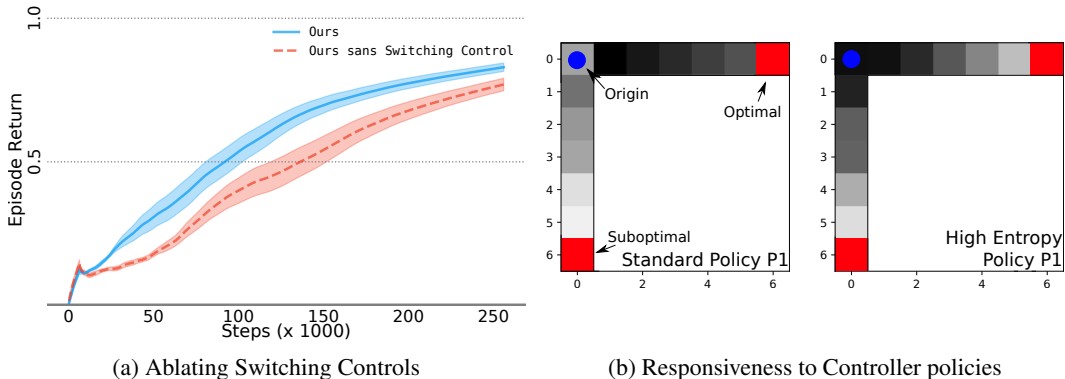

(a) Ablating Switching Controls        (b) Responsiveness to Controller policies

Figure 5: Ablation Experiments

## 12 ABLATION STUDIES

Our reward-shaping method features a mechanism to selectively pick states to which intrinsic rewards are added. It also adapts its shaping rewards according to Controller's learning process. In this section, we present the results of experiments in which we ablated each of these components. In particular, we test the performance of our method in comparison to a version of our method with the switching mechanism removed. We then present the result of an experiment in which we investigated the ability of our method to adapt to different behaviour of Controller.

### ABLATION STUDY 1: SWITCHING CONTROLS

Switching controls enable our method to be selective of states to which intrinsic rewards are added. This improves learnability (specifically, by reducing the computational complexity) of the learning task for Shaper as there are fewer states where it must learn the optimal intrinsic reward to add to Controller objective.

To test the effect of this feature on the performance of our method, we compared our method to a modified version in which Shaper must add intrinsic rewards to all states. That is, for this version of our method we remove the presence of the switching control mechanism for Shaper. Figure 5 (a) shows learning curves on the Maze environment used in the "Optimality of shaping reward" experiments in Section 6. As expected, the agent with the version of our method with switching controls learns significantly faster than the agent that uses the version of our method sans the switching control mechanism. For example, it takes the agent that has no switching control mechanism almost 50,000 more steps to attain an average episode return of 0.5 as compared against the agent that uses the version of our algorithm with switching controls.

This illustrates a key benefit of switching controls which is to reduce the computational burden on Shaper (as it does not need to model the effects of adding intrinsic rewards in *all* states) which in turn leads to both faster computation of solutions and improved performance by Controller. Moreover, Maze is a relatively simple environment, expectedly the importance of the switching control is amplified in more complex environments.

### ABLATION STUDY 2: ADAPTION OF OUR METHOD TO DIFFERENT CONTROLLER POLICIES

We claimed Shaper can design a reward-shaping scheme that can *adapt* its shaping reward guidance of Controller (to achieve the optimal policy) according to Controller's (RL) policy.

To test this claim, we tested two versions of our agent in a corridor Maze. The maze features two goal states that are equidistant from the origin, one is a suboptimal goal with a reward of $0.5$ and the other is an optimal goal which has a reward $1$. There is also a fixed cost for each non-terminal transition. We tested this scenario with two versions of our controller: one with a standard RL Controller policy and another version in which the actions of Controller are determined by a high

entropy policy, we call this version of Controller the *high entropy controller*.[6] The high entropy policy induces actions that may randomly push Controller towards the suboptimal goal. Therefore, in order to guide Controller to the optimal goal state, we expect Shaper to strongly shape the rewards of Controller to guide Controller away from the suboptimal goal (and towards the optimal goal).

Figure 5 (b) shows heatmaps of the added intrinsic reward (darker colours indicate higher intrinsic rewards) for the two versions of Controller. With the standard policy controller, the intrinsic reward is maximal in the state to the right of the origin indicating that Shaper determines that these shaping rewards are sufficient to guide Controller towards the optimal goal state. For the high entropy controller, Shaper introduces high intrinsic rewards to the origin state as well as states beneath the origin. These rewards serve to counteract the random actions taken by the high-entropy policy that lead Controller towards the suboptimal goal state. It can therefore be seen that Shaper adapts the shaping rewards according to the type of Controller it seeks to guide.

---

[6]To generate this policy, we artificially increased the entropy by adjusting the temperature of a softmax function on the policy logits.

## 13  FLEXIBILITY OF ROSA TO ACCOMMODATE DIFFERENT EXPLORATION BONUS TERMS $L$

To demonstrate the robustness of our method to different choices of exploration bonus terms in Shaper's objective, we conducted an Ablation study on the $L$-term (c.f. Equation 1) where we replaced the RND $L$ term with a basic count-based exploration bonus. To exemplify the high degree of flexibility, we replaced the RND with a simple exploration bonus term $L(s) = \frac{1}{\text{Count}(s)+1}$ for any given state $s \in \mathcal{S}$ where $\text{Count}(s)$ refers to a simple count of the number of times the state $s$ has been visited. We conducted the Ablation study in the environment in Experiment 1 presented in Sec. 6. We note that despite the simplicity of the count-based measure, generally the performance of both versions of ROSA is comparable and in fact the count-based variant is slightly superior to the RND version.

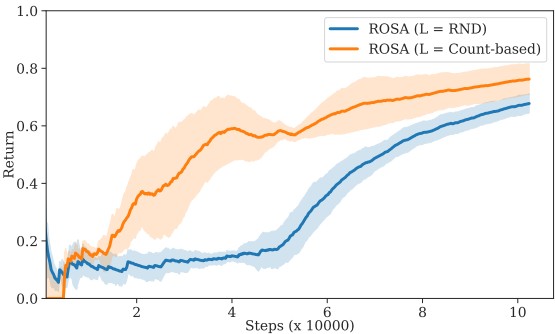

Figure 6: Performance of ROSA compared with the exploration bonus replaced by count-based method.

## 14 ROBUSTNESS TO CHOICES OF $\phi$ PARAMETERS

To demonstrate the robustness of $\phi$ to different choices of weight parameters, we conducted a study with 3 sets of randomly sampled values of weight parameters for the feed forward NN that constructs the $\phi$ function (c.f. Sec. 9).

As is shown in Fig. 7, the performance across all 3 values is very comparable demonstrating the robustness of ROSA to different values of weight parameters for the $\phi$ function.

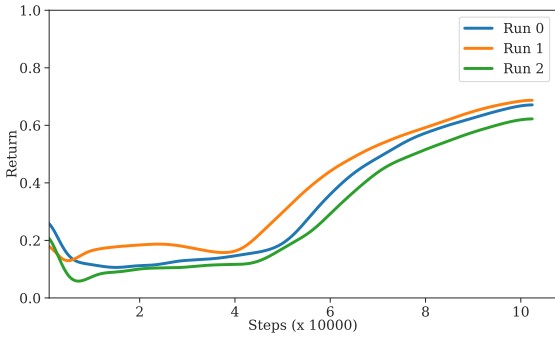

Figure 7: Performance of ROSA compared with different values of $\phi$ parameters.

## 15   NOTATION & ASSUMPTIONS

We assume that $\mathcal{S}$ is defined on a probability space $(\Omega, \mathcal{F}, \mathbb{P})$ and any $s \in \mathcal{S}$ is measurable with respect to the Borel $\sigma$-algebra associated with $\mathbb{R}^p$. We denote the $\sigma$-algebra of events generated by $\{s_t\}_{t \geq 0}$ by $\mathcal{F}_t \subset \mathcal{F}$. In what follows, we denote by $(\mathcal{V}, \|\|)$ any finite normed vector space and by $\mathcal{H}$ the set of all measurable functions.

The results of the paper are built under the following assumptions which are standard within RL and stochastic approximation methods:

**Assumption 1** The stochastic process governing the system dynamics is ergodic, that is the process is stationary and every invariant random variable of $\{s_t\}_{t \geq 0}$ is equal to a constant with probability 1.

**Assumption 2** The constituent functions of the players' objectives $R$, $F$ and $L$ are in $L_2$.

**Assumption 3** For any positive scalar $c$, there exists a scalar $\mu_c$ such that for all $s \in \mathcal{S}$ and for any $t \in \mathbb{N}$ we have: $\mathbb{E}\left[1 + \|s_t\|^c | s_0 = s\right] \leq \mu_c(1 + \|s\|^c)$.

**Assumption 4** There exists scalars $C_1$ and $c_1$ such that for any function $J$ satisfying $|J(s)| \leq C_2(1 + \|s\|^{c_2})$ for some scalars $c_2$ and $C_2$ we have that: $\sum_{t=0}^{\infty} |\mathbb{E}\left[J(s_t)|s_0 = s\right] - \mathbb{E}[J(s_0)]| \leq C_1 C_2(1 + \|s_t\|^{c_1 c_2})$.

**Assumption 5** There exists scalars $c$ and $C$ such that for any $s \in \mathcal{S}$ we have that: $|J(z, \cdot)| \leq C(1 + \|z\|^c)$ for $J \in \{R, F, L\}$.

We also make the following finiteness assumption on set of switching control policies for Shaper:

**Assumption 6** For any policy $\mathfrak{g}_c$, the total number of interventions is given by $K < \infty$.

We lastly make the following assumption on $L$ which can be made true by construction:

**Assumption 7** Let $n(s)$ be the state visitation count for a given state $s \in \mathcal{S}$. For any $a \in \mathcal{A}$, the function $L(s, a) = 0$ for any $n(s) \geq M$ where $0 < M \leq \infty$.

## 16   PROOF OF TECHNICAL RESULTS

We begin the analysis with some preliminary lemmata and definitions which are useful for proving the main results.

**Definition 1** *A.1 An operator $T : \mathcal{V} \to \mathcal{V}$ is said to be a **contraction** w.r.t a norm $\| \cdot \|$ if there exists a constant $c \in [0, 1[$ such that for any $V_1, V_2 \in \mathcal{V}$ we have that:*

$$\|TV_1 - TV_2\| \leq c\|V_1 - V_2\|. \tag{3}$$

**Definition 2** *A.2 An operator $T : \mathcal{V} \to \mathcal{V}$ is **non-expansive** if $\forall V_1, V_2 \in \mathcal{V}$ we have:*

$$\|TV_1 - TV_2\| \leq \|V_1 - V_2\|. \tag{4}$$

**Lemma 1** *For any $f : \mathcal{V} \to \mathbb{R}, g : \mathcal{V} \to \mathbb{R}$, we have that:*

$$\left\| \max_{a \in \mathcal{V}} f(a) - \max_{a \in \mathcal{V}} g(a) \right\| \leq \max_{a \in \mathcal{V}} \|f(a) - g(a)\|. \tag{5}$$

**Proof 1** *We restate the proof given in [23]:*

$$f(a) \leq \|f(a) - g(a)\| + g(a) \tag{6}$$

$$\implies \max_{a \in \mathcal{V}} f(a) \leq \max_{a \in \mathcal{V}}\{\|f(a) - g(a)\| + g(a)\} \leq \max_{a \in \mathcal{V}} \|f(a) - g(a)\| + \max_{a \in \mathcal{V}} g(a). \tag{7}$$

*Deducting $\max_{a \in \mathcal{V}} g(a)$ from both sides of (7) yields:*

$$\max_{a \in \mathcal{V}} f(a) - \max_{a \in \mathcal{V}} g(a) \leq \max_{a \in \mathcal{V}} \|f(a) - g(a)\|. \tag{8}$$

*After reversing the roles of $f$ and $g$ and redoing steps (6) - (7), we deduce the desired result since the RHS of (8) is unchanged.*

**Lemma 2** *A.4 The probability transition kernel $P$ is non-expansive, that is:*

$$\|PV_1 - PV_2\| \le \|V_1 - V_2\|. \tag{9}$$

**Proof 2** *The result is well-known e.g. [40]. We give a proof using the Tonelli-Fubini theorem and the iterated law of expectations, we have that:*

$$\|PJ\|^2 = \mathbb{E}\left[(PJ)^2[s_0]\right] = \mathbb{E}\left(\left[\mathbb{E}\left[J[s_1]|s_0\right]\right)^2\right] \le \mathbb{E}\left[\mathbb{E}\left[J^2[s_1]|s_0\right]\right] = \mathbb{E}\left[J^2[s_1]\right] = \|J\|^2,$$

*where we have used Jensen's inequality to generate the inequality. This completes the proof.*

PROOF OF PROPOSITION 1

**Proof 3 (Proof of Prop 1)** *To prove (i) of the proposition it suffices to prove that the term $\sum_{t=0}^{T} \gamma^t F(\theta_t, \theta_{t-1}) I(t)$ converges to 0 in the limit as $T \to \infty$. As in classic potential-based reward shaping [27], central to this observation is the telescoping sum that emerges by construction of $F$.*

*First recall $v_1^{\pi,\pi^2}(s, I_0)$, for any $(s, I_0) \in \mathcal{S} \times \{0, 1\}$ is given by:*

$$v_1^{\pi,\pi^2}(s, I_0) = \mathbb{E}_{\pi,\pi^2}\left[\sum_{t=0}^{\infty} \gamma^t \left\{ R_i(s_t, a_t) + \hat{F}(s_t, a_t^2; s_{t-1}, a_{t-1}^2) I_t \right\}\right] \tag{10}$$

$$= \mathbb{E}_{\pi,\pi^2}\left[\sum_{t=0}^{\infty} \gamma^t R_i(s_t, a_t) + \sum_{t=0}^{\infty} \gamma^t \hat{F}(s_t, a_t^2; s_{t-1}, a_{t-1}^2) I_t\right] \tag{11}$$

$$= \mathbb{E}_{\pi,\pi^2}\left[\sum_{t=0}^{\infty} \gamma^t R_i(s_t, a_t)\right] + \mathbb{E}_{\pi,\pi^2}\left[\sum_{t=0}^{\infty} \gamma^t \hat{F}(s_t, a_t^2; s_{t-1}, a_{t-1}^2)) I_t\right]. \tag{12}$$

*Hence it suffices to prove that $\mathbb{E}_{\pi,\pi^2}\left[\sum_{t=0}^{\infty} \gamma^t \hat{F}(s_t, a_t^2; s_{t-1}, a_{t-1}^2)) I_t\right] = 0.$*

*Recall there a number of time steps that elapse between $\tau_k$ and $\tau_{k+1}$, now*

$$\sum_{t=0}^{\infty} \gamma^t \hat{F}(s_t, a_t^2; s_{t-1}, a_{t-1}^2)) I(t)$$

$$= \sum_{t=\tau_1+1}^{\tau_2} \gamma^t \phi(s_t, a_t^2) - \gamma^{t-1}\phi(s_{t-1}, a_{t-1}^2) + \gamma^{\tau_1}\phi(s_{\tau_1}, a_{\tau_1}^2)$$

$$+ \sum_{t=\tau_3+1}^{\tau_4} \gamma^t \phi(s_t, a_t^2) - \gamma^{t-1}\phi(s_{t-1}, a_{t-1}^2) + \gamma^{\tau_3}\phi(s_{\tau_3}, a_{\tau_3}^2)$$

$$+ \ldots + \sum_{t=\tau_{(2k-1)}+1}^{\tau_{2k}} \gamma^t \phi(s_t, a_t^2) - \gamma^{t-1}\phi(s_{t-1}, a_{t-1}^2) + \gamma^{\tau_1}\phi(s_{\tau_{2k+1}}, a_{\tau_{2k+1}}^2) + \ldots +$$

$$= \sum_{t=\tau_1}^{\tau_2-1} \gamma^{t+1}\phi(s_{t+1}, a_{t+1}^2) - \gamma^t \phi(s_t, a_t^2) + \gamma^{\tau_1}\phi(s_{\tau_1}, a_{\tau_1}^2)$$

$$+ \sum_{t=\tau_3}^{\tau_4-1} \gamma^{t+1}\phi(s_{t+1}, a_{t+1}^2) - \gamma^t \phi(s_t, a_t^2) + \gamma^{\tau_3}\phi(s_{\tau_3}, a_{\tau_3}^2)$$

$$+ \ldots + \sum_{t=\tau_{(2k-1)}}^{\tau_{2K}-1} \gamma^t \phi(s_t, a_t^2) - \gamma^{t-1}\phi(s_{t-1}, a_{t-1}^2) + \gamma^{\tau_{2k-1}}\phi(s_{\tau_{2k-1}}, a_{\tau_{2k-1}}^2) + \ldots +$$

$$= \sum_{k=1}^{\infty} \sum_{t=\tau_{2k-1}}^{\tau_{2K}-1} \gamma^{t+1}\phi(s_{t+1}, a_{t+1}^2) - \gamma^t \phi(s_t, a_t^2) - \sum_{k=1}^{\infty} \gamma^{\tau_{2k-1}}\phi(s_{\tau_{2k-1}}, a_{\tau_{2k-1}}^2)$$

$$= \sum_{k=1}^{\infty} \gamma^{\tau_{2k}} \phi(s_{\tau_{2k}}, a_{\tau_{2k}}^2) - \sum_{k=1}^{\infty} \gamma^{\tau_{2k-1}} \phi(s_{\tau_{2k-1}}, a_{\tau_{2k-1}}^2)$$

$$= \sum_{k=1}^{\infty} \gamma^{\tau_{2k}} \phi(s_{\tau_{2k}}, 0) - \sum_{k=1}^{\infty} \gamma^{\tau_{2k-1}} \phi(s_{\tau_{2k-1}}, 0) = 0,$$

*where we have used the fact that by construction $a_t^2 \equiv 0$ whenever $t = \tau_1, \tau_2, \ldots$ and by construction $\phi(s, 0) \equiv 0$ for any $s$.*

*With this we readily deduce that $v_i^{\pi, \pi^2}(s) = \mathbb{E}_{\pi, \pi^2} \left[ \sum_{t=0}^{\infty} \gamma^t R_i(s_t, a_t) \right]$ which is a measure of environment rewards only from which statement (i) can be readily deduced.*

*For part (ii) we note first that it is easy to see that $v_2^{\pi, \pi^2}(s_0, I_0)$ is bounded above, indeed using the key result in the proof of part (i) and the properties of c we have that*

$$v_2^{\pi, \pi^2}(s_0, I_0) = \mathbb{E}_{\pi, \pi^2} \left[ \sum_{t=0}^{\infty} \gamma^t \left( \hat{R} + \sum_{k \geq 1} c(I_t, I_{t-1}) \delta_{\tau_{2k-1}}^t + L_n(s_t) \right) \right] \tag{13}$$

$$= \mathbb{E}_{\pi, \pi^2} \left[ \sum_{t=0}^{\infty} \gamma^t \left( R + \sum_{k \geq 1} c(I_t, I_{t-1}) \delta_{\tau_{2k-1}}^t + L_n(s_t) \right) + \sum_{t=0}^{\infty} \gamma^t \hat{F} I_t \right] \tag{14}$$

$$\leq \mathbb{E}_{\pi, \pi^2} \left[ \sum_{t=0}^{\infty} \gamma^t \left( R + L_n(s_t) \right) \right] \tag{15}$$

$$\leq \left| \mathbb{E}_{\pi, \pi^2} \left[ \sum_{t=0}^{\infty} \gamma^t \left( R + L_n(s_t) \right) \right] \right| \tag{16}$$

$$\leq \mathbb{E}_{\pi, \pi^2} \left[ \sum_{t=0}^{\infty} \gamma^t \| R + L_n \| \right] \tag{17}$$

$$\leq \sum_{t=0}^{\infty} \gamma^t \left( \| R \| + \| L_n \| \right) \tag{18}$$

$$= \frac{1}{1 - \gamma} \left( \| R \| + \| L \| \right), \tag{19}$$

*using the triangle inequality, the definition of $\hat{R}$ and the (upper-)boundedness of $L$ and $R$ (Assumption 5). We now note that by the dominated convergence theorem we have that $\forall (s_0, I_0) \in \mathcal{S} \times \{0, 1\}$*

$$\lim_{n \to \infty} v_2^{\pi, \pi^2}(s_0, I_0) = \lim_{n \to \infty} \mathbb{E}_{\pi, \pi^2} \left[ \sum_{t=0}^{\infty} \gamma^t \left( \hat{R} + \sum_{k \geq 1} c(I_t, I_{t-1}) \delta_{\tau_{2k-1}}^t + L_n(s_t) \right) \right] \tag{20}$$

$$= \mathbb{E}_{\pi, \pi^2} \lim_{n \to \infty} \left[ \sum_{t=0}^{\infty} \gamma^t \left( \hat{R} + \sum_{k \geq 1} c(I_t, I_{t-1}) \delta_{\tau_{2k-1}}^t + L_n(s_t) \right) \right] \tag{21}$$

$$= \mathbb{E}_{\pi, \pi^2} \left[ \sum_{t=0}^{\infty} \gamma^t \left( \hat{R} + \sum_{k \geq 1} c(I_t, I_{t-1}) \delta_{\tau_{2k-1}}^t \right) \right] \tag{22}$$

$$= \mathbb{E}_{\pi, \pi^2} \left[ \sum_{t=0}^{\infty} \gamma^t \left( R + \sum_{k \geq 1} c(I_t, I_{t-1}) \delta_{\tau_{2k-1}}^t \right) \right] = \frac{K}{1 - \gamma} + v_1^{\pi}(s_0), \tag{23}$$

*again using the key result in the proof of (i) and Assumption 6 in the last step, after which we deduce (ii).*

*Note that by (ii) we heron may consider the quantity for the Shaper expected return:*

$$\hat{v}_2^{\pi,\pi^2}(s_0, I_0) = \mathbb{E}_{\pi,\pi^2}\left[\sum_{t=0}^{\infty}\gamma^t\left(R + \sum_{k\geq 1}c(I_t, I_{t-1})\delta_{\tau_{2k-1}}^t\right)\right]. \qquad (24)$$

## PROOF OF THEOREM 1

**Proof 4** *Theorem 1 is proved by firstly showing that when the players jointly maximise the same objective there exists a fixed point equilibrium of the game when all players use Markov policies and Shaper uses switching control. The proof then proceeds by showing that the MG $\mathcal{G}$ admits a dual representation as an MG in which jointly maximise the same objective which has a stable point that can be computed by solving an MDP. Thereafter, we use both results to prove the existence of a fixed point for the game as a limit point of a sequence generated by successively applying the Bellman operator to a test function.*

*Therefore, the scheme of the proof is summarised with the following steps:*

**I)** *Prove that the solution to Markov Team games (that is games in which both players maximise identical objectives) in which one of the players uses switching control is the limit point of a sequence of Bellman operators (acting on some test function).*

**II)** *Prove that for the MG $\mathcal{G}$ that is there exists a function $B^{\pi,\pi^2} : \mathcal{S}\times\{0,1\}\to\mathbb{R}$ such that[7] $v_i^{\pi,\pi^2}(z) - v_i^{\pi',\pi^2}(z) = B^{\pi,\pi^2}(z) - B^{\pi',\pi^2}(z), \;\; \forall z \equiv (s, I_0)\in\mathcal{S}\times\{0,1\}, \forall i\in\{1,2\}.$*

**III)** *Prove that the MG $\mathcal{G}$ has a dual representation as a Markov Team Game which admits a representation as an MDP.*

## PROOF OF PART **I**

*Our first result proves that the operator $T$ is a contraction operator. First let us recall that the switching time $\tau_k$ is defined recursively $\tau_k = \inf\{t > \tau_{k-1}|s_t\in A, \tau_k\in\mathcal{F}_t\}$ where $A = \{s\in\mathcal{S}, m\in M|\mathfrak{g}_2(m|s_t) > 0\}$. To this end, we show that the following bounds holds:*

**Lemma 3** *The Bellman operator $T$ is a contraction, that is the following bound holds:*

$$\|T\psi - T\psi'\| \leq \gamma\|\psi - \psi'\|.$$

**Proof 5** *Recall we define the Bellman operator $T_\psi$ of $\mathcal{G}$ acting on a function $\Lambda : \mathcal{S}\times\mathbb{N}\to\mathbb{R}$ by*

$$T_\psi\Lambda(s_{\tau_k}, I(\tau_k)) := \max\left\{\mathcal{M}^{\pi,\pi^2}\Lambda(s_{\tau_k}, I(\tau_k)), \left[\psi(s_{\tau_k}, a) + \gamma\max_{a\in\mathcal{A}}\sum_{s'\in\mathcal{S}}P(s'; a, s_{\tau_k})\Lambda(s', I(\tau_k))\right]\right\} \qquad (25)$$

*In what follows and for the remainder of the script, we employ the following shorthands:*

$$\mathcal{P}_{ss'}^a =: \sum_{s'\in\mathcal{S}}P(s'; a, s), \quad \mathcal{P}_{ss'}^\pi =: \sum_{a\in\mathcal{A}}\pi(a|s)\mathcal{P}_{ss'}^a, \quad \mathcal{R}^\pi(z_t) := \sum_{a_t\in\mathcal{A}}\pi(a_t|s)\hat{R}(z_t, a_t, \theta_t, \theta_{t-1})$$

*To prove that $T$ is a contraction, we consider the three cases produced by (25), that is to say we prove the following statements:*

i) $\left|\Theta(z_t, a, a_t^2, a_{t-1}^2) + \gamma\max_{a\in\mathcal{A}}\mathcal{P}_{s's_t}^a\psi(s', \cdot) - \left(\Theta(z_t, a, a_t^2, a_{t-1}^2) + \gamma\max_{a\in\mathcal{A}}\mathcal{P}_{s's_t}^a\psi'(s', \cdot)\right)\right| \leq \gamma\|\psi - \psi'\|$

ii) $\left\|\mathcal{M}^{\pi,\pi^2}\psi - \mathcal{M}^{\pi,\pi^2}\psi'\right\| \leq \gamma\|\psi - \psi'\|,$      *(and hence $\mathcal{M}$ is a contraction).*

---

[7]This property is analogous to the condition in Markov potential games [19, 26]

*iii)*
$$\left\| \mathcal{M}^{\pi,\pi^2}\psi - \left[\Theta(\cdot,a) + \gamma\max_{a\in\mathcal{A}} \mathcal{P}^a\psi'\right]\right\| \leq \gamma \left\|\psi - \psi'\right\|. \text{ where } z_t \equiv (s_t, I_t) \in \mathcal{S} \times$$
$\{0,1\}$.

*We begin by proving i).*

*Indeed, for any $a \in \mathcal{A}$ and $\forall z_t \in \mathcal{S} \times \{0,1\}, \forall \theta_t, \theta_{t-1} \in \Theta, \forall s' \in \mathcal{S}$ we have that*

$$\left| \Theta(z_t, a, a_t^2, a_{t-1}^2) + \gamma\mathcal{P}^\pi_{s's_t}\psi(s', \cdot) - \left[\Theta(z_t, a, a_t^2, a_{t-1}^2) + \gamma\max_{a\in\mathcal{A}} \mathcal{P}^a_{s's_t}\psi'(s', \cdot)\right]\right|$$
$$\leq \max_{a\in\mathcal{A}} \left| \gamma\mathcal{P}^a_{s's_t}\psi(s', \cdot) - \gamma\mathcal{P}^a_{s's_t}\psi'(s', \cdot)\right|$$
$$\leq \gamma \left\|P\psi - P\psi'\right\|$$
$$\leq \gamma \left\|\psi - \psi'\right\|,$$

*again using the fact that $P$ is non-expansive and Lemma 1.*

*We now prove ii).*

*For any $\tau \in \mathcal{F}$, define by $\tau' = \inf\{t > \tau | s_t \in A, \tau \in \mathcal{F}_t\}$. Now using the definition of $\mathcal{M}$ we have that for any $s_\tau \in \mathcal{S}$*

$$\left|(\mathcal{M}^{\pi,\pi^2}\psi - \mathcal{M}^{\pi,\pi^2}\psi')(s_\tau, I(\tau))\right|$$
$$\leq \max_{a_\tau, a_\tau^2, a_{\tau-1}^2 \in \mathcal{A}\times\Theta^2}\left| \Theta(z_\tau, a_\tau, a_\tau^2, a_{\tau-1}^2) + c(I_\tau, I_{\tau-1}) + \gamma\mathcal{P}^\pi_{s's_\tau}\mathcal{P}^a\psi(s_\tau, I(\tau'))\right.$$
$$\left. - \left(\Theta(z_\tau, a_\tau, a_\tau^2, a_{\tau-1}^2) + c(I_\tau, I_{\tau-1}) + \gamma\mathcal{P}^\pi_{s's_\tau}\mathcal{P}^a\psi'(s_\tau, I(\tau')))\right) \right|$$
$$= \gamma \left| \mathcal{P}^\pi_{s's_\tau}\mathcal{P}^a\psi(s_\tau, I(\tau')) - \mathcal{P}^\pi_{s's_\tau}\mathcal{P}^a\psi'(s_\tau, I(\tau')))\right|$$
$$\leq \gamma \left\|P\psi - P\psi'\right\|$$
$$\leq \gamma \left\|\psi - \psi'\right\|,$$

*using the fact that $P$ is non-expansive. The result can then be deduced easily by applying max on both sides.*

*We now prove iii). We split the proof of the statement into two cases:*

**Case 1:**

$$\mathcal{M}^{\pi,\pi^2}\psi(s_\tau, I(\tau)) - \left(\Theta(z_\tau, a_\tau, a_\tau^2, a_{\tau-1}^2) + \gamma\max_{a\in\mathcal{A}} \mathcal{P}^a_{s's_\tau}\psi'(s', I(\tau))\right) < 0. \qquad (26)$$

*We now observe the following:*

$$\mathcal{M}^{\pi,\pi^2}\psi(s_\tau, I(\tau)) - \Theta(z_\tau, a_\tau, a_\tau^2, a_{\tau-1}^2) + \gamma\max_{a\in\mathcal{A}} \mathcal{P}^a_{s's_\tau}\psi'(s', I(\tau))$$
$$\leq \max\left\{\Theta(z_\tau, a_\tau, a_\tau^2, a_{\tau-1}^2) + \gamma\mathcal{P}^\pi_{s's_\tau}\mathcal{P}^a\psi(s', I(\tau)), \mathcal{M}^{\pi,\pi^2}\psi(s_\tau, I(\tau))\right\}$$
$$- \Theta(z_\tau, a_\tau, a_\tau^2, a_{\tau-1}^2) + \gamma\max_{a\in\mathcal{A}} \mathcal{P}^a_{s's_\tau}\psi'(s', I(\tau))$$
$$\leq \left| \max\left\{\Theta(z_\tau, a_\tau, a_\tau^2, a_{\tau-1}^2) + \gamma\mathcal{P}^\pi_{s's_\tau}\mathcal{P}^a\psi(s', I(\tau)), \mathcal{M}^{\pi,\pi^2}\psi(s_\tau, I(\tau))\right\} \right.$$
$$- \max\left\{\Theta(z_\tau, a_\tau, a_\tau^2, a_{\tau-1}^2) + \gamma\max_{a\in\mathcal{A}} \mathcal{P}^a_{s's_\tau}\psi'(s', I(\tau)), \mathcal{M}^{\pi,\pi^2}\psi(s_\tau, I(\tau))\right\}$$
$$+ \max\left\{\Theta(z_\tau, a_\tau, a_\tau^2, a_{\tau-1}^2) + \gamma\max_{a\in\mathcal{A}} \mathcal{P}^a_{s's_\tau}\psi'(s', I(\tau)), \mathcal{M}^{\pi,\pi^2}\psi(s_\tau, I(\tau))\right\}$$
$$\left. - \Theta(z_\tau, a_\tau, a_\tau^2, a_{\tau-1}^2) + \gamma\max_{a\in\mathcal{A}} \mathcal{P}^a_{s's_\tau}\psi'(s', I(\tau))\right|$$

$$\leq \left| \max\left\{ \Theta(z_\tau, a_\tau, a_\tau^2, a_{\tau-1}^2) + \gamma\max_{a\in\mathcal{A}} \mathcal{P}^a_{s's_\tau}\psi(s', I(\tau)), \mathcal{M}^{\pi,\pi^2}\psi(s_\tau, I(\tau)) \right\} \right.$$

$$\left. - \max\left\{ \Theta(z_\tau, a_\tau, a_\tau^2, a_{\tau-1}^2) + \gamma\max_{a\in\mathcal{A}} \mathcal{P}^a_{s's_\tau}\psi'(s', I(\tau)), \mathcal{M}^{\pi,\pi^2}\psi(s_\tau, I(\tau)) \right\} \right|$$

$$+ \left| \max\left\{ \Theta(z_\tau, a_\tau, a_\tau^2, a_{\tau-1}^2) + \gamma\max_{a\in\mathcal{A}} \mathcal{P}^a_{s's_\tau}\psi'(s', I(\tau)), \mathcal{M}^{\pi,\pi^2}\psi(s_\tau, I(\tau)) \right\} \right.$$

$$\left. - \Theta(z_\tau, a_\tau, a_\tau^2, a_{\tau-1}^2) + \gamma\max_{a\in\mathcal{A}} \mathcal{P}^a_{s's_\tau}\psi'(s', I(\tau)) \right|$$

$$\leq \gamma\max_{a\in\mathcal{A}} \left| \mathcal{P}^\pi_{s's_\tau}\mathcal{P}^a\psi(s', I(\tau)) - \mathcal{P}^\pi_{s's_\tau}\mathcal{P}^a\psi'(s', I(\tau)) \right|$$

$$+ \left| \max\left\{ 0, \mathcal{M}^{\pi,\pi^2}\psi(s_\tau, I(\tau)) - \left( \Theta(z_\tau, a_\tau, a_\tau^2, a_{\tau-1}^2) + \gamma\max_{a\in\mathcal{A}} \mathcal{P}^a_{s's_\tau}\psi'(s', I(\tau)) \right) \right\} \right|$$

$$\leq \gamma\|P\psi - P\psi'\|$$

$$\leq \gamma\|\psi - \psi'\|,$$

*where we have used the fact that for any scalars $a, b, c$ we have that $|\max\{a, b\} - \max\{b, c\}| \leq |a - c|$ and the non-expansiveness of $P$.*

***Case 2:***

$$\mathcal{M}^{\pi,\pi^2}\psi(s_\tau, I(\tau)) - \left( \Theta(z_\tau, a_\tau, a_\tau^2, a_{\tau-1}^2) + \gamma\max_{a\in\mathcal{A}} \mathcal{P}^a_{s's_\tau}\psi'(s', I(\tau)) \right) \geq 0.$$

*For this case, first recall that for any $\tau \in \mathcal{F}$, $-c(I_\tau, I_{\tau-1}) > \lambda$ for some $\lambda > 0$.*

$$\mathcal{M}^{\pi,\pi^2}\psi(s_\tau, I(\tau)) - \left( \Theta(z_\tau, a_\tau, a_\tau^2, a_{\tau-1}^2) + \gamma\max_{a\in\mathcal{A}} \mathcal{P}^a_{s's_\tau}\psi'(s', I(\tau)) \right)$$

$$\leq \mathcal{M}^{\pi,\pi^2}\psi(s_\tau, I(\tau)) - \left( \Theta(z_\tau, a_\tau, a_\tau^2, a_{\tau-1}^2) + \gamma\max_{a\in\mathcal{A}} \mathcal{P}^a_{s's_\tau}\psi'(s', I(\tau)) \right) - c(I_\tau, I_{\tau-1})$$

$$\leq \Theta(z_\tau, a_\tau, a_\tau^2, a_{\tau-1}^2) + c(I_\tau, I_{\tau-1}) + \gamma\mathcal{P}^\pi_{s's_\tau}\mathcal{P}^a\psi(s', I(\tau'))$$

$$- \left( \Theta(z_\tau, a_\tau, a_\tau^2, a_{\tau-1}^2) + c(I_\tau, I_{\tau-1}) + \gamma\max_{a\in\mathcal{A}} \mathcal{P}^a_{s's_\tau}\psi'(s', I(\tau)) \right)$$

$$\leq \gamma\max_{a\in\mathcal{A}} \left| \mathcal{P}^\pi_{s's_\tau}\mathcal{P}^a \left( \psi(s', I(\tau')) - \psi'(s', I(\tau)) \right) \right|$$

$$\leq \gamma \left| \psi(s', I(\tau')) - \psi'(s', I(\tau)) \right|$$

$$\leq \gamma \|\psi - \psi'\|,$$

*again using the fact that $P$ is non-expansive. Hence we have succeeded in showing that for any $\Lambda \in L_2$ we have that*

$$\left\| \mathcal{M}^{\pi,\pi^2}\Lambda - \max_{a\in\mathcal{A}} \left[ \psi(\cdot, a) + \gamma\mathcal{P}^a\Lambda' \right] \right\| \leq \gamma \|\Lambda - \Lambda'\|. \tag{27}$$

*Gathering the results of the three cases gives the desired result.*

PROOF OF PART **II**

*To prove Part **II**, we prove the following result:*

**Proposition 4** *For any $\pi \in \Pi$ and for any Shaper policy $\pi^2$, there exists a function $B^{\pi,\pi^2} : \mathcal{S} \times \{0, 1\} \to \mathbb{R}$ such that*

$$v_i^{\pi,\pi^2}(z) - v_i^{\pi',\pi^2}(z) = B^{\pi,\pi^2}(z) - B^{\pi',\pi^2}(z), \quad \forall z \equiv (s, I_0) \in \mathcal{S} \times \{0, 1\} \tag{28}$$

*where in particular the function $B$ is given by:*

$$B^{\pi,\pi^2}(s_0, I_0) = \mathbb{E}_{\pi,\pi^2}\left[\sum_{t=0}^{\infty} \gamma^t \left(R + \sum_{k\geq 1} c(I_t, I_{t-1})\delta^t_{\tau_{2k-1}}\right)\right], \tag{29}$$

*for any $(s_0, I_0) \in \mathcal{S} \times \{0,1\}$.*

**Proof 6** *Note that by the deduction of (ii) in Prop 1, we immediately observe that*

$$\hat{v}_2^{\pi,\pi^2}(s_0, I_0) = B^{\pi,\pi^2}(s_0, I_0), \quad \forall (s_0, I_0) \in \mathcal{S} \times \{0,1\}. \tag{30}$$

*We therefore immediately deduce that for any two Shaper policies $\pi^2$ and $\pi'^2$ the following expression holds $\forall (s_0, I_0) \in \mathcal{S} \times \{0,1\}$:*

$$\hat{v}_2^{\pi,\pi^2}(s_0, I_0) - \hat{v}_2^{\pi,\pi'^2}(s_0, I_0) = B^{\pi,\pi^2}(s_0, I_0) - B^{\pi,\pi'^2}(s_0, I_0). \tag{31}$$

*Our aim now is to show that the following expression holds $\forall (s_0, I_0) \in \mathcal{S} \times \{0,1\}$:*

$$v_1^{\pi,\pi^2}(I_0, s_0) - v_1^{\pi',\pi^2}(I_0, s_0) = B^{\pi,\pi^2}(I_0, s_0) - B^{\pi',\pi^2}(I_0, s_0), \quad \forall i \in \mathcal{N}$$

*For the finite horizon case, the result is proven by induction on the number of time steps until the end of the game. Unlike the infinite horizon case, for the finite horizon case the value function and policy have an explicit time dependence.*

*We consider the case of the proposition at time $T-1$ that is we evaluate the value functions at the penultimate time step. In this case, we have that:*

$$\mathbb{E}_{s_{T-1}\sim d_\theta}\left[B_{T-1}^{\pi,\pi^2}(I_{T-1}, s_{T-1}) - B_{T-1}^{\pi',\pi^2}(I_{T-1}, s_{T-1})\right]$$

$$= \mathbb{E}_{s_{T-1}\sim d_\theta}\left[\sum_{a_{T-1}\in\mathcal{A}} \pi(a_{T-1}; s_{T-1})\left[R(s_{T-1}, a_{T-1}) + \sum_{k\geq 0}\sum_{j=T-1}^{\infty} c(I_j, I_{j-1})\delta^j_{\tau_k}\right]\right.$$

$$+ \gamma \sum_{s_T\in S}\sum_{a_{T-1}\in\mathcal{A}} \pi(a_{T-1}; s_{T-1})P(s_T; a_{T-1})B_T^{\pi,\pi^2}(I_T, s_T)$$

$$- \left(\sum_{a'_{T-1}\in\mathcal{A}} \pi'(a'_{T-1}; s_{T-1})\left[R(s_{T-1}, a'_{T-1}) + \sum_{k\geq 0}\sum_{j=T-1}^{\infty} c(I_j, I_{j-1})\delta^j_{\tau_k}\right]\right.$$

$$\left.\left. + \gamma \sum_{s_T\in S}\sum_{a'_{T-1}\in\mathcal{A}} \pi'(a'_{T-1}; s_{T-1})P(s_T; a'_{T-1})B_T^{\pi',\pi^2}(I_T, s_T)\right)\right]$$

$$= \mathbb{E}_{s_{T-1}\sim d_\theta}\left[\sum_{a_{T-1}\in\mathcal{A}} \pi(a_{T-1}; s_{T-1})R(s_{T-1}, a_{T-1})\right.$$

$$- \sum_{a'_{T-1}\in\mathcal{A}} \pi'(a'_{T-1}; s_{T-1})R(s_{T-1}, a'_{T-1})$$

$$+ \gamma\left[\sum_{s_T\in\mathcal{S}}\sum_{a_{T-1}\in\mathcal{A}} \pi(a_{T-1}; s_{T-1})P(s_T; a_{T-1})B_T^{\pi,\pi^2}(I_T, s_T)\right.$$

$$\left.\left. - \sum_{s_T\in\mathcal{S}}\sum_{a'_{T-1}\in\mathcal{A}} \pi'(a'_{T-1}; s_{T-1})P(s_T; a'_{T-1})B_T^{\pi',\pi^2}(I_T, s_T)\right]\right]. \tag{32}$$

*We now observe that for any $\pi \in \Pi$ and for any $\pi^2$ we have that $B_T^{\pi,\pi^2}(I_T, s_T) = \mathbb{E}\left[\sum_{a_T\in\mathcal{A}} \pi(a_T; s_T)\left[R(s_T, a_T) + \sum_{k\geq 0}\sum_{j=T}^{\infty} c(I_j, I_{j-1})\delta^j_{\tau_k}\right]\right]$, moreover we have that for any*

$\pi \in \Pi$ *and for any* $\pi^2$

$$\mathbb{E}\left[\sum_{a_T \in \mathcal{A}} \pi(a_T; s_T)\left[R(s_T, a_T) + \sum_{k \geq 0}\sum_{j=T}^{\infty} c(I_j, I_{j-1})\delta_{\tau_k}^j\right]\right.$$

$$\left. - \sum_{a'_T \in \mathcal{A}} \pi'(a_T; s_T)\left[R(s_T, a'_T) + \sum_{k \geq 0}\sum_{j=T}^{\infty} c(I_j, I_{j-1})\delta_{\tau_k}^j\right]\right]$$

$$= \mathbb{E}\left[\sum_{a_T \in \mathcal{A}} \pi(a_T; s_T)R(s_T, a_T) - \sum_{a'_T \in \mathcal{A}} \pi'(a'_T; s_T)R(s_T, a_T)\right]$$

$$+ \mathbb{E}\left[\sum_{a_T \in \mathcal{A}} \pi(a_T; s_T)\sum_{k \geq 0}\sum_{j=T}^{\infty} c(I_j, I_{j-1})\delta_{\tau_k}^j - \sum_{a'_T \in \mathcal{A}} \pi'(a'_T; s_T)\sum_{k \geq 0}\sum_{j=T}^{\infty} c(I_j, I_{j-1})\delta_{\tau_k}^j\right].$$

*Hence we find that*

$$\mathbb{E}_{s_{T-1} \sim d_\theta}\left[B_{T-1}^{\pi, \pi^2}(I_{T-1}, s_{T-1}) - B_{T-1}^{\pi', \pi^2}(I_{T-1}, s_{T-1})\right]$$

$$= \mathbb{E}_{s_{T-1} \sim d_\theta}\left[\sum_{a_{T-1} \in \mathcal{A}} \pi(a_{T-1}; s_{T-1})R(s_{T-1}, a_{T-1})\right.$$

$$\left. - \sum_{a'_{T-1} \in \mathcal{A}} \pi'(a'_{T-1}; s_{T-1})R(s_{T-1}, a'_{T-1})\right) \tag{33}$$

$$+ \gamma\left(\sum_{s_T \in \mathcal{S}}\sum_{a_{T-1} \in \mathcal{A}}\sum_{a_T \in \mathcal{A}} \pi(a_{T-1}; s_{T-1})P(s_T; a_{T-1})\pi(a_T; s_T)R(s_T, a_T)\right. \tag{34}$$

$$- \sum_{s_T \in \mathcal{S}}\sum_{a'_{T-1} \in \mathcal{A}}\sum_{a'_T \in \mathcal{A}} \pi'(a'_{T-1}; s_{T-1})P(s_T; a'_{T-1})\pi'(a'_T; s_T)R(s_T, a_T) \tag{35}$$

$$+ \sum_{s_T \in \mathcal{S}}\sum_{a_{T-1} \in \mathcal{A}}\sum_{a_T \in \mathcal{A}}\sum_{k \geq 0}\sum_{j=T}^{\infty} \pi(a_{T-1}; s_{T-1})P(s_T; a_{T-1})\pi(a_T; s_T)c(I_j, I_{j-1})\delta_{\tau_k}^j \tag{36}$$

$$\left.\left. - \sum_{s_T \in \mathcal{S}}\sum_{a'_{T-1} \in \mathcal{A}}\sum_{a'_T \in \mathcal{A}}\sum_{k \geq 0}\sum_{j=T}^{\infty} \pi'(a'_{T-1}; s_{T-1})P(s_T; a'_{T-1})\pi'(a'_T; s_T)c(I_j, I_{j-1})\delta_{\tau_k}^j\right)\right].$$
$$\tag{37}$$

*Now*

$$\mathbb{E}_{s_{T-1} \sim d_\theta}\left[\sum_{s_T \in \mathcal{S}}\sum_{a_{T-1} \in \mathcal{A}}\sum_{a_T \in \mathcal{A}}\sum_{k \geq 0}\sum_{j=T}^{\infty} \pi(a_{T-1}; s_{T-1})P(s_T; a_{T-1})\pi(a_T; s_T)c(I_j, I_{j-1})\delta_{\tau_k}^j \right.\tag{38}$$

$$\left. - \sum_{s_T \in \mathcal{S}}\sum_{a'_{T-1} \in \mathcal{A}}\sum_{a'_T \in \mathcal{A}}\sum_{k \geq 0}\sum_{j=T}^{\infty} \pi'(a'_{T-1}; s_{T-1})P(s_T; a'_{T-1})\pi'(a'_T; s_T)c(I_j, I_{j-1})\delta_{\tau_k}^j\right] \tag{39}$$

$$= \mathbb{E}_{s_{T-1} \sim d_\theta}\left[\sum_{s_T \in \mathcal{S}}\sum_{a_{T-1} \in \mathcal{A}}\sum_{a_T \in \mathcal{A}} \pi(a_{T-1}; s_{T-1})P(s_T; a_{T-1})\pi(a_T; s_T)\sum_{k \geq 0}\sum_{j=T}^{\infty} c(I_j, I_{j-1})\delta_{\tau_k}^j\right.$$
$$\tag{40}$$

$$\left. - \sum_{s_T \in \mathcal{S}}\sum_{a'_{T-1} \in \mathcal{A}}\sum_{a'_T \in \mathcal{A}} \pi'(a'_{T-1}; s_{T-1})P(s_T; a'_{T-1})\pi'(a'_T; s_T)\sum_{k \geq 0}\sum_{j=T}^{\infty} c(I_j, I_{j-1})\delta_{\tau_k}^j\right] \tag{41}$$

$$= \mathbb{E}_{s_{T-1} \sim d_\theta}\left[\sum_{s_T \in \mathcal{S}}\sum_{a_{T-1} \in \mathcal{A}}\sum_{a_T \in \mathcal{A}} \pi(a_{T-1}; s_{T-1})P(s_T; a_{T-1})\pi(a_T; s_T)\sum_{k \geq 0}\sum_{j=T}^{\infty} c(I_j, I_{j-1})\delta_{\tau_k}^j\right.$$
$$\tag{42}$$

$$- \sum_{s_T \in \mathcal{S}} \sum_{a'_{T-1} \in \mathcal{A}} \sum_{a'_T \in \mathcal{A}} \pi'(a'_{T-1}; s_{T-1}) P(s_T; a'_{T-1}) \pi'(a'_T; s_T) \sum_{k \geq 0} \sum_{j=T}^{\infty} c(I_j, I_{j-1}) \delta_{\tau_k}^j \Bigg] \tag{43}$$

$$= K \mathbb{E}_{s_{T-1} \sim d_\theta} \Bigg[ \sum_{s_T \in \mathcal{S}} \sum_{a_{T-1} \in \mathcal{A}} \sum_{a_T \in \mathcal{A}} \pi(a_{T-1}; s_{T-1}) P(s_T; a_{T-1}) \pi(a_T; s_T) \tag{44}$$

$$- \sum_{s_T \in \mathcal{S}} \sum_{a'_{T-1} \in \mathcal{A}} \sum_{a'_T \in \mathcal{A}} \pi'(a'_{T-1}; s_{T-1}) P(s_T; a'_{T-1}) \pi'(a'_T; s_T) \Bigg] \tag{45}$$

$$= K \left( \sum_{a_T \in \mathcal{A}} \pi(a_T) - \sum_{a'_T \in \mathcal{A}} \pi'(a'_T) \right) = 0. \tag{46}$$

*Hence, we find that*

$$\mathbb{E}_{s_{T-1} \sim d_\theta} \left[ B_{T-1}^{\pi, \pi^2}(I_{T-1}, s_{T-1}) - B_{T-1}^{\pi', \pi^2}(I_{T-1}, s_{T-1}) \right]$$

$$= \mathbb{E}_{s_{T-1} \sim d_\theta} \Bigg[ \sum_{a_{T-1} \in \mathcal{A}} \pi(a_{T-1}; s_{T-1}) R(s_{T-1}, a_{T-1})$$
$$- \sum_{a'_{T-1} \in \mathcal{A}} \pi'(a'_{T-1}; s_{T-1}) R(s_{T-1}, a'_{T-1}) \tag{47}$$

$$+ \gamma \left( \sum_{s_T \in \mathcal{S}} \sum_{a_{T-1} \in \mathcal{A}} \sum_{a_T \in \mathcal{A}} \pi(a_{T-1}; s_{T-1}) P(s_T; a_{T-1}) \pi(a_T; s_T) R(s_T, a_T) \right. \tag{48}$$

$$\left. - \sum_{s_T \in \mathcal{S}} \sum_{a'_{T-1} \in \mathcal{A}} \sum_{a'_T \in \mathcal{A}} \pi'(a'_{T-1}; s_{T-1}) P(s_T; a'_{T-1}) \pi'(a'_T; s_T) R(s_T, a_T) \right) \Bigg] \tag{49}$$

$$= \mathbb{E}_{s_{T-1} \sim d_\theta} \left[ v_{i, T-1}^{\pi, \pi^2}(s_{T-1}) - v_{i, T-1}^{\pi', \pi^2}(s_{T-1}) \right]. \tag{50}$$

*Hence, we have succeeded in proving that the expression (28) holds for $T - k$ when $k = 1$.*

*Our next goal is to prove that the expression holds for any $0 < k \leq T$.*

*Note that for any $T \geq k > 0$, we can write $B_{T-k}^{\pi, \pi^2}$ as*

$$B_{T-k}^{\pi, \pi^2}(I_0, s_0) = \mathbb{E}_\pi \Bigg[ R(s_{T-k}, a_{T-k}) + \sum_{k \geq 0} \sum_{j=T-j}^{\infty} c(I_j, I_{j-1}) \delta_{\tau_k}^j \tag{51}$$

$$+ \gamma \sum_{s_{k+1} \in S} P(s'; s_{T-k}, a_{T-k}) B_{T-(k+1)}^{\pi, \pi^2}(I_{T-(k+1)}, s_{T-(k+1)}) \Bigg]. \tag{52}$$

*Now we consider the case when we evaluate the expression (28) for any $0 < k \leq T$. Our inductive hypothesis is the the expression holds for some $0 < k \leq T$, that is for any $0 < k \leq T$ we have that:*

$$\sum_{s_{T-k} \in \mathcal{S}} \sum_{a_{T-(k+1)} \in \mathcal{A}} \pi(a_{T-(k+1)}; s_{T-(k+1)}) P(s_{T-k}; a_{T-(k+1)}) v_{i, T-k}^{\pi, \pi^2}(I_{T-k}, s_{T-k})$$

$$- \sum_{s_{T-k} \in \mathcal{S}} \sum_{a'_{T-(k+1)} \in \mathcal{A}} \pi'(a'_{T-(k+1)}; s_{T-(k+1)}) P(s_{T-k}; a'_{T-(k+1)}) v_{i, T-k}^{\pi', \pi^2}(I_{T-k}, s_{T-k}) \tag{53}$$

$$= \sum_{s_{T-k} \in \mathcal{S}} \sum_{a_{T-(k+1)} \in \mathcal{A}} \pi(a_{T-(k+1)}; s_{T-(k+1)}) P(s_{T-k}; a_{T-(k+1)}) B_{T-k}^{\pi, \pi^2}(I_{T-k}, s_{T-k}) \tag{54}$$

$$- \sum_{s_{T-k} \in \mathcal{S}} \sum_{a'_{T-(k+1)} \in \mathcal{A}} \pi'(a'_{T-(k+1)}; s_{T-(k+1)}) P(s_{T-k}; a'_{T-(k+1)}) B_{T-k}^{\pi', \pi^2}(I_{T-k}, s_{T-k}). \tag{55}$$

*It remains to show that the expression holds for $k + 1$ time steps prior to the end of the horizon. The result can be obtained using the dynamic programming principle and the base case ($k = 1$) result, indeed we have that*

$$\mathbb{E}_{s_{T-(k+1)} \sim d_\theta} \left[ B^{\pi, \pi^2}_{T-(k+1)}(I_{T-(k+1)}, s_{T-(k+1)}) - B^{\pi', \pi^2}_{T-(k+1)}(I_{T-(k+1)}, s_{T-(k+1)}) \right]$$

$$= \mathbb{E}_{s_{T-(k+1)} \sim d_\theta} \left[ \sum_{a_{T-(k+1)} \in \mathcal{A}} \pi(a_{T-(k+1)}; s_{T-(k+1)}) \phi(s_{T-(k+1)}, a_{T-(k+1)}) \right.$$

$$- \sum_{a'_{T-(k+1)} \in \mathcal{A}} \pi'(a'_{T-(k+1)}; s_{T-(k+1)}) \phi(s_{T-(k+1)}, a'_{T-(k+1)})$$

$$+ \gamma \sum_{s_{T-k} \in S} \sum_{a_{T-(k+1)} \in \mathcal{A}} \pi(a_{T-(k+1)}; s_{T-(k+1)}) P(s_{T-k}; a_{T-(k+1)}) B^{\pi, \pi^2}_{T-k}(I_{T-k}, s_{T-k})$$

$$\left. - \gamma \sum_{s_{T-k} \in S} \sum_{a'_{T-(k+1)} \in \mathcal{A}} \pi'(a'_{T-(k+1)}; s_{T-(k+1)}) P(s_{T-k}; a'_{T-(k+1)}) B^{\pi', \pi^2}_{T-k}(I_{T-k}, s_{T-k}) \right]$$

$$= \mathbb{E}_{s_{T-(k+1)} \sim d_\theta} \left[ \sum_{a_{T-(k+1)} \in \mathcal{A}} \pi(a_{T-(k+1)}; s_{T-(k+1)}) \left[ R(s_{T-(k+1)}, a_{T-(k+1)}) + \sum_{k \geq 0} \sum_{j=T-1}^{\infty} c(I_j, I_{j-1}) \delta^j_{\tau_k} \right] \right.$$

$$+ \gamma \sum_{s_{T-k} \in S} \sum_{a_{T-(k+1)} \in \mathcal{A}} \pi(a_{T-(k+1)}; s_{T-(k+1)}) P(s_{T-k}; a_{T-(k+1)}) B^{\pi, \pi^2}_{T-k}(I_{T-k}, s_{T-k})$$

$$- \left( \sum_{a'_{T-(k+1)} \in \mathcal{A}} \pi'(a'_{T-(k+1)}; s_{T-(k+1)}) \left[ R(s_{T-(k+1)}, a'_{T-(k+1)}) + \sum_{k \geq 0} \sum_{j=T-1}^{\infty} c(I_j, I_{j-1}) \delta^j_{\tau_k} \right] \right.$$

$$\left. \left. + \gamma \sum_{s_{T-k} \in S} \sum_{a'_{T-(k+1)} \in \mathcal{A}} \pi'(a'_{T-(k+1)}; s_{T-(k+1)}) P(s_{T-k}; a'_{T-(k+1)}) B^{\pi', \pi^2}_{T-k}(I_{T-k}, s_{T-k}) \right] \right)$$

$$= \mathbb{E}_{s_{T-(k+1)} \sim d_\theta} \left[ \sum_{a_{T-(k+1)} \in \mathcal{A}} \pi(a_{T-(k+1)}; s_{T-(k+1)}) R(s_{T-(k+1)}, a_{T-(k+1)}) \right.$$

$$- \sum_{a'_{T-(k+1)} \in \mathcal{A}} \pi'(a'_{T-(k+1)}; s_{T-(k+1)}) R(s_{T-(k+1)}, a'_{T-(k+1)})$$

$$+ \gamma \left[ \sum_{s_{T-k} \in \mathcal{S}} \sum_{a_{T-(k+1)} \in \mathcal{A}} \pi(a_{T-(k+1)}; s_{T-(k+1)}) P(s_{T-k}; a_{T-(k+1)}) B^{\pi, \pi^2}_{T-k}(I_{T-k}, s_{T-k}) \right.$$

$$\left. \left. - \sum_{s_{T-k} \in \mathcal{S}} \sum_{a'_{T-(k+1)} \in \mathcal{A}} \pi'(a'_{T-(k+1)}; s_{T-(k+1)}) P(s_{T-k}; a'_{T-(k+1)}) B^{\pi', \pi^2}_{T-k}(I_{T-k}, s_{T-k}) \right] \right].$$

$$\tag{56}$$

$$= \mathbb{E}_{s_{T-(k+1)} \sim d_\theta} \left[ \sum_{a_{T-(k+1)} \in \mathcal{A}} \pi(a_{T-(k+1)}; s_{T-(k+1)}) R(s_{T-(k+1)}, a_{T-(k+1)}) \right.$$

$$- \sum_{a'_{T-(k+1)} \in \mathcal{A}} \pi'(a'_{T-(k+1)}; s_{T-(k+1)}) R(s_{T-(k+1)}, a'_{T-(k+1)})$$

$$+ \gamma \left[ \sum_{s_{T-k} \in \mathcal{S}} \sum_{a_{T-(k+1)} \in \mathcal{A}} \pi(a_{T-(k+1)}; s_{T-(k+1)}) P(s_{T-k}; a_{T-(k+1)}) v^{\pi, \pi^2}_{i, T-k}(I_{T-k}, s_{T-k}) \right.$$

$$\left. \left. - \sum_{s_{T-k} \in \mathcal{S}} \sum_{a'_{T-(k+1)} \in \mathcal{A}} \pi'(a'_{T-(k+1)}; s_{T-(k+1)}) P(s_{T-k}; a'_{T-(k+1)}) v^{\pi', \pi^2}_{i, T-k}(I_{T-k}, s_{T-k}) \right] \right].$$

$$\tag{57}$$

$$= \mathbb{E}_{s_{T-(k+1)} \sim d_\theta} \left[ v_{i,T-(k+1)}^{\pi,\pi^2}(I_{T-(k+1)}, s_{T-(k+1)}) - v_{T-(k+1)}^{\pi',\pi^2}(I_{T-(k+1)}, s_{i,T-(k+1)}) \right], \qquad (58)$$

*using the inductive hypothesis and where we have used the fact that*

$$\mathbb{E}_{s_{T-(k+1)} \sim d_\theta} \Bigg[ \sum_{s_{T-k} \in \mathcal{S}} \sum_{a_{T-(k+1)} \in \mathcal{A}} \sum_{a_{T-k} \in \mathcal{A}} \sum_{k \geq 0} \sum_{j=T}^{\infty} \pi(a_{T-(k+1)}; s_{T-(k+1)}) P(s_{T-k}; a_{T-(k+1)}) \pi(a_{T-k}; s_{T-k}) c(I_j, I_{j-1}) \delta_{\tau_k}^j$$

$$(59)$$

$$- \sum_{s_{T-k} \in \mathcal{S}} \sum_{a'_{T-(k+1)} \in \mathcal{A}} \sum_{a'_{T-k} \in \mathcal{A}} \sum_{k \geq 0} \sum_{j=T}^{\infty} \pi'(a'_{T-(k+1)}; s_{T-(k+1)}) P(s_{T-k}; a'_{T-(k+1)}) \pi'(a'_{T-k}; s_{T-k}) c(I_j, I_{j-1}) \delta_{\tau_k}^j \Bigg]$$

$$(60)$$

$$= K \left( \sum_{a_{T-k} \in \mathcal{A}} \pi(a_{T-k}) - \sum_{a'_{T-k} \in \mathcal{A}} \pi'(a'_{T-k}) \right) = 0, \qquad (61)$$

*via similar reasoning as before and after which which we deduce the result in the finite case.*

*For the infinite horizon case, we must prove that there exists a measurable function $B : \Pi \times \mathcal{S} \to \mathbb{R}$ such that the following holds for any $i \in \mathcal{N}$ and $\forall \pi, \pi'_i \pi' \in \Pi, \forall \pi_{-i} \pi' \in \Pi_{-i}$ and $\forall s \in \mathcal{S}$:*

$$\mathbb{E}_{s \sim P} \left[ \left( v_i^{\pi,\pi^2} - v_i^{\pi',\pi^2} \right)(z) \right] = \mathbb{E}_{s \sim P} \left[ \left( B^{\pi,\pi^2} - B^{\pi',\pi^2} \right)(z) \right]. \qquad (62)$$

*The result is proven by contradiction.*

*To this end, let us firstly assume $\exists c \neq 0$ such that*

$$\mathbb{E}_{s \sim P} \left[ \left( v_i^{\pi,\pi^2} - v_i^{\pi',\pi^2} \right)(z) \right] - \mathbb{E}_{s \sim P} \left[ \left( B_i^{\pi,\pi^2} - B_i^{\pi',\pi^2} \right)(z) \right] = c.$$

*Let us now define the following quantities for any $s \in \mathcal{S}$ and for each $\pi\pi' \in \Pi$ and $\pi_{-i}\pi' \in \Pi_{-i}$ and $\forall i \in \mathcal{N}$:*

$$v_{i,T'}^{\pi,\pi^2}(z) := \sum_{t=0}^{T'} \mu(s_0)\pi(a_0^i, s_0)\pi_{-i}(a_0^{-i}, s_0) \prod_{j=0}^{t-1} \sum_{s_{j+1} \in \mathcal{S}} \gamma^t P(s_{j+1}; s_j, a_j)\pi(a_j^i|s_j)\pi_{-i}(a_j^{-i}|s_j)R_i(z_j, a_j),$$

*and*

$$B_{T'}^{\pi,\pi^2}(z) := \sum_{t=0}^{T'} \mu(s_0)\pi(a_0^i, s_0)\pi_{-i}(a_0^{-i}, s_0) \prod_{j=0}^{t-1} \sum_{s_{j+1} \in \mathcal{S}} P(s_{j+1}; s_j, a_j) \cdot \pi(a_j^i|s_j)\pi_{-i}(a_j^{-i}|s_j)\Theta(z_j, a_j),$$

*so that the quantity $v_{i,T'}^{\pi}(s)$ measures the expected cumulative return until the point $T' < \infty$.*

*Hence, we deduce that*

$$v_i^\pi(z) \equiv v_{i,\infty}^\pi(z)$$

$$= v_{i,T'}^\pi(z) + \gamma^{T'} \mu(s_0)\pi(a_0^i, s_0)\pi_{-i}(a_0^{-i}, s_0) \prod_{j=0}^{T'-1} \sum_{s_{j+1} \in \mathcal{S}} \gamma^t P(s_{j+1}; s_j, a_j)\pi(a_j^i|s_j)\pi_{-i}(a_j^{-i}|s_j)v_i^\pi(s_{T'}).$$

*Next we observe that:*

$$c = \mathbb{E}_{s \sim P} \left[ \left( v_i^{\pi,\pi^2} - v_i^{\pi',\pi^2} \right)(z) \right] - \mathbb{E}_{s \sim P} \left[ \left( B^{\pi,\pi^2} - B^{\pi',\pi^2} \right)(z) \right]$$

$$= \mathbb{E}_{s \sim P} \left[ \left( v_{i,T'}^{\pi,\pi^2} - v_{i,T'}^{\pi',\pi^2} \right)(z) \right] - \mathbb{E}_{s \sim P} \left[ \left( B_{T'}^{\pi,\pi^2} - B_{T'}^{\pi',\pi^2} \right)(s) \right]$$

$$+ \gamma^{T'} \mathbb{E}_{s_{T'} \sim P} \left[ \mu(s_0)\pi(a_0^i, s_0)\pi_{-i}(a_0^{-i}, s_0) \prod_{j=0}^{T'-1} \sum_{s_{j+1} \in \mathcal{S}} P(s_{j+1}; s_j, a_j)\pi(a_j^i|s_j)\pi_{-i}(a_j^{-i}|s_j) \left( v_i^{\pi,\pi^2}(z_{T'}) - B^{\pi,\pi^2}(z_{T'}) \right) \right]$$

$$- \mu(s_0)\pi'_i(a'^i_0, s_0)\pi_{-i}(a^{-i}_0, s_0) \prod_{j=0}^{T'-1} \sum_{s_{j+1} \in \mathcal{S}} P(s_{j+1}; s_j, a'_j)\pi'_i(a'^i_j|s_j)\pi_{-i}(a^{-i}_j|s_j) \left( v_i^{\pi',\pi^2}(z_{T'}) - B^{\pi',\pi^2}(z_{T'}) \right) \Bigg].$$

*Considering the last expectation and its coefficient and denoting the product by $\kappa$, using the fact that by the Cauchy-Schwarz inequality we have $\|AX - BY\| \le \|A\|\|X\| + \|B\|\|Y\|$, moreover whenever $A, B$ are non-expansive we have that $\|AX - BY\| \le \|X\| + \|Y\|$, hence we observe the following $\kappa \le \|\kappa\| \le 2\gamma^{T'} (\|v_i\| + \|B\|)$. Since we can choose $T'$ freely and $\gamma \in ]0,1[$, we can choose $T'$ to be sufficiently large so that $\gamma^{T'} (\|v_i\| + \|B\|) < \frac{1}{4}|c|$. This then implies that*

$$\left| \mathbb{E}_{s \sim P} \left[ \left( v_{i,T'}^{\pi,\pi^2} - v_{i,T'}^{\pi',\pi^2} \right)(z) - \left( B_{T'}^{\pi,\pi^2} - B_{T'}^{\pi',\pi^2} \right)(z) \right] \right| > \frac{1}{2}c,$$

*which is a contradiction since we have proven that for any finite $T'$ it is the case that*

$$\mathbb{E}_{s \sim P} \left[ \left( v_{i,T'}^{\pi,\pi^2} - v_{i,T'}^{\pi',\pi^2} \right)(z) - \left( B_{T'}^{\pi,\pi^2} - B_{T'}^{\pi',\pi^2} \right)(z) \right] = 0,$$

*and hence we deduce the result in the infinite horizon case.*

### PROOF OF PART **III**

*We begin by recalling that a Markov strategy is a policy $\pi^i : \mathcal{S} \times \mathcal{A}_i \to [0,1]$ which requires as input only the current system state (and not the game history or the other player's action or strategy [22]). With this, we give a formal description of the stable points of $\mathcal{G}$ in Markov strategies.*

***Definition 3*** *A policy profile $\hat{\boldsymbol{\pi}} = (\hat{\pi}^1, \hat{\pi}^2) \in \boldsymbol{\Pi}$ is a Markov perfect equilibrium (MPE) if the following holds $\forall i \ne j \in \{1,2\}$, $\forall \hat{\pi}' \in \Pi_i$: $v_i^{(\hat{\pi}^i, \hat{\pi}^j)}(s_0, I_0) \ge v_i^{(\hat{\pi}', \hat{\pi}^j)}(s_0, I_0), \forall (s_0, I_0) \in \mathcal{S} \times \{0,1\}$.*

*The MPE describes a configuration in policies in which no player can increase their payoff by changing (unilaterally) their policy. Crucially, it defines the stable points to which independent learners converge (if they converge at all).*

### PROOF OF PROPOSITION 3

***Proof 7*** *We do the proof by contradiction. Let $\sigma = (\pi, g) \in \arg\sup_{\pi' \in \Pi, g'} B^{\pi', g'}(s)$ for any $s \in \mathcal{S}$. Let us now therefore assume that $\sigma \notin NE\{\mathcal{G}\}$, hence there exists some other strategy profile $\tilde{\sigma} = (g, \tilde{\pi})$ for which Controller has a profitable deviation where $\pi' \ne \pi$ i.e. $v_1^{\pi', \pi^2}(s) > v_1^{\pi, \pi^2}(s)$ (using the preservation of signs of integration). Prop. 4 however implies that $B^{\pi', \pi^2}(s) - B^{\pi, \pi^2}(s) > 0$ which is a contradiction since $\sigma = (\pi, g)$ is a maximum of B. The proof can be straightforwardly adapted to cover the case in which the deviating player is Shaper after which we deduce the desired result.*

*The last result completes the proof of Theorem 1.*

### PROOF OF PROPOSITION 2

***Proof 8 (Proof of Prop. 2)*** *The proof is given by establishing a contradiction. Therefore suppose that $\mathcal{M}^{\pi, \pi^2} \psi(s_{\tau_k}, I(\tau_k)) \le \psi(s_{\tau_k}, I(\tau_k))$ and suppose that the intervention time $\tau'_1 > \tau_1$ is an optimal intervention time. Construct the Player 2 $\pi'^2 \in \Pi^2$ and $\tilde{\pi}^2$ policy switching times by $(\tau'_0, \tau'_1, \dots,)$ and $\pi'^2 \in \Pi^2$ policy by $(\tau'_0, \tau_1, \dots)$ respectively. Define by $l = \inf\{t > 0; \mathcal{M}^{\pi, \pi^2} \psi(s_t, I_0) = \psi(s_t, I_0)\}$ and $m = \sup\{t; t < \tau'_1\}$. By construction we have that*

$$v_2^{\pi^1, \pi'^2}(s, I_0)$$
$$= \mathbb{E}\left[ R(s_0, a_0) + \mathbb{E}\left[ \dots + \gamma^{l-1}\mathbb{E}\left[ R(s_{\tau_1-1}, a_{\tau_1-1}) + \dots + \gamma^{m-l-1}\mathbb{E}\left[ R(s_{\tau'_1-1}, a_{\tau'_1-1}) + \gamma\mathcal{M}^{\pi^1, \pi'^2} v_2^{\pi^1, \pi'^2}(s', I(\tau'_1)) \right] \right] \right] \right]$$

$$< \mathbb{E}\left[R(s_0, a_0) + \mathbb{E}\left[\ldots + \gamma^{l-1}\mathbb{E}\left[R(s_{\tau_1-1}, a_{\tau_1-1}) + \gamma\mathcal{M}^{\pi^1, \tilde{\pi}^2}v_2^{\pi^1, \pi'^2}(s_{\tau_1}, I(\tau_1))\right]\right]\right]$$

*We now use the following observation* $\mathbb{E}\left[R(s_{\tau_1-1}, a_{\tau_1-1}) + \gamma\mathcal{M}^{\pi^1, \tilde{\pi}^2}v_2^{\pi^1, \pi'^2}(s_{\tau_1}, I(\tau_1))\right]$

$$\leq \max\left\{\mathcal{M}^{\pi^1, \tilde{\pi}^2}v_2^{\pi^1, \pi'^2}(s_{\tau_1}, I(\tau_1)), \max_{a_{\tau_1}\in\mathcal{A}}\left[R(s_{\tau_k}, a_{\tau_k}) + \gamma\sum_{s'\in\mathcal{S}}P(s'; a_{\tau_1}, s_{\tau_1})v_2^{\pi^1, \pi^2}(s', I(\tau_1))\right]\right\}$$

*Using this we deduce that*

$$v_2^{\pi^1, \pi'^2}(s, I_0) \leq \mathbb{E}\left[R(s_0, a_0) + \mathbb{E}\left[\ldots\right.\right.$$

$$+ \gamma^{l-1}\mathbb{E}\left[R(s_{\tau_1-1}, a_{\tau_1-1}) + \gamma\max\left\{\mathcal{M}^{\pi^1, \tilde{\pi}^2}v_2^{\pi^1, \pi'^2}(s_{\tau_1}, I(\tau_1)), \max_{a_{\tau_1}\in\mathcal{A}}\left[R(s_{\tau_k}, a_{\tau_k}) + \gamma\sum_{s'\in\mathcal{S}}P(s'; a_{\tau_1}, s_{\tau_1})v_2^{\pi^1, \pi^2}(s', I(\tau_1))\right.\right.\right.$$

$$= \mathbb{E}\left[R(s_0, a_0) + \mathbb{E}\left[\ldots + \gamma^{l-1}\mathbb{E}\left[R(s_{\tau_1-1}, a_{\tau_1-1}) + \gamma\left[Tv_2^{\pi^1, \tilde{\pi}^2}\right](s_{\tau_1}, I(\tau_1))\right]\right]\right] = v_2^{\pi^1, \tilde{\pi}^2}(s, I_0)),$$

*where the first inequality is true by assumption on $\mathcal{M}$. This is a contradiction since $\pi'^2$ is an optimal policy for Player 2. Using analogous reasoning, we deduce the same result for $\tau'_k < \tau_k$ after which deduce the result. Moreover, by invoking the same reasoning, we can conclude that it must be the case that $(\tau_0, \tau_1, \ldots, \tau_{k-1}, \tau_k, \tau_{k+1}, \ldots,)$ are the optimal switching times.*

## PROOF OF THEOREM 2

To prove the theorem, we make use of the following result:

**Theorem 3 (Theorem 1, pg 4 in [18])** *Let $\Xi_t(s)$ be a random process that takes values in $\mathbb{R}^n$ and given by the following:*

$$\Xi_{t+1}(s) = (1 - \alpha_t(s))\,\Xi_t(s)\alpha_t(s)L_t(s), \tag{63}$$

*then $\Xi_t(s)$ converges to $0$ with probability $1$ under the following conditions:*

    *i)  $0 \leq \alpha_t \leq 1, \sum_t \alpha_t = \infty$ and $\sum_t \alpha_t < \infty$*

    *ii)  $\|\mathbb{E}[L_t|\mathcal{F}_t]\| \leq \gamma\|\Xi_t\|$, with $\gamma < 1$;*

    *iii)  $\text{Var}\,[L_t|\mathcal{F}_t] \leq c(1 + \|\Xi_t\|^2)$ for some $c > 0$.*

**Proof 9** *To prove the result, we show (i) - (iii) hold. Condition (i) holds by choice of learning rate. It therefore remains to prove (ii) - (iii). We first prove (ii). For this, we consider our variant of the Q-learning update rule:*

$$Q_{t+1}(s_t, I_t, a_t) = Q_t(s_t, I_t, a_t)$$
$$+ \alpha_t(s_t, I_t, a_t)\left[\max\left\{\mathcal{M}^{\pi, \pi^2}Q(s_{\tau_k}, I_{\tau_k}, a), \phi(s_{\tau_k}, a) + \gamma\max_{a'\in\mathcal{A}}Q(s', I_{\tau_k}, a')\right\} - Q_t(s_t, I_t, a_t)\right].$$

*After subtracting $Q^\star(s_t, I_t, a_t)$ from both sides and some manipulation we obtain that:*

$$\Xi_{t+1}(s_t, I_t, a_t)$$
$$= (1 - \alpha_t(s_t, I_t, a_t))\Xi_t(s_t, I_t, a_t)$$
$$+ \alpha_t(s_t, I_t, a_t))\left[\max\left\{\mathcal{M}^{\pi, \pi^2}Q(s_{\tau_k}, I_{\tau_k}, a), \phi(s_{\tau_k}, a) + \gamma\max_{a'\in\mathcal{A}}Q(s', I_{\tau_k}, a')\right\} - Q^\star(s_t, I_t, a_t)\right],$$

*where $\Xi_t(s_t, I_t, a_t) := Q_t(s_t, I_t, a_t) - Q^\star(s_t, I_t, a_t)$.*

*Let us now define by*

$$L_t(s_{\tau_k}, I_{\tau_k}, a) := \max\left\{\mathcal{M}^{\pi, \pi^2}Q(s_{\tau_k}, I_{\tau_k}, a), \phi(s_{\tau_k}, a) + \gamma\max_{a'\in\mathcal{A}}Q(s', I_{\tau_k}, a')\right\} - Q^\star(s_t, I_t, a).$$

*Then*

$$\Xi_{t+1}(s_t, I_t, a_t) = (1 - \alpha_t(s_t, I_t, a_t))\Xi_t(s_t, I_t, a_t) + \alpha_t(s_t, I_t, a_t)) \left[ L_t(s_{\tau_k}, a) \right]. \qquad (64)$$

*We now observe that*

$$\mathbb{E} \left[ L_t(s_{\tau_k}, I_{\tau_k}, a) | \mathcal{F}_t \right] = \sum_{s' \in \mathcal{S}} P(s'; a, s_{\tau_k}) \max \left\{ \mathcal{M}^{\pi, \pi^2} Q(s_{\tau_k}, I_{\tau_k}, a), \phi(s_{\tau_k}, a) + \gamma \max_{a' \in \mathcal{A}} Q(s', I_{\tau_k}, a') \right\} - Q^\star(s_{\tau_k}, a)$$
$$= T_\phi Q_t(s, I_{\tau_k}, a) - Q^\star(s, I_{\tau_k}, a). \qquad (65)$$

*Now, using the fixed point property that implies $Q^\star = T_\phi Q^\star$, we find that*

$$\mathbb{E} \left[ L_t(s_{\tau_k}, a) | \mathcal{F}_t \right] = T_\phi Q_t(s, I_{\tau_k}, a) - T_\phi Q^\star(s, I_{\tau_k}, a)$$
$$\leq \| T_\phi Q_t - T_\phi Q^\star \|$$
$$\leq \gamma \| Q_t - Q^\star \|_\infty = \gamma \| \Xi_t \|_\infty. \qquad (66)$$

*using the contraction property of $T$ established in Lemma 3. This proves (ii).*

*We now prove iii), that is*

$$\mathrm{Var} \left[ L_t | \mathcal{F}_t \right] \leq c(1 + \| \Xi_t \|^2). \qquad (67)$$

*Now by (65) we have that*

$$\mathrm{Var} \left[ L_t | \mathcal{F}_t \right] = \mathrm{Var} \left[ \max \left\{ \mathcal{M}^{\pi, \pi^2} Q(s_{\tau_k}, I_{\tau_k}, a), \phi(s_{\tau_k}, a) + \gamma \max_{a' \in \mathcal{A}} Q(s', I_{\tau_k}, a') \right\} - Q^\star(s_t, I_t, a) \right]$$

$$= \mathbb{E} \left[ \left( \max \left\{ \mathcal{M}^{\pi, \pi^2} Q(s_{\tau_k}, I_{\tau_k}, a), \phi(s_{\tau_k}, a) + \gamma \max_{a' \in \mathcal{A}} Q(s', I_{\tau_k}, a') \right\} \right. \right.$$

$$\left. \left. - Q^\star(s_t, I_t, a) - (T_\Phi Q_t(s, I_{\tau_k}, a) - Q^\star(s, I_{\tau_k}, a)) \right)^2 \right]$$

$$= \mathbb{E} \left[ \left( \max \left\{ \mathcal{M}^{\pi, \pi^2} Q(s_{\tau_k}, I_{\tau_k}, a), \phi(s_{\tau_k}, a) + \gamma \max_{a' \in \mathcal{A}} Q(s', I_{\tau_k}, a') \right\} - T_\Phi Q_t(s, I_{\tau_k}, a) \right)^2 \right]$$

$$= \mathrm{Var} \left[ \max \left\{ \mathcal{M}^{\pi, \pi^2} Q(s_{\tau_k}, I_{\tau_k}, a), \phi(s_{\tau_k}, a) + \gamma \max_{a' \in \mathcal{A}} Q(s', I_{\tau_k}, a') \right\} - T_\Phi Q_t(s, I_{\tau_k}, a))^2 \right]$$

$$\leq c(1 + \| \Xi_t \|^2),$$

*for some $c > 0$ where the last line follows due to the boundedness of $Q$ (which follows from Assumptions 2 and 4). This concludes the proof of the Theorem.*

## PROOF OF CONVERGENCE WITH LINEAR FUNCTION APPROXIMATION

First let us recall the statement of the theorem:

**Theorem 3** *ROSA converges to a limit point $r^\star$ which is the unique solution to the equation:*

$$\Pi \mathfrak{F}(\Phi r^\star) = \Phi r^\star, \qquad a.e. \qquad (68)$$

*where we recall that for any test function $\Lambda \in \mathcal{V}$, the operator $\mathfrak{F}$ is defined by $\mathfrak{F}\Lambda := \Theta + \gamma P \max\{\mathcal{M}\Lambda, \Lambda\}$.*

*Moreover, $r^\star$ satisfies the following:*

$$\| \Phi r^\star - Q^\star \| \leq c \| \Pi Q^\star - Q^\star \|. \qquad (69)$$

The theorem is proven using a set of results that we now establish. To this end, we first wish to prove the following bound:

**Lemma 4** *For any $Q \in \mathcal{V}$ we have that*

$$\|\mathfrak{F}Q - Q'\| \leq \gamma \|Q - Q'\|, \tag{70}$$

*so that the operator $\mathfrak{F}$ is a contraction.*

**Proof 10** *Recall, for any test function $\psi$, a projection operator $\Pi$ acting $\Lambda$ is defined by the following*

$$\Pi\Lambda := \underset{\bar{\Lambda} \in \{\Phi r | r \in \mathbb{R}^p\}}{\arg\min} \|\bar{\Lambda} - \Lambda\|.$$

*Now, we first note that in the proof of Lemma 3, we deduced that for any $\Lambda \in L_2$ we have that*

$$\left\| \mathcal{M}\Lambda - \left[ \psi(\cdot, a) + \gamma \max_{a \in \mathcal{A}} \mathcal{P}^a \Lambda' \right] \right\| \leq \gamma \|\Lambda - \Lambda'\|,$$

*(c.f. Lemma 3).*

*Setting $\Lambda = Q$ and $\psi = \Theta$, it can be straightforwardly deduced that for any $Q, \hat{Q} \in L_2$: $\left\| \mathcal{M}Q - \hat{Q} \right\| \leq \gamma \left\| Q - \hat{Q} \right\|$. Hence, using the contraction property of $\mathcal{M}$, we readily deduce the following bound:*

$$\max \left\{ \left\| \mathcal{M}Q - \hat{Q} \right\|, \left\| \mathcal{M}Q - \mathcal{M}\hat{Q} \right\| \right\} \leq \gamma \left\| Q - \hat{Q} \right\|, \tag{71}$$

*We now observe that $\mathfrak{F}$ is a contraction. Indeed, since for any $Q, Q' \in L_2$ we have that:*

$$\begin{aligned}
\|\mathfrak{F}Q - \mathfrak{F}Q'\| &= \|\Theta + \gamma P \max\{\mathcal{M}Q, Q\} - (\Theta + \gamma P \max\{\mathcal{M}Q', Q'\})\| \\
&= \gamma \|P \max\{\mathcal{M}Q, Q\} - P \max\{\mathcal{M}Q', Q'\}\| \\
&\leq \gamma \|\max\{\mathcal{M}Q, Q\} - \max\{\mathcal{M}Q', Q'\}\| \\
&\leq \gamma \|\max\{\mathcal{M}Q - \mathcal{M}Q', Q - \mathcal{M}Q', \mathcal{M}Q - Q', Q - Q'\}\| \\
&\leq \gamma \max\{\|\mathcal{M}Q - \mathcal{M}Q'\|, \|Q - \mathcal{M}Q'\|, \|\mathcal{M}Q - Q'\|, \|Q - Q'\|\} \\
&= \gamma \|Q - Q'\|,
\end{aligned}$$

*using (71) and again using the non-expansiveness of $P$.*

We next show that the following two bounds hold:

**Lemma 5** *For any $Q \in \mathcal{V}$ we have that*

    *i)*             $\left\| \Pi\mathfrak{F}Q - \Pi\mathfrak{F}\bar{Q} \right\| \leq \gamma \left\| Q - \bar{Q} \right\|,$

    *ii)*            $\|\Phi r^\star - Q^\star\| \leq \frac{1}{\sqrt{1-\gamma^2}} \|\Pi Q^\star - Q^\star\|.$

**Proof 11** *The first result is straightforward since as $\Pi$ is a projection it is non-expansive and hence:*

$$\left\| \Pi\mathfrak{F}Q - \Pi\mathfrak{F}\bar{Q} \right\| \leq \left\| \mathfrak{F}Q - \mathfrak{F}\bar{Q} \right\| \leq \gamma \left\| Q - \bar{Q} \right\|,$$

*using the contraction property of $\mathfrak{F}$. This proves i). For ii), we note that by the orthogonality property of projections we have that $\langle \Phi r^\star - \Pi Q^\star, \Phi r^\star - \Pi Q^\star \rangle$, hence we observe that:*

$$\begin{aligned}
\|\Phi r^\star - Q^\star\|^2 &= \|\Phi r^\star - \Pi Q^\star\|^2 + \|\Phi r^\star - \Pi Q^\star\|^2 \\
&= \|\Pi\mathfrak{F}\Phi r^\star - \Pi Q^\star\|^2 + \|\Phi r^\star - \Pi Q^\star\|^2 \\
&\leq \|\mathfrak{F}\Phi r^\star - Q^\star\|^2 + \|\Phi r^\star - \Pi Q^\star\|^2 \\
&= \|\mathfrak{F}\Phi r^\star - \mathfrak{F}Q^\star\|^2 + \|\Phi r^\star - \Pi Q^\star\|^2 \\
&\leq \gamma^2 \|\Phi r^\star - Q^\star\|^2 + \|\Phi r^\star - \Pi Q^\star\|^2,
\end{aligned}$$

*after which we readily deduce the desired result.*

**Lemma 6** *Define the operator $H$ by the following:* $HQ(z) = \begin{cases} \mathcal{M}Q(z), & \text{if } \mathcal{M}Q(z) > \Phi r^\star, \\ Q(z), & \text{otherwise}, \end{cases}$

*and $\tilde{\mathfrak{F}}$ by:* $\tilde{\mathfrak{F}}Q := \Theta + \gamma PHQ$.

*For any $Q, \bar{Q} \in L_2$ we have that*

$$\left\| \tilde{\mathfrak{F}}Q - \tilde{\mathfrak{F}}\bar{Q} \right\| \le \gamma \left\| Q - \bar{Q} \right\| \tag{72}$$

*and hence $\tilde{\mathfrak{F}}$ is a contraction mapping.*

**Proof 12** *Using (71), we now observe that*

$$\begin{aligned}
\left\| \tilde{\mathfrak{F}}Q - \tilde{\mathfrak{F}}\bar{Q} \right\| &= \left\| \Theta + \gamma PHQ - (\Theta + \gamma PH\bar{Q}) \right\| \\
&\le \gamma \left\| HQ - H\bar{Q} \right\| \\
&\le \gamma \left\| \max\{\mathcal{M}Q - \mathcal{M}\bar{Q}, Q - \bar{Q}, \mathcal{M}Q - \bar{Q}, \mathcal{M}\bar{Q} - Q\} \right\| \\
&\le \gamma \max\{\left\| \mathcal{M}Q - \mathcal{M}\bar{Q} \right\|, \left\| Q - \bar{Q} \right\|, \left\| \mathcal{M}Q - \bar{Q} \right\|, \left\| \mathcal{M}\bar{Q} - Q \right\|\} \\
&\le \gamma \max\{\gamma \left\| Q - \bar{Q} \right\|, \left\| Q - \bar{Q} \right\|, \left\| \mathcal{M}Q - \bar{Q} \right\|, \left\| \mathcal{M}\bar{Q} - Q \right\|\} \\
&= \gamma \left\| Q - \bar{Q} \right\|,
\end{aligned}$$

*again using the non-expansive property of $P$.*

**Lemma 7** *Define by $\tilde{Q} := \Theta + \gamma P v^{\tilde{\pi}}$ where*

$$v^{\tilde{\pi}}(z) := \Theta(s_{\tau_k}, a) + \gamma \max_{a \in \mathcal{A}} \sum_{s' \in \mathcal{S}} P(s'; a, s_{\tau_k}) \Phi r^\star(s', I(\tau_k)), \tag{73}$$

*then $\tilde{Q}$ is a fixed point of $\tilde{\mathfrak{F}}\tilde{Q}$, that is $\tilde{\mathfrak{F}}\tilde{Q} = \tilde{Q}$.*

**Proof 13** *We begin by observing that*

$$\begin{aligned}
H\tilde{Q}(z) &= H\left(\Theta(z) + \gamma P v^{\tilde{\pi}}\right) \\
&= \begin{cases} \mathcal{M}Q(z), & \text{if } \mathcal{M}Q(z) > \Phi r^\star, \\ Q(z), & \text{otherwise}, \end{cases} \\
&= \begin{cases} \mathcal{M}Q(z), & \text{if } \mathcal{M}Q(z) > \Phi r^\star, \\ \Theta(z) + \gamma P v^{\tilde{\pi}}, & \text{otherwise}, \end{cases} \\
&= v^{\tilde{\pi}}(z).
\end{aligned}$$

*Hence,*

$$\tilde{\mathfrak{F}}\tilde{Q} = \Theta + \gamma PH\tilde{Q} = \Theta + \gamma P v^{\tilde{\pi}} = \tilde{Q}. \tag{74}$$

*which proves the result.*

**Lemma 8** *The following bound holds:*

$$\mathbb{E}\left[v^{\hat{\pi}}(z_0)\right] - \mathbb{E}\left[v^{\tilde{\pi}}(z_0)\right] \le 2\left[(1-\gamma)\sqrt{(1-\gamma^2)}\right]^{-1} \left\| \Pi Q^\star - Q^\star \right\|. \tag{75}$$

**Proof 14** *By definitions of $v^{\hat{\pi}}$ and $v^{\tilde{\pi}}$ (c.f (73)) and using Jensen's inequality and the stationarity property we have that,*

$$\begin{aligned}
\mathbb{E}\left[v^{\hat{\pi}}(z_0)\right] - \mathbb{E}\left[v^{\tilde{\pi}}(z_0)\right] &= \mathbb{E}\left[P v^{\hat{\pi}}(z_0)\right] - \mathbb{E}\left[P v^{\tilde{\pi}}(z_0)\right] \\
&\le \left| \mathbb{E}\left[P v^{\hat{\pi}}(z_0)\right] - \mathbb{E}\left[P v^{\tilde{\pi}}(z_0)\right] \right| \\
&\le \left\| P v^{\hat{\pi}} - P v^{\tilde{\pi}} \right\|. \tag{76}
\end{aligned}$$

*Now recall that $\tilde{Q} := \Theta + \gamma P v^{\tilde{\pi}}$ and $Q^\star := \Theta + \gamma P v^{\pi^\star}$, using these expressions in (76) we find that*

$$\mathbb{E}\left[v^{\hat{\pi}}(z_0)\right] - \mathbb{E}\left[v^{\tilde{\pi}}(z_0)\right] \le \frac{1}{\gamma}\left\| \tilde{Q} - Q^\star \right\|.$$

*Moreover, by the triangle inequality and using the fact that $\mathfrak{F}(\Phi r^\star) = \tilde{\mathfrak{F}}(\Phi r^\star)$ and that $\mathfrak{F}Q^\star = Q^\star$ and $\mathfrak{F}\tilde{Q} = \tilde{Q}$ (c.f. (75)) we have that*

$$\left\| \tilde{Q} - Q^\star \right\| \leq \left\| \tilde{Q} - \mathfrak{F}(\Phi r^\star) \right\| + \left\| Q^\star - \tilde{\mathfrak{F}}(\Phi r^\star) \right\|$$
$$\leq \gamma \left\| \tilde{Q} - \Phi r^\star \right\| + \gamma \left\| Q^\star - \Phi r^\star \right\|$$
$$\leq 2\gamma \left\| \tilde{Q} - \Phi r^\star \right\| + \gamma \left\| Q^\star - \tilde{Q} \right\|,$$

*which gives the following bound:*

$$\left\| \tilde{Q} - Q^\star \right\| \leq 2 \left( 1 - \gamma \right)^{-1} \left\| \tilde{Q} - \Phi r^\star \right\|,$$

*from which, using Lemma 5, we deduce that* $\left\| \tilde{Q} - Q^\star \right\| \leq 2 \left[ (1-\gamma)\sqrt{(1-\gamma^2)} \right]^{-1} \left\| \tilde{Q} - \Phi r^\star \right\|$, *after which by (77), we finally obtain*

$$\mathbb{E} \left[ v^{\hat{\pi}}(z_0) \right] - \mathbb{E} \left[ v^{\tilde{\pi}}(z_0) \right] \leq 2 \left[ (1-\gamma)\sqrt{(1-\gamma^2)} \right]^{-1} \left\| \tilde{Q} - \Phi r^\star \right\|,$$

*as required.*

Let us rewrite the update in the following way:

$$r_{t+1} = r_t + \gamma_t \Xi(w_t, r_t),$$

where the function $\Xi : \mathbb{R}^{2d} \times \mathbb{R}^p \to \mathbb{R}^p$ is given by:

$$\Xi(w, r) := \phi(z) \left( \Theta(z) + \gamma \max \left\{ (\Phi r)(z'), \mathcal{M}(\Phi r)(z') \right\} - (\Phi r)(z) \right),$$

for any $w \equiv (z, z') \in (\mathbb{N} \times \mathcal{S})^2$ where $z = (t, s) \in \mathbb{N} \times \mathcal{S}$ and $z' = (t, s') \in \mathbb{N} \times \mathcal{S}$ and for any $r \in \mathbb{R}^p$. Let us also define the function $\Xi : \mathbb{R}^p \to \mathbb{R}^p$ by the following:

$$\Xi(r) := \mathbb{E}_{w_0 \sim (\mathbb{P}, \mathbb{P})} \left[ \Xi(w_0, r) \right]; w_0 := (z_0, z_1).$$

**Lemma 9** *The following statements hold for all $z \in \{0, 1\} \times \mathcal{S}$:*

    *i)* $(r - r^\star)\Xi_k(r) < 0, \qquad \forall r \neq r^\star,$

    *ii)* $\Xi_k(r^\star) = 0.$

**Proof 15** *To prove the statement, we first note that each component of $\Xi_k(r)$ admits a representation as an inner product, indeed:*

$$\Xi_k(r) = \mathbb{E} \left[ \phi_k(z_0)(\Theta(z_0) + \gamma \max \left\{ \Phi r(z_1), \mathcal{M}\Phi(z_1) \right\} - (\Phi r)(z_0) \right]$$
$$= \mathbb{E} \left[ \phi_k(z_0)(\Theta(z_0) + \gamma \mathbb{E} \left[ \max \left\{ \Phi r(z_1), \mathcal{M}\Phi(z_1) \right\} | z_0 \right] - (\Phi r)(z_0) \right]$$
$$= \mathbb{E} \left[ \phi_k(z_0)(\Theta(z_0) + \gamma P \max \left\{ (\Phi r, \mathcal{M}\Phi) \right\} (z_0) - (\Phi r)(z_0) \right]$$
$$= \left\langle \phi_k, \mathfrak{F}\Phi r - \Phi r \right\rangle,$$

*using the iterated law of expectations and the definitions of $P$ and $\mathfrak{F}$.*

*We now are in position to prove i). Indeed, we now observe the following:*

$$(r - r^\star) \Xi_k(r) = \sum_{l=1} (r(l) - r^\star(l)) \left\langle \phi_l, \mathfrak{F}\Phi r - \Phi r \right\rangle$$
$$= \left\langle \Phi r - \Phi r^\star, \mathfrak{F}\Phi r - \Phi r \right\rangle$$
$$= \left\langle \Phi r - \Phi r^\star, (\mathbf{1} - \Pi)\mathfrak{F}\Phi r + \Pi \mathfrak{F}\Phi r - \Phi r \right\rangle$$
$$= \left\langle \Phi r - \Phi r^\star, \Pi \mathfrak{F}\Phi r - \Phi r \right\rangle,$$

*where in the last step we used the orthogonality of $(\mathbf{1} - \Pi)$. We now recall that $\Pi \mathfrak{F}\Phi r^\star = \Phi r^\star$ since $\Phi r^\star$ is a fixed point of $\Pi \mathfrak{F}$. Additionally, using Lemma 5 we observe that $\| \Pi \mathfrak{F}\Phi r - \Phi r^\star \| \leq \gamma \| \Phi r - \Phi r^\star \|$. With this we now find that*

$$\left\langle \Phi r - \Phi r^\star, \Pi \mathfrak{F}\Phi r - \Phi r \right\rangle$$

$$
\begin{aligned}
&= \langle \Phi r - \Phi r^\star, (\Pi \mathfrak{F} \Phi r - \Phi r^\star) + \Phi r^\star - \Phi r \rangle \\
&\le \| \Phi r - \Phi r^\star \| \, \| \Pi \mathfrak{F} \Phi r - \Phi r^\star \| - \| \Phi r^\star - \Phi r \|^2 \\
&\le (\gamma - 1) \, \| \Phi r^\star - \Phi r \|^2,
\end{aligned}
$$

*which is negative since $\gamma < 1$ which completes the proof of part i).*

*The proof of part ii) is straightforward since we readily observe that*

$$
\Xi_k(r^\star) = \langle \phi_l, \mathfrak{F} \Phi r^\star - \Phi r \rangle = \langle \phi_l, \Pi \mathfrak{F} \Phi r^\star - \Phi r \rangle = 0,
$$

*as required and from which we deduce the result.*

To prove the theorem, we make use of a special case of the following result:

**Theorem 4 (Th. 17, p. 239 in [3])** *Consider a stochastic process $r_t : \mathbb{R} \times \{\infty\} \times \Omega \to \mathbb{R}^k$ which takes an initial value $r_0$ and evolves according to the following:*

$$
r_{t+1} = r_t + \alpha \Xi(s_t, r_t), \tag{77}
$$

*for some function $s : \mathbb{R}^{2d} \times \mathbb{R}^k \to \mathbb{R}^k$ and where the following statements hold:*

1. *$\{s_t | t = 0, 1, \ldots\}$ is a stationary, ergodic Markov process taking values in $\mathbb{R}^{2d}$*

2. *For any positive scalar $q$, there exists a scalar $\mu_q$ such that $\mathbb{E}\left[1 + \|s_t\|^q | s \equiv s_0\right] \le \mu_q \left(1 + \|s\|^q\right)$*

3. *The step size sequence satisfies the Robbins-Monro conditions, that is $\sum_{t=0}^\infty \alpha_t = \infty$ and $\sum_{t=0}^\infty \alpha_t^2 < \infty$*

4. *There exists scalars $c$ and $q$ such that $\|\Xi(w, r)\| \le c \left(1 + \|w\|^q\right) \left(1 + \|r\|\right)$*

5. *There exists scalars $c$ and $q$ such that $\sum_{t=0}^\infty \|\mathbb{E}\left[\Xi(w_t, r) | z_0 \equiv z\right] - \mathbb{E}\left[\Xi(w_0, r)\right]\| \le c \left(1 + \|w\|^q\right) \left(1 + \|r\|\right)$*

6. *There exists a scalar $c > 0$ such that $\|\mathbb{E}[\Xi(w_0, r)] - \mathbb{E}[\Xi(w_0, \bar{r})]\| \le c \|r - \bar{r}\|$*

7. *There exists scalars $c > 0$ and $q > 0$ such that $\sum_{t=0}^\infty \|\mathbb{E}\left[\Xi(w_t, r) | w_0 \equiv w\right] - \mathbb{E}\left[\Xi(w_0, \bar{r})\right]\| \le c \|r - \bar{r}\| \left(1 + \|w\|^q\right)$*

8. *There exists some $r^\star \in \mathbb{R}^k$ such that $\Xi(r)(r - r^\star) < 0$ for all $r \ne r^\star$ and $\bar{s}(r^\star) = 0$.*

*Then $r_t$ converges to $r^\star$ almost surely.*

In order to apply the Theorem 4, we show that conditions 1 - 7 are satisfied.

**Proof 16** *Conditions 1-2 are true by assumption while condition 3 can be made true by choice of the learning rates. Therefore it remains to verify conditions 4-7 are met.*

*To prove 4, we observe that*

$$
\begin{aligned}
\|\Xi(w, r)\| &= \|\phi(z) \left(\Theta(z) + \gamma \max \{(\Phi r)(z'), \mathcal{M}\Phi(z')\} - (\Phi r)(z)\right)\| \\
&\le \|\phi(z)\| \, \|\Theta(z) + \gamma \left(\|\phi(z')\| \, \|r\| + \mathcal{M}\Phi(z')\right)\| + \|\phi(z)\| \, \|r\| \\
&\le \|\phi(z)\| \left(\|\Theta(z)\| + \gamma \|\mathcal{M}\Phi(z')\|\right) + \|\phi(z)\| \left(\gamma \|\phi(z')\| + \|\phi(z)\|\right) \|r\|.
\end{aligned}
$$

*Now using the definition of $\mathcal{M}$, we readily observe that $\|\mathcal{M}\Phi(z')\| \le \|\Theta\| + \gamma \|\mathcal{P}^\pi_{s' s_t} \Phi\| \le \|\Theta\| + \gamma \|\Phi\|$ using the non-expansiveness of $P$.*

*Hence, we lastly deduce that*

$$
\begin{aligned}
\|\Xi(w, r)\| &\le \|\phi(z)\| \left(\|\Theta(z)\| + \gamma \|\mathcal{M}\Phi(z')\|\right) + \|\phi(z)\| \left(\gamma \|\phi(z')\| + \|\phi(z)\|\right) \|r\| \\
&\le \|\phi(z)\| \left(\|\Theta(z)\| + \gamma \|\Theta\| + \gamma \|\psi\|\right) + \|\phi(z)\| \left(\gamma \|\phi(z')\| + \|\phi(z)\|\right) \|r\|,
\end{aligned}
$$

*we then easily deduce the result using the boundedness of $\phi, \Theta$ and $\psi$.*

*Now we observe the following Lipschitz condition on $\Xi$:*

$\|\Xi(w, r) - \Xi(w, \bar{r})\|$

$= \|\phi(z) \left(\gamma \max \left\{(\Phi r)(z'), \mathcal{M}\Phi(z')\right\} - \gamma \max \left\{(\Phi\bar{r})(z'), \mathcal{M}\Phi(z')\right\}\right) - ((\Phi r)(z) - \Phi\bar{r}(z))\|$

$\leq \gamma \|\phi(z)\| \|\max \left\{\phi'(z')r, \mathcal{M}\Phi'(z')\right\} - \max \left\{(\phi'(z')\bar{r}), \mathcal{M}\Phi'(z')\right\}\| + \|\phi(z)\| \|\phi'(z)r - \phi(z)\bar{r}\|$

$\leq \gamma \|\phi(z)\| \|\phi'(z')r - \phi'(z')\bar{r}\| + \|\phi(z)\| \|\phi'(z)r - \phi(z)\bar{r}\|$

$\leq \|\phi(z)\| \left(\|\phi(z)\| + \gamma \|\phi(z)\| \|\phi'(z')\|\right) \|r - \bar{r}\|$

$\leq c \|r - \bar{r}\|,$

*using Cauchy-Schwarz inequality and that for any scalars $a, b, c$ we have that $|\max\{a, b\} - \max\{b, c\}| \leq |a - c|$.*

*Using Assumptions 3 and 4, we therefore deduce that*

$$\sum_{t=0}^{\infty} \|\mathbb{E}\left[\Xi(w, r) - \Xi(w, \bar{r})|w_0 = w\right] - \mathbb{E}\left[\Xi(w_0, r) - \Xi(w_0, \bar{r})\right]\| \leq c \|r - \bar{r}\| \left(1 + \|w\|^l\right). \quad (78)$$

*Part 2 is assured by Lemma 5 while Part 4 is assured by Lemma 8 and lastly Part 8 is assured by Lemma 9.*

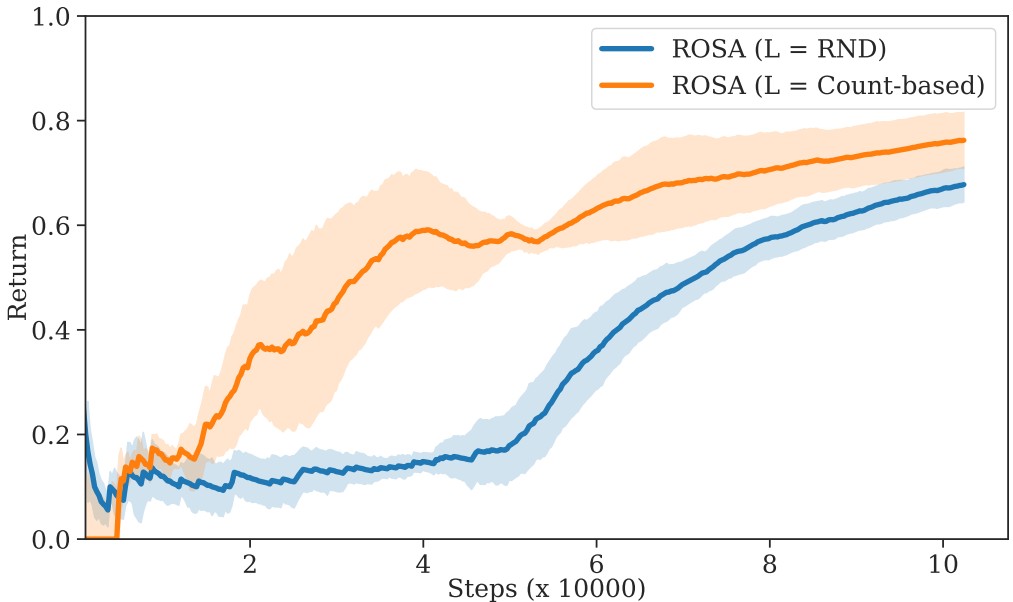

Figure 8: ROSA is robust to the component used to generate exploration bonus $L$. ROSA works equally well when we RND or Count-based method for $L$.

# Rebuttal

## 17  ADDITIONAL EXPERIMENT 1 - REPLACING RND FOR BONUS REWARD

In this experiment we sought to ascertain if ROSA is robust to the component used to generate exploration bonus $L$. We ran two versions of ROSA, one where $L$ is computed using RND and one where $L$ is computed using a simple count-based measure $\frac{1}{\text{Count}(s)}$ where $s \in \mathcal{S}$ (the function Count$(s)$ simply tallies the number of times state $s$ has been visited). Figure 8 shows performance of these two versions of ROSA on the Maze environment shown in Figure 1. ROSA performs equally well with both components, indicating that it is not dependent on a fine-tuned exploration bonus. The additional machinery of switching controls and choices of intrinsic rewards to add mean that ROSA can work with basic exploration bonuses. Note that we generally use RND since it is simple to implement and works equally well on discrete and continuous state spaces.

## 18  ADDITIONAL EXPERIMENT 2 - ROBUSTNESS TO INITIALISATION OF $\phi$

Some reviewers raised important questions about the requirement of a carefully initialised $phi$ function. Here, we show performance of individual runs of ROSA on the Maze shown in Figure 1. In each run, $\phi$ is randomly initialised using Xavier initialisation, and as a consequence ROSA works with a different $\phi$ function in each run. As can be seen in Figure 9, despite the randomness in $\phi$ the performance of ROSA does not vary significantly over the runs. This suggests that ROSA can adapt to the $\phi$ function it is presented with, and still come up with good reward shaping.

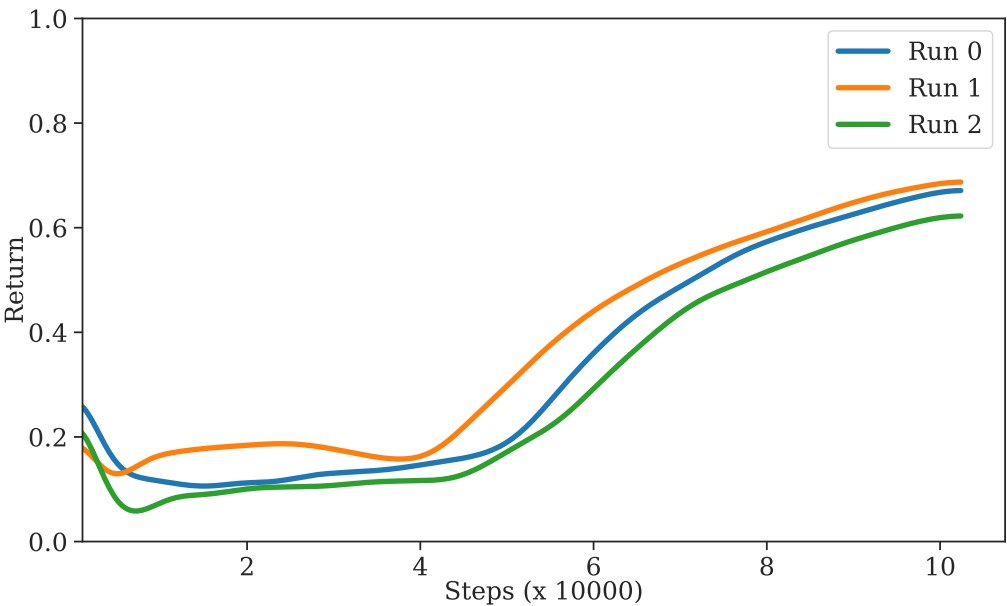

Figure 9: ROSA is robust to the component used to generate exploration bonus $L$. ROSA works equally well when we RND or Count-based method for $L$.

