# OpenReview forum: "Learning to Shape Rewards using a Game of Two Partners"
_ICLR.cc/2022/Conference — ICLR 2022 Submitted_

### Official Review · Reviewer_XHiG · 2021-10-31

**Correctness:** 2
**Technical Novelty And Significance:** 3
**Empirical Novelty And Significance:** 2
**Recommendation:** 5
**Confidence:** 4

**Main Review:**

Strengths
------------
-Framing reward shaping within the Markov game (MG) framework is novel

-Some theoretical guarantees are provided, in particular Proposition 1

Weaknesses
------------------
-The presentation and clarity of the paper is lacking

-Many low level choices are brushed upon and hardly motivated

-Empirical improvements are quite limited and the code is not provided


Details
---------
Presenting a solution to the potential function in PBRS is an important contribution as it is still to this day an open problem. Moreover, most of the research on PBRS has not used neural networks and therefore can be seen as limited. As such, framing the method as a Markov game is an interesting avenue. That being said, the paper does a poor job at presenting clearly the ideas, motivations, limitations as well as some related work. For example, the idea of introducing switching controls can by itself be an interesting idea, but is completely orthogonal to framing the method as a two player MG. Yet the method that the authors present ties the two concepts together. Around the idea of switching is the claim that since the method only learns to shape rewards at relevant states, it is more computational efficient. First, is the proposed method, in practice, really learning what are the relevant states? Second, wouldn't be the computational cost be equivalent if one first has to learn on what states to perform reward shaping? I would argue that the computational cost even greater since both elements need to work together (reward shaping and switching controls), unless a more trivial solution is proposed, as in the paper. Given these considerations, I would be much more careful about the kind of claims being made.

The idea of two player MG can be an interesting avenue, but I am wondering if the fact that only the Controller affects the dynamics, is it really relevant to talk a two player game? In the end, the Controller is a network that is simply using reward shaping, while the Shaper is separate network that learns how to shape the rewards. What does framing the setup as a two player game bring in terms of insights/methods/solutions? This kind of questions is not really addressed in the paper.

The presentation of the method has a structure that doesn't help clarity where many of the details are only later presented and are not really motivated. The presentation itself is interleaved with claims that hinders clarity and bring more questions, such as first paragraph of 4.2 where claims about the effect of the switching cost are made. The idea of termination times is presented quickly but most of the details are in the appendix, which again hinders a nice and clean presentation. The final algorithm makes choices such as adding an exploration bonus to the Shaper or using a fixed network f(s) for the potential function. Especially with respect to the latter point, why would it make any sense to have a random network output the value for reward shaping? The algorithm also mentions that $a^2_{\tau_k} =0 \forall k$, but then it also mentions that when $g(s_t)=1$ we sample from the Shaper's policy? Why are such choices being made? Is it to make sure that the theoretical derivation work out, such as in Proposition 1? How does it help in practice?

Finally, related work is missing.The paper claims that "In RS the question of which φ to insert has not been addressed." This is simply false. There is recent and relevant work on this such as (Klissarov et al. 2020), as well as more classic work such as (Brys et al., 2015, Grzes and Kudenko 2010). These references should be included in the paper given their close conenction and the claims adjusted. Moreover, there is many landmark papers on curiosity that are missing, such as  (Schmidhuber 2010,  Oudeyer et al. 2007)

J. Schmidhuber. Formal theory of creativity, fun, and intrinsic motivation (1990–2010)

Brys et al. 2015, Reinforcement learning from demonstration through shaping

Grzes and Kudenko 2010,  Online learning of shaping rewards in reinforcement learning

Klissarov et al. 2020, Reward propagation using graph convolutional networks

Oudeyer et al. 2007, What is intrinsic motivation? A typology of computational approaches



**Summary Of The Paper:**

The paper proposes to frame the reward shaping method as a Markov game between two players. In this setting the first player is learning to act in the environment while the second player is learning to provide reward shaping to the first player. Additionally, the authors propose to use a switching scheme that indicates whether or not to perform reward shaping for the first player. The authors provide some theoretical guarantees under a large set of assumptions. The authors perform experiments on a set of toy tasks and a few more complex domains.

**Summary Of The Review:**

The paper proposes an original way to frame reward shaping. However, the presentation is lacking clarity and many of the choices/details are not well motivated. Despite my negative review I think there is potential in the approach, but I believe that a major rewriting is necessary as well as more qualitative empirical analysis. The quantitive empirical results are not especially favourable either.

---

> ### Author Response · Authors · 2021-11-25
> **Authors' response**
>
> We thank the reviewer for their careful reading and valuable comments. We respond to your points individually.
>
> **Improvements to empirical evaluation**
>
> Thanks for the valuable suggestion. Further to the reviewer's suggestion, we have now added a detailed investigation that confirms ROSA performs well without the RND term (and in fact entirely comparably with a very simple count-based bonus instead of RND) in Shaper's objective - please see Sec 13. We also demonstrated that ROSA is robust to choices of the neural network weights for constructing $\phi$.
>
> **“Is the proposed method really learning what are the relevant states?”**
>
> **A.** We refer the reviewer to the visualisation in Exp. 1 which shows exactly where the Shaper adds intrinsic rewards. There it is shown that the Shaper adds intrinsic rewards at precisely (and only) the set of states that guide the controller to the optimal goal state (and away from the suboptimal goal state). This is formally asserted in our theoretical result in Prop. 2.
>
> **“Wouldn't the computational cost be equivalent if one first has to learn on what states to perform reward shaping?... I would argue that the computational cost even greater“**
>
> **A.**  Key to ROSA is the switching control framework. It reduces the set of states that the Shaper learns the shaping rewards - note that decision space for determining to add shaping rewards is binary $\\{0,1\\}$. This means that the Shaper must learn to add the best shaping reward only at a subset of states while discarding all others. In our experiments, we chose the size of the action set for Shaper to be a singleton which means that the decision space for Shaper is $|\mathcal{S}|\times \\{0,1\\}$. This is what produces the computational efficiency.
>
> **“Only the Controller affects the dynamics - is it a two player game?”**
>
> **A.**  In this system, the two players strategically interact over multiple rounds. In game theory, these systems are known as single-controller stochastic games [1]. The naming convention represents the fact that for any agent, its rewards are affected by the actions of the other agent, leading to a strategic interaction.
>
> **”What does framing the setup as a two player game bring?”**
>
> A.** We refer the reviewer to paragraph 2 on Pg 2. Here we discuss the fact that the two-player game allows the Shaper to produce shaping rewards adaptively in response to the behaviour of the controller. With this, we prove convergence of our framework which is fairly rare for systems with more than one adaptive learner. Lastly, since the shaper has a separate objective, it is able to generate rewards that deviate significantly from the environment rewards to enable the controller to learn complex exploration patterns.
>
> **“Why would it make any sense to have a random network output the value for reward shaping?”**
>
> **A.** We resolve this misconception. As is described in Sec. 4.3, the output of the value for the reward shaping is determined by the actions of the shaper and a state dependent function which is fixed neural network. The key point is that this merely defines a map from $\mathcal{S}\to\mathbb{R}^m$. To construct this map, we need to choose the fixed weights of the neural network. To do this, we chose to sample these weights which are then fixed. We have added an Ablation study in Sec. 14 which shows our method is not sensitive to the choice of these weights.
>
> **“The algorithm mentions that $a^2_{\tau_k}=0$ but also mentions that when $g(s_t)=1$ we sample from the Shaper's policy?"**
>
> **A.** We explain this core aspect of the framework. The policy $g$ is used by the Shaper decides whether or not to add a shaping reward at a given state. If $g(s_t)=0$ the Shaper does not add a shaping reward. If $g(s_t)=1$ it adds a shaping reward,  the magnitude of which is determined by two objects: the Shaper’s action which is sampled from the shaper’s policy $\pi^2$ (only sampled when $g(s_t)=1$) and the state dependent map $f(s)$. Fixing $a^2_{\tau_k}=0$ ensures that the maximum total expected return is not altered by the shaper as stated  in Prop. 1. We will add this sentence in the updated version.
>
> **Missing references and “In RS the question of which φ to insert has not been addressed."**
>
> **A.** We thank the reviewer for referring us to those papers – indeed they are relevant. We will include them in our updated version. We agree with the reviewer about the claim – we intended to convey the point about the question of learning $\phi$ autonomously which remains a challenge. We have revised this sentence accordingly.
>
> **Readability**
>
> **A.** Thanks for the comments, we have made various adjustments to the structure to improve the flow and readability as per the reviewer's comments. We have also given the paper a thorough polish to eliminate small typos.
>
> [1] Raghavan, T. E. S.,  Z. Syed. "Computing stationary Nash equilibria of undiscounted single-controller stochastic games." Mathematics of Operations Research (2002)

---

> > ### Comment · Reviewer_XHiG · 2021-11-25
> > **Response**
> >
> > I would like to thank the authors for taking the time to answer and perform additional experiments. Given these clarifications I would raise my score to a 5.
> >
> > I also appreciate the authors acknowledging the importance of the suggested related work, I believe it will set a better context for the current contribution.
> >
> > At the moment I can hardly raise my score higher as issues still remain. Unfortunately I doubt enough time remains to address these, and as such I would have really appreciated if the authors had answered earlier (I understand it is not always possible). That being said I am now listing three major issues.
> >
> > From a performance point of view, I can agree that adding a switching action can help, as it can be seen as a form of attention over states, which can eventually help the Shaper focus on the right parts of the state space. That being said, as the switching control can change throughout learning, it is very unlikely that there would be computational gains, as the Shaper would learn to shape in many states that in the end might not be useful. The only case where computational gains could really happen is if some kind of prior knowledge is used to decide the switching control. Therefore I would be careful with such a claim. I believe a ablation analysis where the switching controller is not added would clearly show the benefits of it.
> >
> > My second concern is that a fixed neural network is used to map from S to $\mathb{R}^m$. Although $i(a^2)$ will choose which entry to take from the output of the fixed neural network, this still severely limits the expressiveness of the shaping function. I am really curious as to why such a choice has been made, as it seems to almost go against the basic logic of learning of to shape. No explanation currently appear in the paper.
> >
> > Finally, no information about how many runs have been used appear in the main paper. There is also no provided code. I think it is our duty to contribute to the reproducibility of this field, which is clearly stated in the Authors' guideline of ICLR.

---

> > > ### Author Response · Authors · 2021-11-25
> > > **Authors' response**
> > >
> > > We thank the reviewer for taking the time to read our response.
> > >
> > > We would like to respond to the set of points last raised.
> > >
> > > While the reviewer is correct that the shaper must learn which states to active the shaping reward, as we alluded to earlier, our reasoning behind this claim is that the decision space for the switching policy is $|\mathcal{S}|\times\\{0,1\\}$ i.e at each state it makes a binary decision. Consequently, this component of the learning process is much quicker than the controller's policy which must optimise over a decision space which is $|\mathcal{S}||\mathcal{A}|$ (choosing an action from its action space at every state). This results in the shaper learning its policy rapidly (relative to the controller) which guides the controller agent towards its optimal policy during its learning phase.
> > >
> > > We produced heatmap visualisations of the shaping rewards over iterations (we included the terminal heatmap in Fig. 1). These heatmaps show that the shaper quickly converges to the heatmap depicted in Fig. 1. We were reluctant to add this given the length of the paper, we however agree with the reviewer that it is a useful analysis and are happy to include this in the supplementary material.
> > >
> > > In terms of the expressiveness of the shaping function, the root of the reviewer's concern is likely due to the idea that factorising the shaping reward into a state based function part and a function of the shaper's action part prevents the shaping function as a whole from allowing interactions between the shaper's action and the state. However, in our case this factorisation is not restrictive since the action is sampled from a policy which is already contingent on the state.
> > >
> > > In terms of the code and runs, thanks for pointing that out. We can easily add these details to the paper and are happy to commit to doing so.

---

### Official Review · Reviewer_uREp · 2021-10-31

**Correctness:** 3
**Technical Novelty And Significance:** 3
**Empirical Novelty And Significance:** 3
**Recommendation:** 5
**Confidence:** 3

**Main Review:**

# Strengths

This paper proposes an interesting and (as far as I know) novel approach to learning how to shape rewards. The idea of training an agent to output actions that modulate a potential-based reward is quite interesting, since this Shaper agent can use heuristic metrics (e.g. the RND bonus) to guide exploration while preserving the Controller agent’s final performance thanks to the extended policy invariance results. As the paper notes, this multi-agent approach could lead to interesting new solutions for RL. The experiments in the paper compare against appropriate baselines and demonstrate promising results. The visualizations are useful, and overall the paper is easy to follow.

# Weaknesses

One concern with the paper is that it is the combination of many heuristic components (e.g. how L is chosen, how c is chosen, how I is chosen, and how phi is chosen). This runs a bit counter to the author’s motivation to avoid manually engineering components. I understand that part of the paper is demonstrating that despite these heuristics, the Controller policy is still optimizing the correct objective, but these additional design choices may result in a method that is overall rather brittle and sensitive to the design choices, even if the Controller is *theoretically* optimizing the right objective. Demonstrating that the method still performs well when L, c, I, and/or phi is altered would provide evidence for the generality of this approach.

The other main weaknesses of the paper come in the writing and experiments, which I detail below.

## Writing:
One major concern is that the main paper does not describe how phi is chosen. This decision is critical to the success of the paper, and I could not find much in the paper other than a hypothetical description (“φ can be, for example, a neural network with fixed weights with input (s, a2)”).

The authors claim that there are two advantages over past work: (1) learning the reward shaping term concurrently as opposed to iteratively and (2) avoiding the computational cost of computing the reward at every state. The second concern seems rather minor. For the first claim, the authors do not provide evidence that, “ROSA performs these operations concurrently leading to a faster, more efficient procedure.” It would help to evaluate these methods as a comparison. Without this comparison, the difference between updating the other networks concurrently vs iteratively seems superficial (e.g., can’t you just run the prior methods with a faster iteration loop)?

There are also other claims that seem unsubstantiated. For example:

“it ensures that the information-gain from Shaper encouraging Controller to explore a given set of states is sufficiently high to merit activating the stream of rewards”
Is there evidence or proof of information being maximized by the Shaper?

“Controller’s rewards only at states that are important for guiding Controller to its optimal policy.
This enables Shaper to quickly determine its policy π2 and how to choose the values of F unlike Controller whose policy must learned for all states”
Why does this enable the Shaper to quickly determine its policy? Is there evidence of this?


## Experiments
One limitation of the experiments is that the method is only applied to a relatively easy robot environment and a limited number of video game domains. In particular, on Solaris and Super Mario, most of the learning happens within the first few time steps, suggesting that exploration is not a big challenge in these domains. Although the authors do demonstrate that the other methods have some instability issues, it would be great to test the environment on more domains where the bottleneck is clearly exploration.

It would be great to compare to [3,4] since, as the authors say, these are “closest to our work”. Although I understand that [3] requires a shaping reward to begin with, ROSA also assumes access to a bonus (through L(s)), and so comparing to [3] with the same RND bonus seems like a fair comparison.


## Minor comments

The statement “Unlike [15] which requires a useful shaping reward to begin with” seems unfair since the whole point of [15] is that you can learn when to ignore a non-useful shaping reward.

“ Moreover, in [15,34], the agent’s policy and shaping rewards are learned with consecutive updates. In contrast, ROSA performs these operations concurrently leading to a faster, more efficient procedure.”
As written, the difference sounds rather superficial because it’s relatively simple to run [15, 34] concurrently. Do the authors mean to highlight the difference between a bi-level optimization and having a separate Shaper policy that can generate on-the-fly actions?

In Figure 1, why does your method start off better at steps=0 than RND and count-Based ICM? Shouldn’t they start at the same performance?
Could the authors report normalized returns, which I believe is more standard for Atari?

“Another issue is that using an optimisation procedure to find θ directly does not make use of information generated by intermediate state-action-reward tuples of the RL problem which can help to guide the optimisation”
I’m not sure what problem is being highlighted here. Are you saying that trying to optimzie theta without using any samples is a bad idea? While I agree, it’s unclear why anyone would do this.

“Now the problem is to find θ...such that F(st+1, st) = \hat F(st+1, st; θ)”
What is F? Is it just any “useful shaping-reward function F” or is it a specific F?

“these methods provide no performance guarantees nor do they ensure the optimal policy (of the underlying MDP) is preserved”
Some curiosity-based reward shaping methods provide guarantees (though “curiosity” is a bit ambiguous of a category). See [1,2]


## typos/low-level:
- during training. [16].
- Moreover, they naively reward exploration to unvisited states without consideration of the environment reward
- when Shaper influences Controller)
- Figure legends and axes labels are very small and hard to read
[1] Hazan, Elad, et al. "Provably efficient maximum entropy exploration." International Conference on Machine Learning. PMLR, 2019.
[2] Misra, Dipendra, et al. "Kinematic state abstraction and provably efficient rich-observation reinforcement learning." International conference on machine learning. PMLR, 2020.
[3]  Yujing Hu et al. “Learning to Utilize Shaping Rewards: A New Approach of Reward Shaping”. In: Advances in Neural Information Processing Systems 33 (2020).
[4] Bradly Stadie, Lunjun Zhang, and Jimmy Ba. “Learning Intrinsic Rewards as a Bi-Level
Optimization Problem”. In: Conference on Uncertainty in Artificial Intelligence. PMLR. 2020,
pp. 111–120.

**Summary Of The Paper:**

The authors propose ROSA, a reward shaping method that trains a separate “Shaper” policy to learn how to generate reward bonuses. The problem is formulated as a two-player Markov game, in which the environment dynamics are only affected by the primary, “Controller” policy, but the Shaper and Controller policies each optimize their own rewards. The Controller’s reward is the normal reward plus a potential-based reward that depends on the previous two states and the previous two Shaper actions. The shapers’ reward is a combination of the Controller’s reward, a penalty for switching often, and a count-based exploration bonus. The shaper’s rewards are also automatically gated by a switching function, which is hard-coded to switch with higher probability whenever a curiosity-metric increases (in this case using the RND exploration metric). The authors extend the standard reward shaping result to show that including the actions of the Shaper in the potential-based reward shaping term maintains policy invariance. Then, the authors empirically demonstrate on a grid-world environment that ROSA provides useful reward shaping that guides the agent towards the goal and ignores irrelevant parts of the state space. The authors also compare to and find that ROSA outperforms ICM, RND, PPO, and LIRPG on Gravitar, and gets similar performance on Solaris and Super Mario as some of these prior works.

**Summary Of The Review:**

The paper presents an interesting idea and has some promising results. However, there are a number of clarity issues (including lack of details on an important design), claims that seem unsupported, and limitations in the experiment (in particular, not comparing to certain baselines and evaluating on tasks that may not be challenging enough) that make me currently believe that the paper is not ready.

---

> ### Author Response · Authors · 2021-11-26
> **Authors' response**
>
> We thank the reviewer for their careful reading and valuable comments. We respond to your points individually.
>
> **In the reviewer’s summary it is written: ”The shaper’s rewards are automatically gated by a switching function, which is hard-coded to switch with higher probability whenever a curiosity-metric increases (...using the RND exploration metric)**
>
> **A.** This is a misconception. The shaper learns when to activate the shaping rewards. This is central aspect to the workings of ROSA. We demonstrate that the shaper learns to activate shaping rewards at the most relevant states in Exp. 1, please see the heatmap in Fig. 1. This is formally proven in Prop. 2.
>
> **How are $L, c, I, \phi$ chosen.**
>
> **A.**
>
> * $L$ can be any exploration bonus term; we used an RND. We have now included a study in Sec. 13 showing ROSA performs well even when $L$ is fixed to be a rudimentary count-based bonus – therefore ROSA is not sensitive to the choice of this term.
>
> * $I$ is not chosen, this is a misconception. When the Shaper activates the shaping rewards $F$ using its policy, the times of these activations are denoted by $\tau_1, \tau_3, \ldots$. The role of $I$ in the controller’s objective simply expresses that $F$ is activated at the points $\\{\tau_{2k+1}\\}$ (here $I=1$) and continues to be added until points $\\{\tau_{2k}\\}|_{k>0}$ (here $I=0$).
>
> * $c$ - this simply introduces a cost each time the shaper performs an action. In keeping with switching control methods, this is simply chosen to be a very small positive constant.
>
> * $\phi$ - the construction is described on Pg. 6, moew details are in Sec. 9. To explain, $\phi(s,a)=f(s)\cdot i(a^2)$ where $i(a^2)$ is a one hot encoding of the Shaper’s action and $f$ is a fixed neural network (NN). Note that as we show in the Ablation study in Sec. 14, ROSA's performance is not sensitive to the choice of weights for the NN.
>
>
> **Claims**
>
> **(1) learning the reward shaping term concurrently not iteratively (2) avoiding the computational cost of computing the reward at every state.**
>
> **A.** Claim 1 immediately follows from the fact that shaper and controller learn simultaneously and make decisions at each time step. This is manifest in the framework and algorithm.
>
> **A.** In fact, Claim 2 is a very important feature as it means that the Shaper can learn its shaping reward rapidly relative to the controller- the latter must perform optimisations over its complete action space at each state. After discarding a subset of states to add shaping rewards, Shaper must only learn which shaping rewards it should add at a subset of states. We refer the reviewer to the heatmap in Fig. 1 which shows the subset of states that shaper learns to optimise its shaping rewards over.
>
> **”Why does this enable the Shaper to quickly determine its policy? Is there evidence of this?”**
>
> **A.** The decision space for the switching policy is $|\mathcal{S}|\times\\{0,1\\}$ i.e at each state it makes a binary decision. Consequently, this component of the learning process is much quicker than the controller's policy which must optimise over a decision space which is $|\mathcal{S}||\mathcal{A}|$ (choosing an action from its action space at every state). This results in the shaper learning its policy rapidly (relative to the controller) which guides the controller agent towards its optimal policy during its learning phase.
>
> **The statement “Unlike [15] which requires a useful shaping reward to begin with” seems unfair**
>
> **A.** While we agree with the reviewer that the referenced paper learns to ignore the shaping reward function when it is not useful, that method requires that a shaping reward function be provided from the outset. This is unlike our method that constructs that shaping rewards from scratch.
>
> **Experiments.**
>
> **A.** We used a range of benchmark environments that appear within reward shaping papers to allow us to perform easy comparisons e.g. [3], [4]. We also added a series of more complex experiments in Atari to further support our empirical evaluations.
>
> **In the intro: "Is it a specific F?”**
>
> **A.** As is typical in the reward-shaping literature [1], [2], in our preliminary discussion, we describe the desired $F$ as one that promotes faster, efficient learning. In keeping with the reward shaping literature, we do not specify a functional form of $F$ given the non-uniqueness of a function that fulfils this role.
>
> **Typos**
>
> **A.** Thanks to the reviewer for pointing out the typos. We will polish the paper and make the various amendments.
>
> [1] A. Ng, et al.. “Policy invariance under reward transformations: Theory and application to reward shaping”. ICML, 1999
>
> [2] S. Devlin, et al. “Theoretical considerations of potential-based reward shaping for multi-agent systems”. AAMAS, 2011
>
> [3] B. Stadie, et al. “Learning Intrinsic Rewards as a Bi-Level Optimization Problem”, UAI, 2020
>
> [4] Y. Hu et al. “Learning to Utilize Shaping Rewards: A New Approach of Reward Shaping”, NeurIPS, 2020

---

> > ### Comment · Reviewer_uREp · 2021-11-26
> > **Re: Author's response**
> >
> > ## The shaper’s rewards are automatically gated by a switching function...
> >
> > > A. This is a misconception. The shaper learns when to activate the shaping rewards. This is central aspect to the workings of ROSA. We demonstrate that the shaper learns to activate shaping rewards at the most relevant states in Exp. 1, please see the heatmap in Fig. 1. This is formally proven in Prop. 2.
> >
> > I was referring to Equation 2 in the appendix (10 Shaper Termination Time), which is a hard-coded rule. Is there a derivation of this rule? Otherwise, it seems like a hard-coded heuristic.
> >
> >
> > ## How are parameters chosen.
> >
> > Thank you for the additional experiment with a different exploration bonus.
> >
> > > I  is not chosen, this is a misconception.
> >
> > Again, Equation 2 seems to be a design choice. Of course, once the equation of I is chosen, the output is computed automatically, but this equation is chosen.
> >
> > > this is simply chosen to be a very small positive constant.
> >
> > I understand this. My concern is that choosing additional hyperparameters can be difficult. Is there a sweep that demonstrates that this value is easy to choose (E.g. it can range from 1e-5 to 0.1)?
> >
> > Thank you for explaining how $\phi$ is chosen and demonstrating that the method is no sensitive to it.
> >
> > ## Additional Results
> >
> > > We also added a series of more complex experiments in Atari to further support our empirical evaluations.
> >
> > Where and what are these results?

---

> > > ### Author Response · Authors · 2021-11-27
> > > **Re:**
> > >
> > > **Thanks for the additional feedback**
> > >
> > > Thanks for clarifying what you are referring to. For the first point about the switching interventions, the key point that we highlight is that the activations of the shaping rewards are chosen by the shaper. We agree with the reviewer that the specific choice of termination criteria we have considered is in partly a heuristic. Having a random part for the termination times ensures the Markov property. As described in Sec. 10, the criterion we have used is based on the insight that this random termination time can also have a dependence on the information gain which is described by $\Delta L$.
> > >
> > > **Choice of intervention cost parameter $c$**
> > >
> > > Thanks for the comment. The role of this parameter is simply to ensure that shaper is no longer indifferent about adding a shaping reward at irrelevant states. For this reason we just simply set the reward to a very small positive constant (without tuning). In our experience this parameter did not require special tuning - we can easily include evidence of this in the paper.
> > >
> > > **Results**
> > >
> > > We mean that we have considered a very broad range of experimental environments with respect to similar studies on reward shaping. For example [16], a very relevant baseline consider Mujoco and cartpole while also a very relevant baseline [43] consider Atari and Mujoco and there it is stated that their "results show improved performance on most but not all of the domains"

---

### Official Review · Reviewer_sEbz · 2021-11-02

**Correctness:** 3
**Technical Novelty And Significance:** 3
**Empirical Novelty And Significance:** 3
**Recommendation:** 6
**Confidence:** 4

**Main Review:**

The idea of jointly training an agent to assist the controller is relevant to the RL community and new (to my knowledge). In my mind, the primary insight is to connect reward shaping to Markov games. The theoretical results built off [25] appear to support the methodology and experiments suggest ROSA works well and is robust (up to the points below). I appreciate that the authors attempted to show the method performs well in various task contexts, e.g., for subgoal discovery.

On the downside, there are several aspects of the work which are unclear to me and several claims are not well established by the theory nor the experiments:

- the paper claims that ROSA's concurrent learning leads to "a faster, more efficient procedure" compared to [15,34]. Can the authors provide additional theoretical or experimental justification for this claim?
- Sec 4.1. on the switching controls can be better explained; I don't understand the definition for the switch being $I_{\tau_{k+1}} = 1 - I_{\tau_k}$; the switch is flipped after each step? This seems at odds with the later explanations about the switching behavior. Looking at the main events, the switch is decided upon at each step?
- Sec 4.2. The role of the bonus reward L remains rather opaque to me. Perhaps I missed it but I didn't see any experiments studying its effect. Perhaps the authors can clarify?
- Sec 5. It is unclear what $v_1^{\pi, \pi^2}$ means. In Sec. 4, it is defined as the objective for controller, incorporating both the extrinsic and intrinsic rewards. However, in Proposition 3, it appears to denote extrinsic expected return. Likewise, the corresponding proof (Proof 7) doesn’t explicitly write out $v_1^{\pi, \pi^2}$.
- Sec 6. Key details on the experimental settings are missing, especially the number of runs. It is well-known that RL performance can vary significantly between random initializations. How does the incorporation of Shaper affect this variance? I appreciate space is tight so perhaps this can be put in the supplementary.
- Sec 3. "Red-herrings"; I'm puzzled as to how Shaper learns to do this since there is a bonus for exploration (L). The mechanism by which the exploration/exploitation trade-off is resolved is not apparent to me from reading the paper.

## Minor comments
- Sec 4: "RIGA" should be "ROSA"
- Sec 5: "The result is established by a careful adaptation of the policy invariance result in [25] ...." is repeated after Corollary 1.


**Summary Of The Paper:**

The key idea in this paper is to jointly train a pair of agents, namely a Controller that performs the RL task and a Shaper agent that shapes the Controller's reward function to better its performance. Rather than shape all states, Shaper learns "switching controls" to determine states on which to place its modeling effort on, and ablation experiments suggest this works well compared to the straightforward approach of shaping all states. A natural concern is whether this Markov game approach results in stable training and convergence. To address this issue, the authors provide theoretical convergence results which show ROSA convergences to a Nash Equilibrium (NE) with weakly higher total return. Experiments on several domains suggest the method works well relative to alternative reward shaping approaches (e.g., those based on curiosity or bi-level optimization).

**Summary Of The Review:**

The paper makes interesting contributions, but the clarity could be improved. At this point, my score is a reflection of my uncertainty about the above issues.

---

> ### Author Response · Authors · 2021-11-26
> **Authors' response**
>
> We thank the reviewer for their careful reading and valuable comments. We respond to your points individually.
>
> **Claim that ROSA's concurrent learning leads to "a faster, more efficient procedure"**
>
> **A.** Key to ROSA is the switching control framework. It reduces the set of states that the Shaper adds shaping rewards to only those that are important to improving the controller's payoff. at each state it makes a binary decision  so that the decision space for the switching policy is just $|\mathcal{S}|\times\\{0,1\\}$. Consequently, this component of the learning process is quick. This is to be compared with the controller's learning process which must optimise over a decision space which is $|\mathcal{S}||\mathcal{A}|$ (choosing an action from its action space at every state) and as with other shaping reward methods that may seek to attempt to learn shaping reward magnitudes at each state. As in our experiments, the size of the action set for Shaper can be a singleton so that Shaper’s entire decision space is $|\mathcal{S}|\times \\{0,1\\}$. This is what produces the computational efficiency.
>
> **Meaning of the switch**
>
> Please firstly note the expression on $I$ (which is the coefficient on $F$) is $I_{\tau_{k+1}}=1-I_{\tau_{k}}$ - involving the switching times not $I_{t+1}=1- I_t$ not time steps as the reviewer’s remark suggest. When Shaper activates the shaping rewards $F$, the times of these activations are denoted by $\tau_1, \tau_3, \ldots$ (as we explain on Pg. 4, $\tau_1=4$ and $\tau_2 =8$, means that the shaping rewards are activated at times steps $t=4,5,6,7$). As described in the *Summary of Events*, when I is off, shaper decides at each time step whether or not to activate the shaping reward using its switching policy frak$g_2$. The switch $I$ expresses that $F$ is activated at the points $\\{\tau_{2k+1}\\}$ (here $I=1$) and continues to be added until points $\\{\tau_{2k}\\}|_{k>0}$ (here $I=1-1=0$).
>
> **Role of the bonus reward $L$**
>
> **A.** The role of $L$ is to reward shaper for inducing exploration by the agent of unvisited states. As explained on paragraph 1 on Pg. 6, we used an RND for $L$.  We note that ROSA markedly outperforms both PPO with RND and is able to significantly augment the benefits of applying RND directly to the agents’ objectives. This is due to the added benefit of switching controls and shaping reward selection performed by Shaper. Following the reviewer’s suggestion to see more analysis of this term, we have now also included a study that shows that ROSA performs well even if the RND is replaced by a rudimentary count-based method, please see Sec. 13.
>
> **Meaning of $v_1^{\pi,\pi^2}$ (and $v_1^{\pi}$)**
>
> $v_1^{\pi,\pi^2}$ is the total expected return for controller with the shaping rewards being chosen according the shaper’s policy $\pi^2$ (as the reviewer points out).  $v_1^{\pi}$ is the total expected return for controller with no shaping rewards included (c.f. Sec. 3). Prop. 3, therefore provides the useful result that adding shaping rewards produced by ROSA never decreases the agent’s expected return unlike the case for other reward-shaping methods where this decrease is possible.
>
> **Details on number of runs.**
>
> **A.** Many thanks to the reviewer for this suggestion. We will update the paper to include this detail.
>
> **How shaper learns to tackle "Red-herrings"**
>
> **A.** Thanks to the reviewer for this question. This highlights a major benefit of the two-player framework. As we explain on Pg. 2, shaper has its own distinct objective which seeks to improve the return for controller and, its own policy. This enables Shaper to introduce shaping rewards that promote complex exploration patterns. Crucially, the switching control aspect of Shaper’s intervention means that it can choose not to add shaping rewards at states that are not helpful.
>
> **Typos**
>
> **A.** Thanks to the reviewer for pointing out the typos. We will polish the paper and make the various amendments.

---

> > ### Comment · Reviewer_sEbz · 2021-11-29
> > **Thanks for your response**
> >
> > Thank you for your response and for clarifying my questions wrt the switching variable and also about the bonus reward. My opinion remains that this is an interesting paper with some limitations.
> >
> > With regards to Prop 3, the explanation of the proposition should be improved. the paper states that Prop 3 compares the **environment (extrinsic)** rewards accrued by the agents doesn't decrease with Shaper, but this is not what the specific statement says since "$v_1^{\pi. \pi^2}(s, \cdot) \geq v_1^\pi(s), \forall s$" means that the rewards **including the shaping** is larger than without. A distinction should be made between the extrinsic/environmental rewards and those provided by Shaper.
> >
> > Minor
> > - "Theorem 3" should be "Prop 3"

---

> > > ### Author Response · Authors · 2021-11-30
> > > **Re: Thanks for your response**
> > >
> > > Thank you to the reviewer for their further comments.
> > >
> > > We would like to resolve the point about Prop. 3 which in fact does not require the distinction the reviewer has suggested.
> > >
> > > To see this, we refer the reviewer to Prop. 1 which clarifies this in full. As we state in the paper, the result of Prop. 1 proves that the (expected) total return received by the (controller) agent under the influence of the shaper is that from the environment (extrinsic rewards) only. With this our statement of Prop. 3 can immediately be deduced.
> > >
> > > Thanks for pointing out the minor typo.

---

> > > > ### Comment · Reviewer_sEbz · 2021-12-01
> > > > **Thanks and understood!**
> > > >
> > > > Thanks for clarifying the point about Prop 3! I see now how the argument is structured.

---

### Official Review · Reviewer_g7fZ · 2021-11-07

**Correctness:** 2
**Technical Novelty And Significance:** 2
**Empirical Novelty And Significance:** 1
**Recommendation:** 3
**Confidence:** 3

**Main Review:**

This paper aims to tackle an important problem in RL and does a good job of conducting some experiments to show what the method learns. However, there are some significant concerns with the paper. The first is a lack of clarity in the presentation of the method and problem setup. The second is that it is unclear where or if any benefit is coming from the proposed method. The issues are expanded below.



Clarity:
The paper’s clarity can be improved by being more precise in writing and avoiding redundant arguments.

The first thing to fix is to make sure every term and symbol has a definition before it is used. In section, when an expression for F is given, phi is used, but it is unclear what phi is required to be. This creates confusion later Section 3 third paragraph, where F hat uses some phi hat, and \theta^\star is supposed to make \hat \phi equal to \phi. \theta^\star is often used to indicate optimal weights, but it is not clear what optimal is because phi is undefined. It is said that \theta^* is supposed to yield a useful shaping reward, but it is unclear what that means. Without proper definitions, it is not easy to understand the proper motivation and reason about whether the method accomplishes the stated goals.

There are also many redundant arguments made throughout the paper. For example, at the start of the third paragraph in Section 3, an argument is given that was discussed several times above.


At the start of Section 4, further discussion is given about the goal/objective for each learner, but they are both not provided until later. This lack of specificity is frustrating as a reader because all statements given so far have been vague and the exact relationship of the two algorithms continues to go misunderstood. Also, for both objective functions, it is specified that each learner needs to maximize a value function at some given state, but this is not a proper objective function because a distribution or weighting over states is not given.

Method benefits:
The stated objective is to tackle the problem of sparse and uninformative rewards through autonomously learning a reward function. Additionally, it is said that the “…shaping rewards supplement the environment reward and promote effective learning, our framework therefore addresses the key challenges in RS”. One issue with this claim is that learning of the shaping rewards cannot be more efficient than just learning about the reward function of the environment, i.e., what makes a good reward function can only be learned once the agent has already learned what the good behavior is. Thus, it seems unlikely that this method could be doing anything more effective than encouraging exploration. This brings up another issue, in that it is unclear what components of the method are necessary or helpful in solving sparse reward problems. The experiments demonstrate some learned behavior, but they do not clearly show how each component helps the algorithm change the learning dynamics, so sparse reward problems are manageable. These experiments are an essential step in proposing new methods so other researchers can build off the knowledge generated in a paper to solve new and different problems.


**Summary Of The Paper:**

This paper introduces a new algorithm to solve the sparse reward settings. The algorithm tries to find an optimal policy by leveraging two learners: a controller and a shaper. The controller learns to maximize the environment reward signal plus the signal provided by the shaper. The shaper learns a potential-based reward shaping function by maximizing the controller's objective plus a cost penalty for providing reward feedback and an exploration bonus. The shapers reward function is created using a randomly generated state function and a dot product based on the shaper's "action." Some theoretical analysis is provided along with experiments demonstrate to demonstrate that this algorithm is helping solve sparse reward problems.


**Summary Of The Review:**

This paper lacks proper justification for the design choices of the presented method and is thus not ready for acceptance.

---

> ### Author Response · Authors · 2021-11-25
> **Authors' response**
>
> We thank the reviewer for their reading and many comments.
>
> It seems that the reviewer has missed many important aspects of the paper and has several misconceptions/made incorrect assertions which we now seek to resolve.
>
> We address the reviewer's comments individually:
>
> **In the reviewer's summary it is written: "The shapers reward function is created using a randomly generated state function and a dot product based on the shaper's "action."**
>
> Please note that this is not the shaper's reward function (as the reviewer has written) but this describes the shaping function that enters the controller's objective. The "state function" is a fixed neural network whose weights are randomly chosen (we show in Sec. 14 that the choice of these randomly chosen weights has no significant affect on overall performance).
>
> **”Learning the shaping rewards cannot be more efficient than just learning the reward function”**
>
> **A.** In fact the reviewer’s statement is incorrect for the reasons we now explain. Crucially, in our case, for the Shaper to guide the agent towards the optimal trajectory, it is not necessary to add optimal intrinsic rewards at all states nor is it required to determine the environment rewards of the environment (whose values can be drawn from a large range) at all states. This is to be contrasted with the computation required to learn the reward function from (possibly noisy) reward signals at all states.  The switching control is a key aspect of our framework that exploits these facts with which the shaper learns only to tune the shaping rewards at the most relevant state. Note also the reviewer's statement overlooks the benefits offered by reward shaping  such as the ability to increase exploration efficiency and generate subgoals in contrast to learning the reward function (at all states!) by supervised learning or otherwise.
>
> This point is exemplified in Exp. 1, where it is shown that the Shaper adds new intrinsic rewards (while preserving the MDP) in a small subset states that are most relevant to guiding the agent to the optimal goal state. This results in more efficient learning and higher performance in the given number of training episodes.
>
> **”It seems unlikely that this method could be doing anything more effective than encouraging exploration”**
>
> **A.** The reviewer’s assertion is incorrect since as we explain in a background discussion in Sec. 2, a key purpose of reward-shaping in general is to densify the rewards received by the agent. This is again exemplified in Exp. 1, where it is shown that the Shaper adds new intrinsic rewards (while preserving the MDP) at a set of states that guide the agent towards its optimal trajectory. Moreover, as is shown in Experiment 2, our framework performs the task of subgoal discovery – in that same experiment it is shown that pure exploration based methods do not achieve comparable performance.
>
> **“It is unclear what components of the method are necessary or helpful in solving sparse reward problems”**
>
> **A.** A key aspect of the shaping reward methodology is to densify rewards. Since the shaping-reward adds rewards to supplement the environment (extrinsic) reward, the helpfulness of the framework in tackling sparse reward environment follows from its ability to add useful shaping rewards. We also refer the reviewer to our experiment in Sparse Cartpole in Sec. 6 which by construction is a sparse reward problem. There the agent only receives a penalty of $-1$ if the pole collapses and receives no other rewards otherwise. In this experiment ROSA outperforms all other baselines except for the version of BiPars which has a manually engineered shaping function already included.
>
> **“It is not clear what optimal is because $\phi$ is undefined and, $\theta^{\star}$”**
>
> **A.** We refer the reviewer to the discussion on the standard technology of reward shaping in Sec. 2 where we introduce $\phi$, which is a (potential) function which is defined by its relationship through $F$. As is typical in the reward-shaping literature [1], [2], the shaping function $F$ we seek is one that promotes faster, efficient learning. In keeping with the reward shaping literature, we do not specify a functional form of $F$ given the non-uniqueness of a function that fulfils this role.
>
> **“The objective is not proper objective function because a distribution or weighting over states is not given”**
>
> **A.** We state the probability transition kernel $P$ that specifies the distribution over states at the beginning of Sec. 3. As is usual throughout reinforcement learning literature, to avoid cumbersome notation, we suppress the measure $P$ under which the expectation in our objective is defined.
>
> [1] A. Ng, D. Harada, S. Russell. “Policy invariance under reward transformations: Theory and application to reward shaping”. ICML, 1999
> [2] S. Devlin, D Kudenko. “Theoretical considerations of potential-based reward shaping for multi-agent systems”. AAMAS, 2011

---

### Author Response · Authors · 2021-11-30
**Response to ALL REVIEWERS**

Dear Reviewers

Thank you for your careful reading of our paper and your insightful comments. We would like the authors and AC to consider the novelty of the framework which introduces a new way of performing reward-shaping autonomously. We believe that our paper offers a very good first step for developing this new approach towards tackling issues such as reward sparsity and subgoal discovery. We have now added all the additional material requested by the reviewers. We have also put our best efforts into giving the paper a thorough polish and clarifying the answers to your questions.

This includes:

• **In response to Reviewers sEbz, uREp, XHiG**: we added a study of the impact of (parameter) choices on the framework. Specifically, we showed that LIGS accommodates different bonus terms in Generator's objective and performs well even when  is a rudimentary count-based bonus.

• **In response to Reviewers sEbz, uREp,XHiG**: we performed an ablation study of the different parameter weights in the $\phi$ function. Please note the paper contains full details of the construction of $\phi$ in Sec. 4.3 and Sec. 9.

• **As requested by Reviewers g7fZ, XHiG**: we have now improved the presentation and structure of the paper.

We have also written out explanations to explain some of the small misconceptions that some reviewers had.

We would appreciate if the reviewers can inspect our updates and consider our responses to see that they are now satisfied with our responses.

---

### Decision · Program_Chairs · 2022-01-20

**Decision:**

Reject

**Comment:**

This paper tackled the reward shaping problem under the framework of Markov games. The authors proposed reward shaping algorithms for RL with mild theoretical guarantees. The AC agrees with the reviewers that the empirical performance is ambiguous. The paper should be substantially improved before being accepted.